# Continuous Mean-Covariance Bandits

**Yihan Du**
IIIS, Tsinghua University
Beijing, China
duyh18@mails.tsinghua.edu.cn

**Siwei Wang**
CST, Tsinghua University
Beijing, China
wangsw2020@mail.tsinghua.edu.cn

**Zhixuan Fang**
IIIS, Tsinghua University, Beijing, China
Shanghai Qi Zhi Institute, Shanghai, China
zfang@mail.tsinghua.edu.cn

**Longbo Huang**[*]
IIIS, Tsinghua University
Beijing, China
longbohuang@mail.tsinghua.edu.cn

## Abstract

Existing risk-aware multi-armed bandit models typically focus on risk measures of individual options such as variance. As a result, they cannot be directly applied to important real-world online decision making problems with correlated options. In this paper, we propose a novel Continuous Mean-Covariance Bandit (CMCB) model to explicitly take into account option correlation. Specifically, in CMCB, there is a learner who sequentially chooses weight vectors on given options and observes random feedback according to the decisions. The agent's objective is to achieve the best trade-off between reward and risk, measured with option covariance. To capture different reward observation scenarios in practice, we consider three feedback settings, i.e., full-information, semi-bandit and full-bandit feedback. We propose novel algorithms with optimal regrets (within logarithmic factors), and provide matching lower bounds to validate their optimalities. The experimental results also demonstrate the superiority of our algorithms. To the best of our knowledge, this is the first work that considers option correlation in risk-aware bandits and explicitly quantifies how arbitrary covariance structures impact the learning performance. The novel analytical techniques we developed for exploiting the estimated covariance to build concentration and bounding the risk of selected actions based on sampling strategy properties can likely find applications in other bandit analysis and be of independent interests.

## 1 Introduction

The stochastic Multi-Armed Bandit (MAB) [3, 28, 2] problem is a classic online learning model, which characterizes the exploration-exploitation trade-off in decision making. Recently, due to the increasing requirements of risk guarantees in practical applications, the Mean-Variance Bandits (MVB) [26, 30, 34] which aim at balancing the rewards and performance variances have received extensive attention. While MVB provides a successful risk-aware model, it only considers discrete decision space and focuses on the variances of individual arms (assuming independence among arms).

However, in many real-world scenarios, a decision often involves multiple options with certain correlation structure, which can heavily influence risk management and cannot be ignored. For instance, in finance, investors can select portfolios on multiple correlated assets, and the investment risk is closely related to the correlation among the chosen assets. The well-known "risk diversification" strategy [4] embodies the importance of correlation to investment decisions. In clinical trials, a

---

[*]Corresponding author.

35th Conference on Neural Information Processing Systems (NeurIPS 2021).

treatment often consists of different drugs with certain ratios, and the correlation among drugs plays an important role in the treatment risk. Failing to handle the correlation among multiple options, existing MVB results cannot be directly applied to these important real-world tasks.

Witnessing the above limitation of existing risk-aware results, in this paper, we propose a novel Continuous Mean-Covariance Bandit (CMCB) model, which considers a set of options (base arms) with continuous decision space and measures the risk of decisions with the option correlation. Specifically, in this model, a learner is given $d$ base arms, which are associated with an unknown joint reward distribution with a mean vector and covariance. At each timestep, the environment generates an underlying random reward for each base arm according to the joint distribution. Then, the learner selects a weight vector of base arms and observes the rewards. The goal of the learner is to minimize the expected cumulative regret, i.e., the total difference of the reward-risk (mean-covariance) utilities between the chosen actions and the optimal action, where the optimal action is defined as the weight vector that achieves the best trade-off between the expected reward and covariance-based risk. To capture important observation scenarios in practice, we consider three feedback settings in this model, i.e., full-information (CMCB-FI), semi-bandit (CMCB-SB) and full-bandit (CMCB-FB) feedback, which vary from seeing rewards of all options to receiving rewards of the selected options to only observing a weighted sum of rewards.

The CMCB framework finds a wide range of real-world applications, including finance [23], company operation [24] and online advertising [27]. For example, in stock markets, investors choose portfolios based on the observed prices of all stocks (full-information feedback), with the goal of earning high returns and meanwhile minimizing risk. In company operation, managers allocate investment budgets to several correlated business and only observe the returns of the invested business (semi-bandit feedback), with the objective of achieving high returns and low risk. In clinical trials, clinicians select a treatment comprised of different drugs and only observe an overall therapeutic effect (full-bandit feedback), where good therapeutic effects and high stability are both desirable.

For both CMCB-FI and CMCB-SB, we propose optimal algorithms (within logarithmic factors) and establish matching lower bounds for the problems, and contribute novel techniques in analyzing the risk of chosen actions and exploiting the covariance information. For CMCB-FB, we develop a novel algorithm which adopts a carefully designed action set to estimate the expected rewards and covariance, with non-trivial regret guarantees. Our theoretical results offer an explicit quantification of the influences of arbitrary covariance structures on learning performance, and our empirical evaluations also demonstrate the superior performance of our algorithms.

Our work differs from previous works on bandits with covariance [32, 33, 11, 25] in the following aspects. (i) We consider the reward-risk objective under continuous decision space and stochastic environment, while existing works study either combinatorial bandits, where the decision space is discrete and risk is not considered in the objective, or adversarial online optimization. (ii) We do not assume a prior knowledge or direct feedback on the covariance matrix as in [32, 33, 11]. (iii) Our results for full-information and full-bandit feedback explicitly characterize the impacts of arbitrary covariance structures, whereas prior results, e.g., [11, 25], only focus on independent or positively-correlated cases. These differences pose new challenges in algorithm design and analysis, and demand new analytical techniques.

We summarize the main contributions as follows.

- We propose a novel risk-aware bandit model called continuous mean-covariance bandit (CMCB), which considers correlated options with continuous decision space, and characterizes the trade-off between reward and covariance-based risk. Motivated by practical reward observation scenarios, three feedback settings are considered under CMCB, i.e., full-information (CMCB-FI), semi-bandit (CMCB-SB) and full-bandit (CMCB-FB).

- We design an algorithm `MC-Empirical` for CMCB-FI with an optimal $O(\sqrt{T})$ regret (within logarithmic factors), and develop a novel analytical technique to build a relationship on risk between chosen actions and the optimal one using properties of the sampling strategy. We also derive a matching lower bound, by analyzing the gap between hindsight knowledge and available empirical information under a Bayesian environment.

- For CMCB-SB, we develop `MC-UCB`, an algorithm that exploits the estimated covariance information to construct confidence intervals and achieves the optimal $O(\sqrt{T})$ regret (up to

logarithmic factors). A matching regret lower bound is also established, by investigating the necessary regret paid to differentiate two well-chosen distinct instances.

- We propose a novel algorithm MC-ETE for CMCB-FB, which employs a well-designed action set to carefully estimate the reward means and covariance, and achieves an $O(T^{\frac{2}{3}})$ regret guarantee under the severely limited feedback.

To our best knowledge, our work is the *first* to explicitly characterize the influences of *arbitrary* covariance structures on learning performance in risk-aware bandits. Our results shed light into optimal risk management in online decision making with correlated options. Due to space limitation, we defer all detailed proofs to the supplementary material.

## 2   Related Work

**(Risk-aware Bandits)** Sani et al. [26] initiate the classic mean-variance paradigm [23, 14] in bandits, and formulate the mean-variance bandit problem, where the learner plays a single arm each time and the risk is measured by the variances of individual arms. Vakili & Zhao [29, 30] further study this problem under a different metric and complete the regret analysis. Zhu & Tan [34] provide a Thompson Sampling-based algorithm for mean-variance bandits. In addition to variance, several works consider other risk criteria. The VaR measure is studied in [10], and CVaR is also investigated to quantify the risk in [13, 16]. Cassel et al. [5] propose a general risk measure named empirical distributions performance measure (EDPM) and present an algorithmic framework for EDPM. All existing studies on risk-aware bandits only consider discrete decision space and assume independence among arms, and thus they cannot be applied to our CMCB problem.

**(Bandits with Covariance)** In the stochastic MAB setting, while there have been several works [11, 25] on covariance, they focus on the combinatorial bandit problem without considering risk. Degenne & Perchet [11] study the combinatorial semi-bandits with correlation, which assume a known upper bound on the covariance, and design an algorithm with this prior knowledge of covariance. Perrault et al. [25] further investigate this problem without the assumption on covariance under the sub-exponential distribution framework, and propose an algorithm with a tight asymptotic regret analysis. In the adversarial setting, Warmuth & Kuzmin [32, 33] consider an online variance minimization problem, where at each timestep the learner chooses a weight vector and receives a covariance matrix, and propose the exponentiated gradient based algorithms. Our work differs from the above works in the following aspects: compared to [11, 25], we consider a continuous decision space instead of combinatorial space, study the reward-risk objective instead of only maximizing the expected reward, and investigate two more feedback settings other than the semi-bandit feedback. Compared to [32, 33], we consider the stochastic environment and in our case, the covariance cannot be directly observed and needs to be estimated.

## 3   Continuous Mean-Covariance Bandits (CMCB)

Here we present the formulation for the Continuous Mean-Covariance Bandits (CMCB) problem. Specifically, a learner is given $d$ base arms labeled $1, \ldots, d$ and a decision (action) space $\mathcal{D} \subseteq \triangle_d$, where $\triangle_d = \{\boldsymbol{w} \in \mathbb{R}^d : 0 \leq w_i \leq 1, \forall i \in [d], \sum_i w_i = 1\}$ denotes the probability simplex in $\mathbb{R}^d$. The base arms are associated with an unknown $d$-dimensional joint reward distribution with mean vector $\boldsymbol{\theta}^*$ and positive semi-definite covariance matrix $\Sigma^*$, where $\Sigma_{ii}^* \leq 1$ for any $i \in [d]$ without loss of generality. For any action $\boldsymbol{w} \in \mathcal{D}$, which can be regarded as a weight vector placed on the base arms, the instantaneous reward-risk utility is given by the following *mean-covariance* function

$$f(\boldsymbol{w}) = \boldsymbol{w}^\top \boldsymbol{\theta}^* - \rho \boldsymbol{w}^\top \Sigma^* \boldsymbol{w}, \tag{1}$$

where $\boldsymbol{w}^\top \boldsymbol{\theta}^*$ denotes the expected reward, $\boldsymbol{w}^\top \Sigma^* \boldsymbol{w}$ represents the risk, i.e., reward variance, and $\rho > 0$ is a risk-aversion parameter that controls the weight placed on the risk. We define the optimal action as $\boldsymbol{w}^* = \text{argmax}_{\boldsymbol{w} \in \mathcal{D}} f(\boldsymbol{w})$. Compared to linear bandits [1, 17], the additional quadratic term in $f(\boldsymbol{w})$ raises significant challenges in estimating the covariance, bounding the risk of chosen actions and deriving covariance-dependent regret bounds.

At each timestep $t$, the environment generates an underlying (unknown to the learner) random reward vector $\boldsymbol{\theta}_t = \boldsymbol{\theta}^* + \boldsymbol{\eta}_t$ according to the joint distribution, where $\boldsymbol{\eta}_t$ is a zero-mean noise vector and it is

---
**Algorithm 1** MC-Empirical
---
1: **Input:** Risk-aversion parameter $\rho > 0$.
2: Initialization: Pull action $\boldsymbol{w}_1 = (\frac{1}{d}, \ldots, \frac{1}{d})$, and observe $\boldsymbol{\theta}_1 = (\theta_{1,1}, \ldots, \theta_{d,1})^\top$. $\hat{\theta}_{1,i}^* \leftarrow \theta_{1,i}, \ \forall i \in [d]$. $\hat{\Sigma}_{1,ij} = (\theta_{1,i} - \hat{\theta}_{1,i}^*)(\theta_{1,j} - \hat{\theta}_{1,j}^*), \ \forall i,j \in [d]$.
3: **for** $t = 2, 3, \ldots$ **do**

4:     $\boldsymbol{w}_t = \underset{\boldsymbol{w} \in \triangle_d}{\operatorname{argmax}}(\boldsymbol{w}^\top \hat{\boldsymbol{\theta}}_{t-1}^* - \rho \boldsymbol{w}^\top \hat{\Sigma}_{t-1} \boldsymbol{w})$
5:     Pull $\boldsymbol{w}_t$, observe $\boldsymbol{\theta}_t = (\theta_{t,1}, \ldots, \theta_{t,d})^\top$
6:     $\hat{\theta}_{t,i}^* \leftarrow \frac{1}{t} \sum_{s=1}^t \theta_{s,i}, \ \forall i \in [d]$
7:     $\hat{\Sigma}_{t,ij} = \frac{1}{t} \sum_{s=1}^t (\theta_{s,i} - \hat{\theta}_{t,i}^*)(\theta_{s,j} - \hat{\theta}_{t,j}^*), \forall i,j \in [d]$
8: **end for**
---

independent among different timestep $t$. Note that here we consider an additive vector noise to the parameter $\boldsymbol{\theta}^*$, instead of the simpler scalar noise added in the observation (i.e., $y_t = \boldsymbol{w}_t^\top \boldsymbol{\theta}^* + \eta_t$) as in linear bandits [1, 17]. Our noise setting better models the real-world scenarios where distinct actions incur different risk, and enables us to explicitly quantify the correlation effects. Following the standard assumption in the bandit literature [22, 11, 34], we assume the noise is sub-Gaussian, i.e., $\forall \boldsymbol{u} \in \mathbb{R}^d$, $\mathbb{E}[\exp(\boldsymbol{u}^\top \boldsymbol{\eta}_t)] \leq \exp(\frac{1}{2}\boldsymbol{u}^\top \Sigma^* \boldsymbol{u})$, where $\Sigma^*$ is unknown. The learner selects an action $\boldsymbol{w}_t \in \mathcal{D}$ and observes the feedback according to a certain structure (specified later). For any time horizon $T > 0$, define the expected cumulative regret as

$$\mathbb{E}[\mathcal{R}(T)] = \sum_{t=1}^T \mathbb{E}[f(\boldsymbol{w}^*) - f(\boldsymbol{w}_t)].$$

The objective of the learner is to minimize $\mathbb{E}[\mathcal{R}(T)]$. Note that our mean-covariance function Eq. (1) extends the popular mean-variance measure [26, 30, 34] to the continuous decision space.

In the following, we consider three feedback settings motivated by reward observation scenarios in practice, including (i) full-information (CMCB-FI), observing random rewards of all base arms after a pull, (ii) semi-bandit (CMCB-SB), only observing random rewards of the selected base arms, and (iii) full-bandit (CMCB-FB), only seeing a weighted sum of the random rewards from base arms. We will present the formal definitions of these three feedback settings in the following sections.

**Notations.** For action $\boldsymbol{w} \in \mathcal{D}$, let $I_{\boldsymbol{w}}$ be a diagonal matrix such that $I_{\boldsymbol{w},ii} = \mathbb{I}\{w_i > 0\}$. For a matrix $A$, let $A_{\boldsymbol{w}} = I_{\boldsymbol{w}} A I_{\boldsymbol{w}}$ and $\Lambda_A$ be a diagonal matrix with the same diagonal as $A$.

## 4  CMCB with Full-Information Feedback (CMCB-FI)

We start with CMCB with full-information feedback (CMCB-FI). In this setting, at each timestep $t$, the learner selects $\boldsymbol{w}_t \in \triangle_d$ and observes the random reward $\theta_{t,i}$ for all $i \in [d]$. CMCB-FI provides an online learning model for the celebrated Markowitz [23, 14] problem in finance, where investors select portfolios and can observe the prices of all stocks at the end of the trading days.

Below, we propose an optimal Mean-Covariance Empirical algorithm (MC-Empirical) for CMCB-FI, and provide a novel regret analysis that fully characterizes how an arbitrary covariance structure affects the regret performance. We also present a matching lower bound for CMCB-FI to demonstrate the optimality of MC-Empirical.

### 4.1  Algorithm for CMCB-FI

Algorithm 1 shows the detailed steps of MC-Empirical. Specifically, at each timestep $t$, we use the empirical mean $\hat{\boldsymbol{\theta}}_t$ and covariance $\hat{\Sigma}_t$ to estimate $\boldsymbol{\theta}^*$ and $\Sigma^*$, respectively. Then, we form $\hat{f}_t(\boldsymbol{w}) = \boldsymbol{w}^\top \hat{\boldsymbol{\theta}}_t - \rho \boldsymbol{w}^\top \hat{\Sigma}_t \boldsymbol{w}$, an empirical mean-covariance function of $\boldsymbol{w} \in \triangle_d$, and always choose the action with the maximum empirical objective value.

Although MC-Empirical appears to be intuitive, its analysis is highly non-trivial due to covariance-based risk in the objective. In this case, a naive universal bound cannot characterize the impact of covariance, and prior gap-dependent analysis (e.g., [11, 25]) cannot be applied to solve our continuous space analysis with gap approximating to zero. Instead, we develop two novel techniques to handle the covariance, including using the actual covariance to analyze the confidence region of the expected rewards, and exploiting the empirical information of the sampling strategy to bound the

risk gap between selected actions and the optimal one. Different from prior works [11, 25], which assume a prior knowledge on covariance or only focus on the independent and positively-related cases, our analysis does not require extra knowledge of covariance and explicitly quantifies the effects of arbitrary covariance structures. The regret performance of `MC-Empirical` is summarized in Theorem 1.

**Theorem 1** (Upper Bound for CMCB-FI). *Consider the continuous mean-covariance bandits with full-information feedback (CMCB-FI). For any $T > 0$, algorithm* `MC-Empirical` *(Algorithm 1) achieves an expected cumulative regret bounded by*

$$O\left(\left(\min\left\{\sqrt{\boldsymbol{w}^{*\top}\Sigma^*\boldsymbol{w}^*} + \rho^{-\frac{1}{2}}\sqrt{\theta^*_{\max} - \theta^*_{\min}},\ \sqrt{\Sigma^*_{\max}}\right\} + \rho\right)\ln T\sqrt{T}\right), \qquad (2)$$

*where* $\theta^*_{\max} = \max_{i\in[d]}\theta^*_i$, $\theta^*_{\min} = \min_{i\in[d]}\theta^*_i$ *and* $\Sigma^*_{\max} = \max_{i\in[d]}\Sigma^*_{ii}$.

*Proof sketch.* Let $D_t$ be the diagonal matrix which takes value $t$ at each diagonal entry. We first build confidence intervals for the expected rewards of actions and the covariance as $|\boldsymbol{w}^\top\boldsymbol{\theta}^* - \boldsymbol{w}^\top\hat{\boldsymbol{\theta}}_{t-1}| \le p_t(\boldsymbol{w}) \triangleq c_1\sqrt{\ln t}\sqrt{\boldsymbol{w}^\top D_{t-1}^{-1}(\lambda\Lambda_{\Sigma^*}D_{t-1} + \sum_{s=1}^{t-1}\Sigma^*)D_{t-1}^{-1}\boldsymbol{w}}$ and $|\Sigma^*_{ij} - \hat{\Sigma}_{ij,t-1}| \le q_t \triangleq c_2\frac{\ln t}{\sqrt{t-1}}$. Here $\lambda = \frac{\boldsymbol{w}^{*\top}\Sigma^*\boldsymbol{w}^*}{\Sigma^*_{\max}}$ and $c_1, c_2$ are positive constants. Then, we obtain the confidence interval of $f(\boldsymbol{w})$ as $|\hat{f}_{t-1}(\boldsymbol{w}) - f(\boldsymbol{w})| \le r_t(\boldsymbol{w}) \triangleq p_t(\boldsymbol{w}) + \rho\boldsymbol{w}^\top Q_t\boldsymbol{w}$, where $Q_t$ is a matrix with all entries equal to $q_t$. Since algorithm `MC-Empirical` always plays the empirical best action, we have $f(\boldsymbol{w}^*) - f(\boldsymbol{w}_t) \le \hat{f}_{t-1}(\boldsymbol{w}^*) + r_t(\boldsymbol{w}^*) - f(\boldsymbol{w}_t) \le \hat{f}_{t-1}(\boldsymbol{w}_t) + r_t(\boldsymbol{w}^*) - f(\boldsymbol{w}_t) \le r_t(\boldsymbol{w}^*) + r_t(\boldsymbol{w}_t)$. Plugging the definitions of $f(\boldsymbol{w})$ and $r_t(\boldsymbol{w})$, we have

$$-\Delta_{\theta^*} + \rho\left(\boldsymbol{w}_t^\top\Sigma^*\boldsymbol{w}_t - \boldsymbol{w}^{*\top}\Sigma^*\boldsymbol{w}^*\right) \le f(\boldsymbol{w}^*) - f(\boldsymbol{w}_t) \overset{(a)}{\le} \frac{c_3\ln t\left(\sqrt{\boldsymbol{w}^{*\top}\Sigma^*\boldsymbol{w}^*} + \sqrt{\boldsymbol{w}_t^\top\Sigma^*\boldsymbol{w}_t} + \rho\right)}{\sqrt{t-1}}, \quad (3)$$

where $\Delta_{\theta^*} = \theta^*_{\max} - \theta^*_{\min}$ and $c_3$ is a positive constant. Since our goal is to bound the regret $f(\boldsymbol{w}^*) - f(\boldsymbol{w}_t)$ and in inequality (a) only the $\sqrt{\boldsymbol{w}_t^\top\Sigma^*\boldsymbol{w}_t}$ term is a variable, the challenge falls on bounding $\boldsymbol{w}_t^\top\Sigma^*\boldsymbol{w}_t$. Note that the left-hand-side of Eq. (3) is linear with respect to $\boldsymbol{w}_t^\top\Sigma^*\boldsymbol{w}_t$ and the right-hand-side only contains $\sqrt{\boldsymbol{w}_t^\top\Sigma^*\boldsymbol{w}_t}$. Then, using the property of sampling strategy on $\boldsymbol{w}_t$, i.e., Eq. (3), again, after some algebraic analysis, we obtain $\boldsymbol{w}_t^\top\Sigma^*\boldsymbol{w}_t \le c_4(\boldsymbol{w}^{*\top}\Sigma^*\boldsymbol{w}^* + \frac{1}{\rho}\Delta_{\theta^*} + \frac{1}{\rho}\sqrt{\frac{\ln t}{t-1}}\sqrt{\boldsymbol{w}^{*\top}\Sigma^*\boldsymbol{w}^*} + \frac{\ln t}{\sqrt{t-1}} + \frac{\ln t}{\rho^2(t-1)})$ for some constant $c_4$. Plugging it into inequality (a) and doing a summation over $t$, we obtain the theorem. $\qquad\square$

**Remark 1.** As we will show in Section 4.2, this $O(\sqrt{T})$ regret matches the lower bound up to a logarithmic factor. Moreover, Theorem 1 fully characterizes how an *arbitrary* covariance structure impacts the regret bound. To see this, note that in Eq. (2), under the **min** operation, the first $\sqrt{\boldsymbol{w}^{*\top}\Sigma^*\boldsymbol{w}^*}$-related term dominates under reasonable $\rho$, and shrinks from positive to negative correlation, which implies that the more the base arms are negatively (positively) correlate, the lower (higher) regret the learner suffers. The intuition behind is that the negative (positive) correlation diversifies (intensifies) the risk of estimation error and narrows (enlarges) the confidence region for the expected reward of an action, which leads to a reduction (an increase) of regret.

Also note that when $\rho = 0$, the CMCB-FI problem reduces to a $d$-armed bandit problem with full-information feedback, and Eq. (2) becomes $\tilde{O}(\sqrt{\Sigma^*_{\max}T})$. For this degenerated case, the optimal gap-dependent regret is $O(\frac{\Sigma^*_{\max}}{\Delta})$ for constant gap $\Delta > 0$. By setting $\Delta = \sqrt{\Sigma^*_{\max}/T}$ at this gap-dependent result, one obtains the optimal gap-independent regret $O(\sqrt{\Sigma^*_{\max}T})$. Hence, when $\rho = 0$, Eq. (2) still offers a tight gap-independent regret bound.

## 4.2 Lower Bound for CMCB-FI

Now we provide a regret lower bound for CMCB-FI, which demonstrates that the $O(\sqrt{T})$ regret of `MC-Empirical` is in fact optimal (up to a logarithmic factor).

Since CMCB-FI considers full-information feedback and continuous decision space where the reward gap $\Delta$ (between the optimal action and the nearest optimal action) approximates to zero, existing lower bound analysis for linear [8, 9] or discrete [19, 11, 25] bandit problems cannot be applied to this problem.

---

**Algorithm 2** MC-UCB

1: **Input:** $\rho > 0$, $c \in (0, \frac{1}{2}]$ and regularization parameter $\lambda > 0$.
2: Initialize: $\forall i \in [d]$, pull $\boldsymbol{e}_i$ that has 1 at the $i$-th entry and 0 elsewhere. $\forall i, j \in [d], i \neq j$, pull $\boldsymbol{e}_{ij}$ that has $\frac{1}{2}$ at the $i$-th and the $j$-th entries, and 0 elsewhere. Update $N_{ij}(d^2)$, $\forall i, j \in [d]$, $\hat{\boldsymbol{\theta}}_{d^2}$ and $\hat{\Sigma}_{d^2}$.
3: **for** $t = d^2 + 1, \ldots$ **do**
4: $\quad \underline{\Sigma}_{t,ij} \leftarrow \hat{\Sigma}_{t-1,ij} - g_{ij}(t)$
5: $\quad \bar{\Sigma}_{t,ij} \leftarrow \hat{\Sigma}_{t-1,ij} + g_{ij}(t)$

6: $\quad \boldsymbol{w}_t \leftarrow \underset{\boldsymbol{w} \in \triangle_d^c}{\arg\max}(\boldsymbol{w}^\top \hat{\boldsymbol{\theta}}_{t-1} + E_t(\boldsymbol{w}) - \rho \boldsymbol{w}^\top \underline{\Sigma}_t \boldsymbol{w})$
7: $\quad$ Pull $\boldsymbol{w}_t$ and observe all $\theta_{t,i}$ s.t. $w_{t,i} > 0$
8: $\quad J_{t,ij} \leftarrow \mathbb{I}\{w_{t,i}, w_{t,j} > 0\}$, $\forall i, j \in [d]$
9: $\quad N_{ij}(t) \leftarrow N_{ij}(t-1) + J_{t,ij}$, $\forall i, j \in [d]$
10: $\quad \hat{\theta}_{t,i}^* \leftarrow \frac{\sum_{s=1}^t J_{t,ii}\theta_{s,i}}{N_{ii}(t)}$, $\forall i \in [d]$
11: $\quad \hat{\Sigma}_{t,ij} \leftarrow \frac{\sum_{s=1}^t J_{t,ij}(\theta_{s,i} - \hat{\theta}_{t,i}^*)(\theta_{s,j} - \hat{\theta}_{t,j}^*)}{N_{ij}(t)}$, $\forall i, j \in [d]$
12: **end for**

---

To tackle this challenge, we contribute a new analytical procedure to establish the lower bound for continuous and full-information bandit problems from the Bayesian perspective. The main idea is to construct an instance distribution, where $\boldsymbol{\theta}^*$ is drawn from a well-chosen prior Gaussian distribution. After $t$ pulls the posterior of $\boldsymbol{\theta}^*$ is still Gaussian with a mean vector $\boldsymbol{u}_t$ related to sample outcomes. Since the hindsight strategy simply selects the action which maximizes the mean-covariance function with respect to $\boldsymbol{\theta}^*$ while a feasible strategy can only utilize the sample information ($\boldsymbol{u}_t$), we show that any algorithm must suffer $\Omega(\sqrt{T})$ regret due to the gap between random $\boldsymbol{\theta}^*$ and its mean $\boldsymbol{u}_t$. Theorem 2 below formally states this lower bound.

**Theorem 2** (Lower Bound for CMCB-FI). *There exists an instance distribution of the continuous mean-covariance bandits with full-information feedback problem (CMCB-FI), for which any algorithm has an expected cumulative regret bounded by $\Omega(\sqrt{T})$.*

**Remark 2.** This parameter-free lower bound demonstrates that the regret upper bound (Theorem 1) of MC-Empirical is optimal (within a logarithmic factor), since under the constructed instance distribution, Theorem 1 also implies a matching $\tilde{O}(\sqrt{T})$ parameter-free result, i.e., when $\rho = 1/\sqrt{T}$, Eq. (2) becomes $\tilde{O}((\sqrt{\Sigma_{\max}^*} + 1/\sqrt{T})\sqrt{T}) = \tilde{O}(\sqrt{T})$. Unlike discrete bandit problems [19, 11, 25] where the optimal regret is usually $\frac{\log T}{\Delta}$ for constant gap $\Delta > 0$, CMCB-FI has continuous decision space with gap $\Delta \to 0$ and a polylogarithmic regret is not achievable in general. In such continuous bandit literature [18, 8, 9], the parameter ($\boldsymbol{\theta}^*$, $\Sigma^*$ and $\rho$) dependent lower bound is an open problem.

## 5 CMCB with Semi-Bandit Feedback (CMCB-SB)

In many practical tasks, the learner may not be able to simultaneously select (place positive weights on) all options and observe full information. Instead, the weight of each option is usually lower bounded and cannot be arbitrarily small. As a result, the learner only selects a subset of options and obtains their feedback, e.g., company investments [12] on multiple business.

Motivated by such tasks, in this section we consider the CMCB problem with semi-bandit feedback (CMCB-SB), where the decision space is a restricted probability simplex $\triangle_d^c = \{\boldsymbol{w} \in \mathbb{R}^d : w_i = 0 \text{ or } c \leq w_i \leq 1, \forall i \in [d] \text{ and } \sum_i w_i = 1\}$ for some constant $0 < c \leq \frac{1}{2}$.[2] In this scenario, at timestep $t$, the learner selects $\boldsymbol{w}_t \in \triangle_d^c$ and only observes the rewards $\{\theta_{t,i} : w_i \geq c\}$ from the base arms that are placed positive weights on. Below, we propose the Mean-Covariance Upper Confidence Bound algorithm (MC-UCB) for CMCB-SB, and provide a regret lower bound, which shows that MC-UCB achieves the optimal performance with respect to $T$.

### 5.1 Algorithm for CMCB-SB

Algorithm MC-UCB for CMCB-SB is described in Algorithm 2. The main idea is to use the optimistic covariance to construct a confidence region for the expected reward of an action and calculate an upper confidence bound of the mean-covariance function, and then select the action with the maximum optimistic mean-covariance value.

---

[2]When $c > \frac{1}{2}$, the learner can only place all weight on one option, and the problem trivially reduces to the mean-variance bandit setting [26, 34]. In this case, our Theorem 3 still provides a tight gap-independent bound.

In Algorithm 2, $N_{ij}(t)$ denotes the number of times $w_{s,i}, w_{s,j} > 0$ occurs among timestep $s \in [t]$. $J_{t,ij}$ is an indicator variable that takes value 1 if $w_{t,i}, w_{t,j} > 0$ and 0 otherwise. $D_t$ is a diagonal matrix such that $D_{t,ii} = N_{ii}(t)$. In Line 2, we update the number of observations by $N_{ii}(d^2) \leftarrow 2d-1$ for all $i \in [d]$ and $N_{ij}(d^2) \leftarrow 2$ for all $i, j \in [d], i \neq j$ (due to the initialized $d^2$ pulls), and calculate the empirical mean $\hat{\theta}_{d^2}^*$ and empirical covariance $\hat{\Sigma}_{d^2}^*$ using the equations in Lines 10,11.

For any $t > 1$ and $i, j \in [d]$, we define the confidence radius of covariance $\Sigma_{ij}^*$ as $g_{ij}(t) \triangleq 16\left(\frac{3\ln t}{N_{ij}(t-1)} \vee \sqrt{\frac{3\ln t}{N_{ij}(t-1)}}\right) + \sqrt{\frac{48\ln^2 t}{N_{ij}(t-1)N_{ii}(t-1)}} + \sqrt{\frac{36\ln^2 t}{N_{ij}(t-1)N_j(t-1)}}$, and the confidence region for the expected reward $\boldsymbol{w}^\top \boldsymbol{\theta}^*$ of action $\boldsymbol{w}$ as

$$E_t(\boldsymbol{w}) \triangleq \sqrt{2\beta(\delta_t)\left(\boldsymbol{w}^\top D_{t-1}^{-1}\left(\lambda \Lambda_{\bar{\Sigma}_t} D_{t-1} + \sum_{s=1}^{t-1} \bar{\Sigma}_{s,\boldsymbol{w}_s}\right) D_{t-1}^{-1} \boldsymbol{w}\right)},$$

where $\lambda > 0$ is the regularization parameter, $\beta(\delta_t) = \ln(\frac{1}{\delta_t}) + d\ln\ln t + \frac{d}{2}\ln(1+\frac{e}{\lambda})$ is the confidence term and $\delta_t = \frac{1}{t\ln^2 t}$ is the confidence parameter. At each timestep $t$, algorithm MC-UCB calculates the upper confidence bound of $f(\boldsymbol{w})$ using $g_{ij}(t)$ and $E_t(\boldsymbol{w})$, and selects the action $\boldsymbol{w}_t$ that maximizes this upper confidence bound. Then, the learner observes rewards $\theta_{t,i}$ with $w_{t,i} > 0$ and update the statistical information according to the feedback.

In regret analysis, unlike [11] which uses a universal upper bound to analyze confidence intervals, we incorporate the estimated covariance into the confidence region for the expected reward of an action, which enables us to derive tighter regret bound and explictly quantify the impact of the covariance structure on algorithm performance. We also contribute a new technique for handling the challenge raised by having different numbers of observations among base arms, in order to obtain an optimal $O(\sqrt{T})$ regret (here prior gap-dependent analysis [11, 25] still cannot be applied to solve this continuous problem). Theorem 3 gives the regret upper bound of algorithm MC-UCB.

**Theorem 3** (Upper Bound for CMCB-SB). *Consider the continuous mean-covariance bandits with semi-bandit feedback problem (CMCB-SB). Then, for any $T > 0$, algorithm MC-UCB (Algorithm 2) with regularization parameter $\lambda > 0$ has an expected cumulative regret bounded by*

$$O\left(\sqrt{L(\lambda)(\|\Sigma^*\|_+ + d^2)d\ln^2 T \cdot T} + \rho d\ln T\sqrt{T}\right),$$

*where $L(\lambda) = (\lambda + 1)(\ln(1 + \lambda^{-1}) + 1)$ and $\|\Sigma^*\|_+ = \sum_{i,j\in[d]}(\Sigma_{ij}^* \vee 0)$ for any $i, j \in [d]$.*

**Remark 3.** Theorem 3 captures the effects of covariance structures in CMCB-SB, i.e., positive correlation renders a larger $\|\Sigma^*\|_+$ factor than the negative correlation or independent case, since the covariance influences the rate of estimate concentration for the expected rewards of actions. The regret bound for CMCB-SB has a heavier dependence on $d$ than that for CMCB-FI. This matches the fact that semi-bandit feedback only reveals rewards of the queried dimensions, and provides less information than full-information feedback in terms of observable dimensions.

## 5.2 Lower Bound for CMCB-SB

In this subsection, we establish a lower bound for CMCB-SB, and show that algorithm MC-UCB achieves the optimal regret with respect to $T$ up to logarithmic factors.

The insight of the lower bound analysis is to construct two instances with a gap in the expected reward vector $\boldsymbol{\theta}^*$, where the optimal actions under these two instances place positive weights on different base arms. Then, when the gap is set to $\sqrt{\ln T/T}$, any algorithm must suffer $\Omega\left(\sqrt{T\ln T}\right)$ regret for differentiating these two instances. Theorem 4 summarizes the lower bound for CMCB-SB.

**Theorem 4** (Lower Bound for CMCB-SB). *There exists an instance distribution of the continuous mean-covariance bandits with semi-bandit feedback (CMCB-SB) problem, for which any algorithm has an expected cumulative regret bounded by $\Omega\left(\sqrt{cdT}\right)$.*

**Remark 4.** Theorem 4 demonstrates that the regret upper bound (Theorem 3) of MC-UCB is optimal with respect to $T$ (up to logarithmic factors). Similar to CMCB-FI, CMCB-SB considers continuous

---

**Algorithm 3** MC-ETE

| | |
|---|---|
| 1: **Input:** $\rho > 0$, $\tilde{d} = \frac{d(d+1)}{2}$ and design action set $\pi = \{\boldsymbol{v}_1, \ldots, \boldsymbol{v}_{\tilde{d}}\}$. | 12: $\quad\quad \boldsymbol{y}_{N_\pi(t)} \leftarrow (y_{N_\pi(t),1}, \ldots, y_{N_\pi(t),\tilde{d}})^\top$ |
| 2: Initialize: $N_\pi(0) \leftarrow 0$. $t \leftarrow 1$. | 13: $\quad\quad \hat{\boldsymbol{y}}_t \leftarrow \frac{\sum_{s=1}^{N_\pi(t)} \boldsymbol{y}_s}{N_\pi(t)}$ |
| 3: **Repeat** lines 4-22: | 14: $\quad\quad \hat{z}_{t,k} = \frac{\sum_{s=1}^{N_\pi(t)} (y_{s,k} - \hat{y}_{t,k})^2}{N_\pi(t)}, \forall k \in [\tilde{d}]$ |
| 4: **if** $N_\pi(t-1) > t^{\frac{2}{3}}/d$ **then** | 15: $\quad\quad \hat{\boldsymbol{z}}_t \leftarrow (\hat{z}_{t,1}, \ldots, \hat{z}_{t,\tilde{d}})^\top$ |
| 5: $\quad \boldsymbol{w}_t = \underset{\boldsymbol{w} \in \triangle_d}{\arg\max} (\boldsymbol{w}^\top \hat{\boldsymbol{\theta}}_{t-1}^* - \rho \boldsymbol{w}^\top \hat{\Sigma}_{t-1} \boldsymbol{w})$ | 16: $\quad\quad \hat{\boldsymbol{\theta}}_t \leftarrow B_\pi^+ \hat{\boldsymbol{y}}_t$ |
| 6: $\quad t \leftarrow t+1$ | 17: $\quad\quad \hat{\boldsymbol{\sigma}}_t \leftarrow C_\pi^+ \hat{\boldsymbol{z}}_t$ |
| 7: **else** | 18: $\quad\quad$ Reshape $\hat{\boldsymbol{\sigma}}_t$ to $d \times d$ matrix $\hat{\Sigma}_t$ |
| 8: $\quad N_\pi(t) \leftarrow N_\pi(t-1) + 1$ | 19: $\quad$ **end if** |
| 9: $\quad$ **for** $k = 1, \ldots, \tilde{d}$ **do** | 20: $\quad t \leftarrow t+1$ |
| 10: $\quad\quad$ Pull $\boldsymbol{v}_k$ and observe $y_{N_\pi(t),k}$ | 21: $\quad$ **end for** |
| 11: $\quad\quad$ **if** $k = \tilde{d}$ **then** | 22: **end if** |

---

decision space with $\Delta \to 0$, and thus the lower bound differs from those gap-dependent results $\frac{\log T}{\Delta}$ in discrete bandit problems [19, 11, 25]. Our lower bound shows that for CMCB-SB, no improvement upon $O(\sqrt{T})$ regret is possible in general.

# 6 CMCB with Full-Bandit Feedback (CMCB-FB)

In this section, we further study the CMCB problem with full-bandit feedback (CMCB-FB), where at timestep $t$, the learner selects $\boldsymbol{w}_t \in \triangle_d$ and only observes the weighted sum of random rewards, i.e., $y_t = \boldsymbol{w}_t^\top \boldsymbol{\theta}_t$. This setting models many real-world decision making tasks, where the learner can only attain an aggregate feedback from the chosen options, such as clinical trials [31].

## 6.1 Algorithm for CMCB-FB

We propose the Mean-Covariance Exploration-Then-Exploitation algorithm (MC-ETE) for CMCB-FB in Algorithm 3. Specifically, we first choose a design action set $\pi = \{\boldsymbol{v}_1, \ldots, \boldsymbol{v}_{\tilde{d}}\}$ which contains $\tilde{d} = d(d+1)/2$ actions and satisfies that $B_\pi = (\boldsymbol{v}_1^\top; \ldots; \boldsymbol{v}_{\tilde{d}}^\top)$ and $C_\pi = (v_{1,1}^2, \ldots, v_{1,d}^2, 2v_{1,1}v_{1,2}, \ldots, 2v_{1,d-1}v_{1,d}; \ldots; v_{\tilde{d},1}^2, \ldots, v_{\tilde{d},d}^2, 2v_{\tilde{d},1}v_{\tilde{d},2}, \ldots, 2v_{\tilde{d},d-1}v_{\tilde{d},d})$ are of full column rank. We also denote their Moore-Penrose inverses by $B_\pi^+$ and $C_\pi^+$, and it holds that $B_\pi^+ B_\pi = I^{d \times d}$ and $C_\pi^+ C_\pi = I^{\tilde{d} \times \tilde{d}}$. There exist more than one feasible $\pi$, and for simplicity and good performance we choose $\boldsymbol{v}_1, \ldots, \boldsymbol{v}_d$ as standard basis vectors in $\mathbb{R}^d$ and $\{\boldsymbol{v}_{d+1}, \ldots, \boldsymbol{v}_{\tilde{d}}\}$ as the set of all $\binom{d}{2}$ vectors where each vector has two entries equal to $\frac{1}{2}$ and others equal to $0$.

In an exploration round (Lines 8-21), we pull the designed actions in $\pi$ and maintain their empirical rewards and variances. Through linear transformation by $B_\pi^+$ and $C_\pi^+$, we obtain the estimators of the expected rewards and covariance of base arms (Lines 16-17). When the estimation confidence is high enough, we exploit the attained information to select the empirical best action (Lines 5). Theorem 5 presents the regret guarantee of MC-ETE.

**Theorem 5** (Upper Bound for CMCB-FB). *Consider the continuous mean-covariance bandits with full-bandit feedback problem (CMCB-FB). Then, for any $T > 0$, algorithm MC-ETE (Algorithm 3) achieves an expected cumulative regret bounded by*

$$O \left( Z(\rho, \pi) \sqrt{d(\ln T + d^2)} \cdot T^{\frac{2}{3}} + d\Delta_{\max} \cdot T^{\frac{2}{3}} \right),$$

*where $Z(\rho, \pi) = \max_{\boldsymbol{w} \in \triangle_d}(\sqrt{\boldsymbol{w}^\top B_\pi^+ \Sigma_\pi^* (B_\pi^+)^\top \boldsymbol{w}} + \rho \|C_\pi^+\|)$, $\Sigma_\pi^* = \mathrm{diag}(\boldsymbol{v}_1^\top \Sigma^* \boldsymbol{v}_1, \ldots, \boldsymbol{v}_{\tilde{d}}^\top \Sigma^* \boldsymbol{v}_{\tilde{d}})$, $\|C_\pi^+\| = \max_{i \in [\tilde{d}]} \sum_{j \in [\tilde{d}]} |C_{\pi,ij}^+|$ and $\Delta_{\max} = f(\boldsymbol{w}^*) - \min_{\boldsymbol{w} \in \triangle_d} f(\boldsymbol{w})$.*

**Remark 5.** The choice of $\pi$ will affect the regret factor $\Sigma_\pi^*$ contained in $Z(\rho, \pi)$. Under our construction, $\Sigma_\pi^*$ can be regarded as a uniform representation of covariance $\Sigma^*$, and thus our regret bound demonstrates how the learning performance is influenced by the covariance structure, i.e., negative (positive) correlation shrinks (enlarges) the factor and leads to a lower (higher) regret.

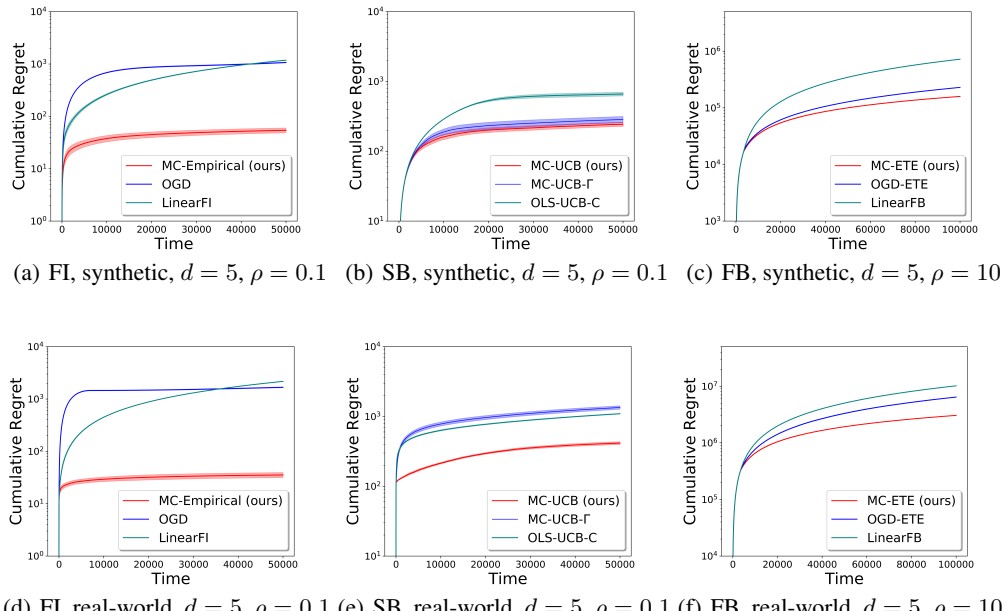

(a) FI, synthetic, $d = 5$, $\rho = 0.1$ (b) SB, synthetic, $d = 5$, $\rho = 0.1$ (c) FB, synthetic, $d = 5$, $\rho = 10$

(d) FI, real-world, $d = 5$, $\rho = 0.1$ (e) SB, real-world, $d = 5$, $\rho = 0.1$ (f) FB, real-world, $d = 5$, $\rho = 10$

Figure 1: Experiments for CMCB-FI, CMCB-SB and CMCB-FB on the synthetic and real-world datasets.

**Discussion on the ETE strategy.** In contrast to common ETE-type algorithms, MC-ETE requires novel analytical techniques in handling the transformed estimate concentration while preserving the covariance information in regret bounds. In analysis, we build a novel concentration using key matrices $B_\pi^+$ and $C_\pi^+$ to adapt to the actual covariance structure, and construct a super-martingale which takes the aggregate noise in an exploration round as analytical basis to prove the concentration. These techniques allow us to capture the correlations in the results, and are new compared to both the former FI/SB settings and covariance-related bandit literature [33, 11, 25].

In fact, under the full-bandit feedback, it is highly challenging to estimate the covariance without using a fixed exploration (i.e., ETE) strategy. Note that even for its simplified offline version, where one uses given (non-fixed) full-bandit data to estimate the covariance, there is *no available solution* in the statistics literature to our best knowledge. Hence, for such online tasks with severely limited feedback, ETE is the most viable strategy currently available, as used in many partial observation works [21, 6, 7]. We remark that our contribution in this setting focuses on designing a practical solution and deriving regret guarantees which explicitly characterize the correlation impacts. The lower bound for CMCB-FB remains open, which we leave for future work.

## 7 Experiments

In this section, we present experimental results for our algorithms on both synthetic and real-world [20] datasets. For the synthetic dataset, we set $\boldsymbol{\theta}^* = [0.2, 0.3, 0.2, 0.2, 0.2]^\top$, and $\Sigma^*$ has all diagonal entries equal to 1 and all off-diagonal entries equal to $-0.05$. For the real-world dataset, we use an open dataset *US Funds from Yahoo Finance* on Kaggle [20], which provides financial data of 1680 ETF funds in 2010-2017. We select five funds and generate a stochastic distribution ($\boldsymbol{\theta}^*$ and $\Sigma^*$) from the data of returns (since we study a stochastic bandit problem). For both datasets, we set $d = 5$ and $\rho \in \{0.1, 10\}$. The random reward $\boldsymbol{\theta}_t$ is drawn i.i.d. from Gaussian distribution $\mathcal{N}(\boldsymbol{\theta}^*, \Sigma^*)$. We perform 50 independent runs for each algorithm and show the average regret and $95\%$ confidence interval across runs,[3] with logarithmic y-axis for clarity of magnitude comparison.

---

[3]In some cases, since algorithms are doing similar procedures (e.g., in Figures 1(c),1(f), the algorithms are exploring the designed actions) and have low performance variance, the confidence intervals are narrow and indistinguishable.

**(CMCB-FI)** We compare our algorithm `MC-Empirical` with two algorithms `OGD` [15] and `LinearFI`. `OGD` (Online Gradient Descent) [15] is designed for general online convex optimization with also a $O(\sqrt{T})$ regret guarantee, but its result cannot capture the covariance impacts as ours. `LinearFI` is a linear adaption of `MC-Empirical` that only aims to maximize the expected rewards. Figures 1(a),1(d) show that our `MC-Empirical` enjoys multiple orders of magnitude reduction in regret compared to the benchmarks, since it efficiently exploits the empirical observations to select actions and well handles the covariance-based risk. In particular, the performance superiority of `MC-Empirical` over `OGD` demonstrates that our sample strategy sufficiently utilize the observed information than conventional gradient descent based policy.

**(CMCB-SB)** For CMCB-SB, we compare `MC-UCB` with two adaptions of `OLS-UCB` [11] (state-of-the-art for combinatorial bandits with covariance), named `MC-UCB-Γ` and `OLS-UCB-C`. `MC-UCB-Γ` uses the confidence region with a universal covariance upper bound $\Gamma$, instead of the adapting one used in our `MC-UCB`. `OLS-UCB-C` directly adapts `OLS-UCB` [11] to the continuous decision space and only considers maximizing the expected rewards in its objective. As shown in Figures 1(b),1(e), `MC-UCB` achieves the lowest regret since it utilizes the covariance information to accelerate the estimate concentration. Due to lack of a covariance-adapting confidence interval, `MC-UCB-Γ` shows an inferior regret performance than `MC-UCB`, and `OLS-UCB-C` suffers the highest regret due to its ignorance of risk.

**(CMCB-FB)** We compare `MC-ETE` with two baselines, `OGD-ETE`, which adopts `OGD` [15] during the exploitation phase, and `LinearFB`, which only investigates the expected reward maximization. From Figures 1(c),1(f), one can see that, `MC-ETE` achieves the best regret performance due to its effective estimation of the covariance-based risk and efficiency in exploitation. Due to the inefficiency of gradient descent based policy in utilizing information, `OGD-ETE` has a higher regret than `MC-ETE`, whereas `LinearFB` shows the worst performance owing to the unawareness of the risk.

# 8    Conclusion and Future Work

In this paper, we propose a novel continuous mean-covariance bandit (CMCB) model, which investigates the reward-risk trade-off measured by option correlation. Under this model, we consider three feedback settings, i.e., full-information, semi-bandit and full-bandit feedback, to formulate different real-world reward observation scenarios. We propose novel algorithms for CMCB to achieve the optimal regrets (within logarithmic factors), and provide lower bounds for the problems to demonstrate our optimality. We also present empirical evaluations to show the superior performance of our algorithms. To our best knowledge, this is the first work to fully characterize the impacts of arbitrary covariance structures on learning performance for risk-aware bandits. There are several interesting directions for future work. For example, how to design an adaptive algorithm for CMCB-FB is a challenging open problem, and the lower bound for CMCB-FB is also worth further investigation.

## Acknowledgments and Disclosure of Funding

The work of Yihan Du and Longbo Huang is supported in part by the Technology and Innovation Major Project of the Ministry of Science and Technology of China under Grant 2020AAA0108400 and 2020AAA0108403.

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
