# Appendix

## A   Technical Lemmas

In this section, we introduce two technical lemmas which will be used in our analysis.

Lemmas 1 and 2 give the concentration guarantees of algorithm `MC-UCB` for CMCB-SB, which sets up a foundation for the concentration guarantees in CMCB-FI. For ease of notation, we use $\Sigma$ for a shorthand of the covariance matrix $\Sigma^*$ in Appendix.

**Lemma 1** (Concentration of Covariance for CMCB-SB). *Consider the CMCB-SB problem and algorithm* `MC-UCB` *(Algorithm 2). Define the event*

$$
\mathcal{G}_t \triangleq \left\{ |\Sigma_{ij} - \hat{\Sigma}_{ij,t-1}| \leq 16 \left( \frac{3\ln t}{N_{ij}(t-1)} \vee \sqrt{\frac{3\ln t}{N_{ij}(t-1)}} \right) \right.
$$
$$
\left. + \sqrt{\frac{61\ln^2 t}{N_{ij}(t-1)N_i(t-1)}} + \sqrt{\frac{36\ln^2 t}{N_{ij}(t-1)N_j(t-1)}}, \forall i,j \in [d] \right\}
$$

*For any $t \geq 2$, we have*

$$
Pr[\mathcal{G}_t] \geq 1 - \frac{10d^2}{t^2}.
$$

*Proof.* According to Proposition 2 in [27], we have that for any $t \geq 2$ and $i,j \in [d]$,

$$
\Pr \left[ |\Sigma_{ij} - \hat{\Sigma}_{ij,t-1}| \leq 16 \left( \frac{3\ln t}{N_{ij}(t-1)} \vee \sqrt{\frac{3\ln t}{N_{ij}(t-1)}} \right) \right.
$$
$$
\left. + \sqrt{\frac{61\ln^2 t}{N_{ij}(t-1)N_i(t-1)}} + \sqrt{\frac{36\ln^2 t}{N_{ij}(t-1)N_j(t-1)}} \right] \leq 1 - \frac{10}{t^2}.
$$

Using a union bound on $i,j \in [d]$, we obtain Lemma 1. $\square$

**Lemma 2** (Concentration of Means for CMCB-SB). *Consider the CMCB-SB problem and algorithm* `MC-UCB` *(Algorithm 2). Let $0 < \lambda < 1$, and define $\delta_t = \frac{1}{t\ln^2 t}$ and $\beta(\delta_t) = \ln(1/\delta_t) + d\ln\ln t + \frac{d}{2}\ln(1 + e/\lambda)$ for $t \geq 2$. Then, for any $t \geq 2$ and $\boldsymbol{w} \in \triangle_d^c$, with probability at least $1 - \delta_t$, we have*

$$
\left| \boldsymbol{w}^\top \boldsymbol{\theta}^* - \boldsymbol{w}^\top \hat{\boldsymbol{\theta}}_{t-1} \right| \leq \sqrt{2\beta(\delta_t)} \sqrt{\boldsymbol{w}^\top D_{t-1}^{-1} \left( \lambda \Lambda_\Sigma D_{t-1} + \sum_{s=1}^{t-1} \Sigma_{\boldsymbol{w}_s} \right) D_{t-1}^{-1} \boldsymbol{w}}.
$$

*Further define $E_t(\boldsymbol{w}) = \sqrt{2\beta(\delta_t)}\sqrt{\boldsymbol{w}^\top D_{t-1}^{-1}(\lambda \Lambda_{\bar{\Sigma}_t} D_{t-1} + \sum_{s=1}^{t-1} \bar{\Sigma}_{s,\boldsymbol{w}_s}) D_{t-1}^{-1} \boldsymbol{w}}$. Then, for any $t \geq 2$ and $\boldsymbol{w} \in \triangle_d^c$, the event $\mathcal{H}_t \triangleq \{|\boldsymbol{w}^\top \boldsymbol{\theta}^* - \boldsymbol{w}^\top \hat{\boldsymbol{\theta}}_{t-1}| \leq E_t(\boldsymbol{w})\}$ satisfies $\Pr[\mathcal{H}_t \mid \mathcal{G}_t] \geq 1 - \delta_t$.*

*Proof.* The proof of Lemma 2 follows the analysis procedure in [13]. Specifically, assuming that event $\mathcal{G}_t$ occurs, we have $\bar{\Sigma}_{t,ij} \geq \Sigma_{ij}$ for any $i,j \in [d]$ and

$$
E_t(\boldsymbol{w}) \geq \sqrt{2\beta(\delta_t)} \sqrt{\boldsymbol{w}^\top D_{t-1}^{-1} \left( \lambda \Lambda_\Sigma D_{t-1} + \sum_{s=1}^{t-1} \Sigma_{\boldsymbol{w}_s} \right) D_{t-1}^{-1} \boldsymbol{w}}.
$$

Hence, to prove Lemma 2, it suffices to prove that

$$\Pr\left[\left|\boldsymbol{w}^\top\boldsymbol{\theta}^* - \boldsymbol{w}^\top\hat{\boldsymbol{\theta}}_{t-1}\right| > \sqrt{2\beta(\delta_t)}\sqrt{\boldsymbol{w}^\top D_{t-1}^{-1}\left(\lambda\Lambda_\Sigma D_{t-1} + \sum_{s=1}^{t-1}\Sigma_{\boldsymbol{w}_s}\right)D_{t-1}^{-1}\boldsymbol{w}}\right] \le \delta_t. \tag{4}$$

Recall that $N_i(t) = \sum_{s=1}^t \mathbb{I}\{w_{s,i} \ge c\}$ and $D_t$ is a diagonal matrix such that $D_{t,ii} = N_i(t)$ for any $t > 0$. For any $\boldsymbol{w} \in \triangle_d^c$, let $I_{\boldsymbol{w}}$ denote the diagonal matrix such that $I_{ii} = 1$ for any $w_i \ge c$ and $I_{jj} = 0$ for any $w_j = 0$, and let $\Sigma_{\boldsymbol{w}} = I_{\boldsymbol{w}}\Sigma I_{\boldsymbol{w}}$. Let $\varepsilon_t$ be the vector such that $\boldsymbol{\eta}_t = \Sigma^{\frac{1}{2}}\varepsilon_t$ for any $t > 0$. Then, we have for any $\boldsymbol{w} \in \triangle_d^c$ that

$$\left|\boldsymbol{w}^\top\left(\boldsymbol{\theta}^* - \hat{\boldsymbol{\theta}}_{t-1}\right)\right| = \left|-\boldsymbol{w}^\top D_{t-1}^{-1}\sum_{s=1}^{t-1}I_{\boldsymbol{w}_s}\Sigma^{\frac{1}{2}}\varepsilon_s\right|$$

$$= \left|-\boldsymbol{w}^\top D_{t-1}^{-1}\left(D + \sum_{s=1}^{t-1}\Sigma_{\boldsymbol{w}_s}\right)^{\frac{1}{2}}\left(D + \sum_{s=1}^{t-1}\Sigma_{\boldsymbol{w}_s}\right)^{-\frac{1}{2}}\sum_{s=1}^{t-1}I_{\boldsymbol{w}_s}\Sigma^{\frac{1}{2}}\varepsilon_s\right|$$

$$\le \sqrt{\boldsymbol{w}^\top D_{t-1}^{-1}\left(D + \sum_{s=1}^{t-1}\Sigma_{\boldsymbol{w}_s}\right)D_{t-1}^{-1}\boldsymbol{w}} \cdot \left\|\sum_{s=1}^{t-1}I_{\boldsymbol{w}_s}\Sigma^{\frac{1}{2}}\varepsilon_s\right\|_{\left(D+\sum_{s=1}^{t-1}\Sigma_{\boldsymbol{w}_s}\right)^{-1}}$$

Let $S_t = \sum_{s=1}^{t-1}I_{\boldsymbol{w}_s}\Sigma^{\frac{1}{2}}\varepsilon_s$, $V_t = \sum_{s=1}^{t-1}\Sigma_{\boldsymbol{w}_s}$ and $I_{D+V_t} = \frac{1}{2}\|S_t\|_{(D+V_t)^{-1}}^2$. We get

$$\left\|\sum_{s=1}^{t-1}I_{\boldsymbol{w}_s}\Sigma^{\frac{1}{2}}\varepsilon_s\right\|_{\left(D+\sum_{s=1}^{t-1}\Sigma_{\boldsymbol{w}_s}\right)^{-1}} = \|S_t\|_{(D+V_t)^{-1}} = \sqrt{2I_{D+V_t}}.$$

Since $D \preceq \lambda\Lambda_\Sigma D_{t-1}$, we have

$$\left|\boldsymbol{w}^\top\left(\boldsymbol{\theta}^* - \hat{\boldsymbol{\theta}}_{t-1}\right)\right| \le \sqrt{\boldsymbol{w}^\top D_{t-1}^{-1}DD_{t-1}^{-1}\boldsymbol{w}^\top + \boldsymbol{w}^\top D_{t-1}^{-1}\left(\sum_{s=1}^{t-1}\Sigma_{\boldsymbol{w}_s}\right)D_{t-1}^{-1}\boldsymbol{w}} \cdot \sqrt{2I_{D+V_t}}$$

$$\le \sqrt{\lambda\boldsymbol{w}^\top D_{t-1}^{-1}\Lambda_\Sigma\boldsymbol{w}^\top + \boldsymbol{w}^\top D_{t-1}^{-1}\left(\sum_{s=1}^{t-1}\Sigma_{\boldsymbol{w}_s}\right)D_{t-1}^{-1}\boldsymbol{w}} \cdot \sqrt{2I_{D+V_t}}$$

$$= \sqrt{\boldsymbol{w}^\top D_{t-1}^{-1}\left(\lambda\Lambda_\Sigma D_{t-1} + \sum_{s=1}^{t-1}\Sigma_{\boldsymbol{w}_s}\right)D_{t-1}^{-1}\boldsymbol{w}} \cdot \sqrt{2I_{D+V_t}}$$

Thus,

$$\Pr\left[\left|\boldsymbol{w}^\top\left(\boldsymbol{\theta}^* - \hat{\boldsymbol{\theta}}_{t-1}\right)\right| > \sqrt{2\beta(\delta_t)}\sqrt{\boldsymbol{w}^\top D_{t-1}^{-1}(\lambda\Lambda_{\bar{\Sigma}_t} D_{t-1} + \sum_{s=1}^{t-1}\bar{\Sigma}_{s,\boldsymbol{w}_s})D_{t-1}^{-1}\boldsymbol{w}}\right]$$

$$\le \Pr\left[\sqrt{\boldsymbol{w}^\top D_{t-1}^{-1}\left(\lambda\Lambda_\Sigma D_{t-1} + \sum_{s=1}^{t-1}\Sigma_{\boldsymbol{w}_s}\right)D_{t-1}^{-1}\boldsymbol{w}} \cdot \sqrt{2I_{D+V_t}}\right.$$

$$\left. > \sqrt{2\beta(\delta_t)}\sqrt{\boldsymbol{w}^\top D_{t-1}^{-1}\left(\lambda\Lambda_\Sigma D_{t-1} + \sum_{s=1}^{t-1}\Sigma_{\boldsymbol{w}_s}\right)D_{t-1}^{-1}\boldsymbol{w}}\right]$$

$$= \Pr\left[ I_{D+V_t} > \beta(\delta_t) \right]$$

Hence, to prove Eq. (4), it suffices to prove

$$\Pr\left[ I_{D+V_t} > \beta(\delta_t) \right] \leq \delta_t. \tag{5}$$

To do so, we introduce some notions. Let $\mathcal{J}_t$ be the $\sigma$-algebra $\sigma(\boldsymbol{w}_1, \varepsilon_1, \ldots, \boldsymbol{w}_{t-1}, \varepsilon_{t-1}, \boldsymbol{w}_t)$. Let $\boldsymbol{u} \in \mathbb{R}^d$ be a multivariate Gaussian random variable with mean $\boldsymbol{0}$ and covariance $D^{-1}$, which is independent of all the other random variables, and use $\varphi(\boldsymbol{u})$ denote its probability density function. Define

$$P_s^{\boldsymbol{u}} = \exp\left( \boldsymbol{u}^\top I_{\boldsymbol{w}_s} \Sigma^{\frac{1}{2}} \varepsilon_s - \frac{1}{2} \boldsymbol{u}^\top \Sigma_{\boldsymbol{w}_s} \boldsymbol{u} \right),$$

$$M_t^{\boldsymbol{u}} \triangleq \exp\left( \boldsymbol{u}^\top S_t - \frac{1}{2} \|\boldsymbol{u}\|_{V_t}^2 \right),$$

and

$$M_t \triangleq \mathbb{E}_{\boldsymbol{u}}[M_t^{\boldsymbol{u}}] = \int_{\mathbb{R}^d} \exp\left( \boldsymbol{u}^\top S_t - \frac{1}{2} \|\boldsymbol{u}\|_{V_t}^2 \right) \varphi(\boldsymbol{u}) du.$$

We have $M_t^{\boldsymbol{u}} = \Pi_{s=1}^{t-1} P_s^{\boldsymbol{u}}$. In the following, we prove $\mathbb{E}[M_t] \leq 1$.

For any $s > 0$, according to the sub-Gaussian property, $\eta_s = \Sigma^{\frac{1}{2}} \varepsilon_s$ satisfies

$$\forall \boldsymbol{v} \in \mathbb{R}^d, \ \mathbb{E}\left[ e^{\boldsymbol{v}^\top \Sigma^{\frac{1}{2}} \varepsilon_s} \right] \leq e^{\frac{1}{2} \boldsymbol{v}^\top \Sigma \boldsymbol{v}},$$

which is equivalent to

$$\forall \boldsymbol{v} \in \mathbb{R}^d, \ \mathbb{E}\left[ e^{\boldsymbol{v}^\top \Sigma^{\frac{1}{2}} \varepsilon_s - \frac{1}{2} \boldsymbol{v}^\top \Sigma \boldsymbol{v}} \right] \leq 1.$$

Thus, we have

$$\mathbb{E}\left[ P_s^{\boldsymbol{u}} | \mathcal{J}_s \right] = \mathbb{E}\left[ \exp\left( \boldsymbol{u}^\top I_{\boldsymbol{w}_s} \Sigma^{\frac{1}{2}} \varepsilon_s - \frac{1}{2} \boldsymbol{u}^\top \Sigma_{\boldsymbol{w}_s} \boldsymbol{u} \right) | \mathcal{J}_s \right] \leq 1.$$

Then, we can obtain

$$\begin{aligned}
\mathbb{E}[M_t^{\boldsymbol{u}} | \mathcal{J}_{t-1}] &= \mathbb{E}\left[ \Pi_{s=1}^{t-1} P_s^{\boldsymbol{u}} | \mathcal{J}_{t-1} \right] \\
&= \left( \Pi_{s=1}^{t-2} P_s^{\boldsymbol{u}} \right) \mathbb{E}\left[ P_{t-1}^{\boldsymbol{u}} | \mathcal{J}_{t-1} \right] \\
&\leq M_{t-1}^{\boldsymbol{u}},
\end{aligned}$$

which implies that $M_t^{\boldsymbol{u}}$ is a super-martingale and $\mathbb{E}[M_t^{\boldsymbol{u}} | \boldsymbol{u}] \leq 1$. Thus,

$$\mathbb{E}[M_t] = \mathbb{E}_{\boldsymbol{u}}[\mathbb{E}[M_t^{\boldsymbol{u}} | \boldsymbol{u}]] \leq 1.$$

According to Lemma 9 in [1], we have

$$M_t \triangleq \int_{\mathbb{R}^d} \exp\left( \boldsymbol{u}^\top S_t - \frac{1}{2} \|\boldsymbol{u}\|_{V_t}^2 \right) \varphi(\boldsymbol{u}) du = \sqrt{\frac{\det D}{\det(D + V_t)}} \exp\left( I_{D+V_t} \right).$$

Thus,

$$\mathbb{E}\left[ \sqrt{\frac{\det D}{\det(D + V_t)}} \exp\left( I_{D+V_t} \right) \right] \leq 1.$$

Now we prove Eq. (5). First, we have

$$\Pr\left[I_{D+V_t} > \beta(\delta_t)\right] = \Pr\left[\sqrt{\frac{\det D}{\det(D+V_t)}}\exp\left(I_{D+V_t}\right) > \sqrt{\frac{\det D}{\det(D+V_t)}}\exp\left(\beta(\delta_t)\right)\right]$$

$$= \Pr\left[M_t > \frac{1}{\sqrt{\det(I + D^{-\frac{1}{2}}V_t D^{-\frac{1}{2}})}}\exp\left(\beta(\delta_t)\right)\right]$$

$$\leq \frac{\mathbb{E}[M_t]\sqrt{\det(I + D^{-\frac{1}{2}}V_t D^{-\frac{1}{2}})}}{\exp\left(\beta(\delta_t)\right)}$$

$$\leq \frac{\sqrt{\det(I + D^{-\frac{1}{2}}V_t D^{-\frac{1}{2}})}}{\exp\left(\beta(\delta_t)\right)} \tag{6}$$

Then, for some constant $\gamma > 0$ and for any $\boldsymbol{a} = (a_1, \ldots, a_d) \in \mathbb{N}^d$, we define the set of timesteps $\mathcal{K}_{\boldsymbol{a}} \subseteq [T]$ such that

$$t \in \mathcal{K}_{\boldsymbol{a}} \Leftrightarrow \forall i \in d,\ (1+\gamma)^{a_i} \leq N_i(t) < (1+\gamma)^{a_i+1}.$$

Define $D_{\boldsymbol{a}}$ a diagonal matrix with $D_{\boldsymbol{a},ii} = (1+\gamma)^{a_i}$.

Suppose $t \in \mathcal{K}_{\boldsymbol{a}}$ for some fixed $\boldsymbol{a}$. We have

$$\frac{1}{1+\gamma}D_t \preceq D_{\boldsymbol{a}} \preceq D_t.$$

Let $D = \lambda\Lambda_\Sigma D_{\boldsymbol{a}} \succeq \frac{\lambda}{1+\gamma}\Lambda_\Sigma D_t$. Then, we have

$$D^{-\frac{1}{2}}V_t D^{-\frac{1}{2}} \preceq \frac{1+\gamma}{\lambda}D_t^{-\frac{1}{2}}\Lambda_\Sigma^{-\frac{1}{2}}V_t\Lambda_\Sigma^{-\frac{1}{2}}D_t^{-\frac{1}{2}},$$

where matrix $D_t^{-\frac{1}{2}}\Lambda_\Sigma^{-\frac{1}{2}}V_t\Lambda_\Sigma^{-\frac{1}{2}}D_t^{-\frac{1}{2}}$ has $d$ ones on the diagonal. Since the determinant of a positive definite matrix is smaller than the product of its diagonal terms, we have

$$\det(I + D^{-\frac{1}{2}}V_t D^{-\frac{1}{2}}) \leq \det(I + \frac{1+\gamma}{\lambda}D_t^{-\frac{1}{2}}\Lambda_\Sigma^{-\frac{1}{2}}V_t\Lambda_\Sigma^{-\frac{1}{2}}D_t^{-\frac{1}{2}})$$

$$\leq \left(1 + \frac{1+\gamma}{\lambda}\right)^d \tag{7}$$

Let $0 < \lambda < 1$ and $\gamma = e - 1$. Using Eqs. (6) and (7), $\beta(\delta_t) = \ln(1/\delta_t) + d\ln\ln t + \frac{d}{2}\ln(1 + e/\lambda) = \ln(t\ln^2 t) + d\ln\ln t + \frac{d}{2}\ln(1 + e/\lambda)$, and a union bound over $\boldsymbol{a}$, we have

$$\Pr\left[I_{D+V_t} > \beta(\delta_t)\right] \leq \sum_{\boldsymbol{a}} \Pr\left[I_{D+V_t} > \beta(\delta_t)|t \in \mathcal{K}_{\boldsymbol{a}}, D = \lambda\Lambda_\Sigma D_{\boldsymbol{a}}\right]$$

$$\leq \sum_{\boldsymbol{a}} \frac{\sqrt{\det(I + D^{-\frac{1}{2}}V_t D^{-\frac{1}{2}})}}{\exp\left(\beta(\delta_t)\right)}$$

$$\leq \left(\frac{\ln t}{\ln(1+\gamma)}\right)^d \cdot \frac{\left(1 + \frac{1+\gamma}{\lambda}\right)^{\frac{d}{2}}}{\exp\left(\ln(t\ln^2 t) + d\ln\ln t + \frac{d}{2}\ln(1 + \frac{e}{\lambda})\right)}$$

$$= (\ln t)^d \cdot \frac{\left(1 + \frac{e}{\lambda}\right)^{\frac{d}{2}}}{t\ln^2 t \cdot (\ln t)^d \cdot \left(1 + \frac{e}{\lambda}\right)^{\frac{d}{2}}}$$

$$= \frac{1}{t \ln^2 t}$$
$$= \delta_t$$

Thus, Eq. (5) holds and we complete the proof of Lemma 2. $\qquad\square$

## B Proof for CMCB-FI

### B.1 Proof of Theorem 1

In order to prove Theorem 1, we first have the following Lemmas 3 and 4, which are adaptions of Lemmas 1 and 2 to CMCB-FI.

**Lemma 3** (Concentration of Covariance for CMCB-FI). *Consider the CMCB-FI problem and algorithm* MC-Empirical *(Algorithm 1). For any $t \geq 2$, the event*

$$\mathcal{E}_t \triangleq \left\{ |\Sigma_{ij} - \hat{\Sigma}_{ij,t-1}| \leq 16 \left( \frac{3 \ln t}{t-1} \vee \sqrt{\frac{3 \ln t}{t-1}} \right) + \left( 6 + 4\sqrt{3} \right) \frac{\ln t}{t-1}, \forall i, j \in [d] \right\}$$

*satisfies*

$$\Pr[\mathcal{E}_t] \geq 1 - \frac{10 d^2}{t^2}$$

*Proof.* In CMCB-FI, we have $N_{ij}(t-1) = t-1$ for any $t \geq 2$ and $i, j \in [d]$. Then, Lemma 3 can be obtained by applying Lemma 1 with $N_{ij}(t-1) = t-1$ for any $i, j \in [d]$. $\qquad\square$

**Lemma 4** (Concentration of Means for CMCB-FI). *Consider the CMCB-FI problem and algorithm* MC-Empirical *(Algorithm 1). Let $0 < \lambda < 1$. Define $\delta_t = \frac{1}{t \ln^2 t}$ and $\beta(\delta_t) = \ln(1/\delta_t) + \ln \ln t + \frac{d}{2} \ln(1 + e/\lambda)$ for $t \geq 2$. Define $E_t(\boldsymbol{w}) = \sqrt{2\beta(\delta_t)} \sqrt{\boldsymbol{w}^\top D_{t-1}^{-1} (\lambda \Lambda_\Sigma D_{t-1} + \sum_{s=1}^{t-1} \Sigma) D_{t-1}^{-1} \boldsymbol{w}}$. Then, for any $t \geq 2$ and $\boldsymbol{w} \in \triangle_d$, the event $\mathcal{F}_t \triangleq \{|\boldsymbol{w}^\top \boldsymbol{\theta}^* - \boldsymbol{w}^\top \hat{\boldsymbol{\theta}}_{t-1}| \leq E_t(\boldsymbol{w})\}$ satisfies $\Pr[\mathcal{F}_t] \geq 1 - \delta_t$.*

*Proof.* In CMCB-FI, $D_t$ is a diagonal matrix such that $D_{t,ii} = N_i(t) = t$. Then, Lemma 4 can be obtained by applying Lemma 2 with $D_t = tI$ and that the union bound on the number of samples only needs to consider one dimension. Specifically, in the proof of Lemma 2, we replace the set of timesteps $\mathcal{K}_a$ with $\mathcal{K}_a \subseteq [T]$ for $a \in \mathbb{N}$, which stands for

$$t \in \mathcal{K}_a \Leftrightarrow (1 + \gamma)^a \leq t < (1 + \gamma)^{a+1}.$$

This completes the proof. $\qquad\square$

Now we are ready to prove Theorem 1.

*Proof.* (Theorem 1) Let $\Delta_t = f(\boldsymbol{w}^*) - f(\boldsymbol{w}_t)$, $g(t) = 16 \left( \frac{3 \ln t}{t-1} \vee \sqrt{\frac{3 \ln t}{t-1}} \right) + \left( 6 + 4\sqrt{3} \right) \frac{\ln t}{t-1}$ denote the confidence radius of covariance $\Sigma_{ij}^*$ for any $i, j \in [d]$, and $G(t)$ be the matrix with all entries equal to $g(t)$. For any $\boldsymbol{w} \in \triangle_d^c$, define $\hat{f}_t(\boldsymbol{w}) = \boldsymbol{w}^\top \hat{\boldsymbol{\theta}}_t - \rho \boldsymbol{w}^\top \hat{\Sigma}_t \boldsymbol{w}$ and $h_t(\boldsymbol{w}) = E_t(\boldsymbol{w}) + \rho \boldsymbol{w}^\top G_t \boldsymbol{w}$.

For any $t \geq 2$, suppose that event $\mathcal{E}_t \cap \mathcal{F}_t$ occurs. Then,

$$|\hat{f}_{t-1}(\boldsymbol{w}) - f(\boldsymbol{w})| \leq h_t(\boldsymbol{w}).$$

Therefore, we have

$$\Delta_t \leq |\hat{f}_{t-1}(\boldsymbol{w}^*) - f(\boldsymbol{w}^*)| + |\hat{f}_{t-1}(\boldsymbol{w}_t) - f(\boldsymbol{w}_t)|.$$

This is because if instead $\Delta_t > |\hat{f}_{t-1}(\boldsymbol{w}^*) - f(\boldsymbol{w}^*)| + |\hat{f}_{t-1}(\boldsymbol{w}_t) - f(\boldsymbol{w}_t)|$, we have

$$\hat{f}_{t-1}(\boldsymbol{w}^*) - \hat{f}_{t-1}(\boldsymbol{w}_t)$$
$$= \hat{f}_{t-1}(\boldsymbol{w}^*) - \hat{f}_{t-1}(\boldsymbol{w}_t) + (f(\boldsymbol{w}^*) - f(\boldsymbol{w}_t)) - (f(\boldsymbol{w}^*) - f(\boldsymbol{w}_t))$$
$$\geq \Delta_t - (f(\boldsymbol{w}^*) - \hat{f}_{t-1}(\boldsymbol{w}^*)) - (\hat{f}_{t-1}(\boldsymbol{w}_t) - f(\boldsymbol{w}_t))$$
$$\geq \Delta_t - |(f(\boldsymbol{w}^*) - \hat{f}_{t-1}(\boldsymbol{w}^*))| - |(\hat{f}_{t-1}(\boldsymbol{w}_t) - f(\boldsymbol{w}_t))|$$
$$> 0,$$

which contradicts the selection strategy of $\boldsymbol{w}_t$ in algorithm MC-Empirical. Thus, we obtain

$$\Delta_t \leq |\hat{f}_{t-1}(\boldsymbol{w}^*) - f(\boldsymbol{w}^*)| + |\hat{f}_{t-1}(\boldsymbol{w}_t) - f(\boldsymbol{w}_t)|$$
$$\leq h_t(\boldsymbol{w}^*) + h_t(\boldsymbol{w}_t)$$
$$= E_t(\boldsymbol{w}^*) + \rho \boldsymbol{w}^{*\top} G_t \boldsymbol{w}^* + E_t(\boldsymbol{w}_t) + \rho \boldsymbol{w}_t^\top G_t \boldsymbol{w}_t \qquad (8)$$

Now, for any $\boldsymbol{w} \in \triangle_d^c$, we have

$$\boldsymbol{w}^\top G_t \boldsymbol{w} = \sum_{i,j \in [d]} g(t) w_i w_j$$
$$= g(t) \sum_{i,j \in [d]} w_i w_j$$
$$= g(t) \left( \sum_i w_i \right)^2$$
$$= g(t)$$

and

$$E_t(\boldsymbol{w}) = \sqrt{2\beta(\delta_t)} \sqrt{\boldsymbol{w}^\top D_{t-1}^{-1} \left( \lambda \Lambda_\Sigma D_{t-1} + \sum_{s=1}^{t-1} \Sigma \right) D_{t-1}^{-1} \boldsymbol{w}}$$
$$= \sqrt{2\beta(\delta_t)} \sqrt{\lambda \boldsymbol{w}^\top D_{t-1}^{-1} \Lambda_\Sigma \boldsymbol{w} + \boldsymbol{w}^\top D_{t-1}^{-1} \left( \sum_{s=1}^{t-1} \Sigma \right) D_{t-1}^{-1} \boldsymbol{w}}$$
$$= \sqrt{2\beta(\delta_t)} \sqrt{\lambda \boldsymbol{w}^\top D_{t-1}^{-1} \Lambda_\Sigma \boldsymbol{w} + \boldsymbol{w}^\top D_{t-1}^{-1} \Sigma \boldsymbol{w}}$$
$$= \sqrt{2\beta(\delta_t)} \sqrt{\frac{1}{t-1} \lambda \boldsymbol{w}^\top \Lambda_\Sigma \boldsymbol{w} + \frac{1}{t-1} \boldsymbol{w}^\top \Sigma \boldsymbol{w}}$$
$$\leq \sqrt{\frac{2\beta(\delta_t)}{t-1}} \sqrt{\lambda \Sigma_{\max} + \boldsymbol{w}^\top \Sigma \boldsymbol{w}},$$

where $\Sigma_{\max}$ denotes the maximum diagonal entry of $\Sigma$.

For any $t \geq 7$, $\frac{3 \ln t}{t-1} < \sqrt{\frac{3 \ln t}{t-1}}$ and $g(t) \leq \left( 6 + 20\sqrt{3} \right) \frac{\ln t}{\sqrt{t-1}}$, Eq. (8) can be written as

$$\Delta_t \leq E_t(\boldsymbol{w}^*) + \rho \boldsymbol{w}^{*\top} G_t \boldsymbol{w}^* + E_t(\boldsymbol{w}_t) + \rho \boldsymbol{w}_t^\top G_t \boldsymbol{w}_t$$
$$\leq \sqrt{\frac{2\beta(\delta_t)}{t-1}} \left( \sqrt{\lambda \Sigma_{\max} + \boldsymbol{w}_t^\top \Sigma \boldsymbol{w}_t} + \sqrt{\lambda \Sigma_{\max} + \boldsymbol{w}^{*\top} \Sigma \boldsymbol{w}^*} \right) + 82\rho \frac{\ln t}{\sqrt{t-1}}$$

$$\leq \sqrt{\frac{2\beta(\delta_t)}{t-1}} \left(2\sqrt{\lambda\Sigma_{\max}} + \sqrt{\boldsymbol{w}_t^\top \Sigma \boldsymbol{w}_t} + \sqrt{\boldsymbol{w}^{*\top}\Sigma \boldsymbol{w}^*}\right) + 82\rho \frac{\ln t}{\sqrt{t-1}}$$

$$= \sqrt{\frac{2\beta(\delta_t)}{t-1}} \left(2\sqrt{\lambda\Sigma_{\max}} + \sqrt{\boldsymbol{w}^{*\top}\Sigma \boldsymbol{w}^*}\right) + 82\rho \frac{\ln t}{\sqrt{t-1}} + \sqrt{\frac{2\beta(\delta_t)}{t-1}} \cdot \sqrt{\boldsymbol{w}_t^\top \Sigma \boldsymbol{w}_t} \qquad (9)$$

Next, we investigate the upper bound of $\boldsymbol{w}_t^\top \Sigma \boldsymbol{w}_t$. According to Eq. (8), we have that $\boldsymbol{w}_t^\top \Sigma \boldsymbol{w}_t$ satisfies

$$\Delta_t \leq h_t(\boldsymbol{w}^*) + h_t(\boldsymbol{w}_t) \qquad (10)$$

In Eq. (10), we have

$$\begin{aligned}
\Delta_t &= f(\boldsymbol{w}^*) - f(\boldsymbol{w}_t) \\
&\geq \theta_{\min}^* - \rho \boldsymbol{w}^{*\top}\Sigma \boldsymbol{w}^* - \theta_{\max}^* + \rho \boldsymbol{w}_t^\top \Sigma \boldsymbol{w}_t
\end{aligned}$$

and

$$\begin{aligned}
h_t(\boldsymbol{w}^*) + h_t(\boldsymbol{w}_t) &= E_t(\boldsymbol{w}^*) + \rho \boldsymbol{w}^{*\top}G_t \boldsymbol{w}^* + E_t(\boldsymbol{w}_t) + \rho \boldsymbol{w}_t^\top G_t \boldsymbol{w}_t \\
&\leq \sqrt{\frac{2\beta(\delta_t)}{t-1}} \left(2\sqrt{\lambda\Sigma_{\max}} + \sqrt{\boldsymbol{w}_t^\top \Sigma \boldsymbol{w}_t} + \sqrt{\boldsymbol{w}^{*\top}\Sigma \boldsymbol{w}^*}\right) + 82\rho \frac{\ln t}{\sqrt{t-1}}.
\end{aligned}$$

Thus, $\boldsymbol{w}_t^\top \Sigma \boldsymbol{w}_t$ satisfies

$$\begin{aligned}
\theta_{\min}^* - \rho \boldsymbol{w}^{*\top}\Sigma \boldsymbol{w}^* - \theta_{\max}^* + \rho \boldsymbol{w}_t^\top \Sigma \boldsymbol{w}_t &\leq \sqrt{\frac{2\beta(\delta_t)}{t-1}} \left(2\sqrt{\lambda\Sigma_{\max}} + \sqrt{\boldsymbol{w}_t^\top \Sigma \boldsymbol{w}_t} + \sqrt{\boldsymbol{w}^{*\top}\Sigma \boldsymbol{w}^*}\right) \\
&\quad + 82\rho \frac{\ln t}{\sqrt{t-1}}
\end{aligned}$$

Rearranging the terms, we have

$$\begin{aligned}
\rho \boldsymbol{w}_t^\top \Sigma \boldsymbol{w}_t - \sqrt{\frac{2\beta(\delta_t)}{t-1}}\sqrt{\boldsymbol{w}_t^\top \Sigma \boldsymbol{w}_t} - \Bigg( &\theta_{\max}^* - \theta_{\min}^* + \rho \boldsymbol{w}^{*\top}\Sigma \boldsymbol{w}^* \\
&+ \sqrt{\frac{2\beta(\delta_t)}{t-1}}\left(2\sqrt{\lambda\Sigma_{\max}} + \sqrt{\boldsymbol{w}^{*\top}\Sigma \boldsymbol{w}^*}\right) + 82\rho \frac{\ln t}{\sqrt{t-1}}\Bigg) \leq 0
\end{aligned} \qquad (11)$$

Let $x = \boldsymbol{w}_t^\top \Sigma \boldsymbol{w}_t$ and $0 < \lambda < 1$. Define function

$$y(x) = \rho x - c_1 \sqrt{x} - c_2 \leq 0,$$

where $c_1 = \sqrt{\frac{2\beta(\delta_t)}{t-1}} > 0$ and $c_2 = \theta_{\max}^* - \theta_{\min}^* + \rho \boldsymbol{w}^{*\top}\Sigma \boldsymbol{w}^* + \sqrt{\frac{2\beta(\delta_t)}{t-1}}\left(2\sqrt{\lambda\Sigma_{\max}} + \sqrt{\boldsymbol{w}^{*\top}\Sigma \boldsymbol{w}^*}\right) + 82\rho \frac{\ln t}{\sqrt{t-1}} > 0$. When $t \geq t_0 \triangleq \max\left\{(1 + e/\lambda)^{\frac{d}{2}}, 7\right\}$, we have

$$\beta(\delta_t) = \ln(t \ln^2 t) + \ln\ln t + \frac{d}{2}\ln(1 + e/\lambda) \leq 5\ln t$$

Now since

$$y(x) = \rho x - c_1 \sqrt{x} - c_2$$

$$=\rho\left(\sqrt{x}-\frac{c_1}{2\rho}\right)^2-\frac{c_1^2}{4\rho}-c_2,$$

by letting $y(x)\le 0$, we have

$$x\le\left(\frac{c_1}{2\rho}+\sqrt{\frac{c_1^2}{4\rho^2}+\frac{c_2}{\rho}}\right)^2$$

$$\le 2\frac{c_1^2}{4\rho^2}+2\frac{c_1^2}{4\rho^2}+2\frac{c_2}{\rho}$$

$$=\frac{c_1^2}{\rho^2}+\frac{2c_2}{\rho}$$

Therefore

$$\boldsymbol{w}_t^\top\Sigma\boldsymbol{w}_t\le\frac{1}{\rho^2}\frac{2\beta(\delta_t)}{t-1}+\frac{2}{\rho}\left(\theta_{\max}^*-\theta_{\min}^*+\rho\boldsymbol{w}^{*\top}\Sigma\boldsymbol{w}^*+\sqrt{\frac{2\beta(\delta_t)}{t-1}}\left(2\sqrt{\lambda\Sigma_{\max}}+\sqrt{\boldsymbol{w}^{*\top}\Sigma\boldsymbol{w}^*}\right)\right.$$

$$\left.+82\rho\frac{\ln t}{\sqrt{t-1}}\right)$$

$$\le 2\boldsymbol{w}^{*\top}\Sigma\boldsymbol{w}^*+\frac{2}{\rho}\left(\theta_{\max}^*-\theta_{\min}^*\right)+\frac{2}{\rho}\sqrt{\frac{2\beta(\delta_t)}{t-1}}\left(2\sqrt{\lambda\Sigma_{\max}}+\sqrt{\boldsymbol{w}^{*\top}\Sigma\boldsymbol{w}^*}\right)+164\frac{\ln t}{\sqrt{t-1}}$$

$$+\frac{1}{\rho^2}\frac{2\beta(\delta_t)}{t-1}$$

Thus, we have that, $\boldsymbol{w}_t^\top\Sigma\boldsymbol{w}_t$ satisfies

$$\boldsymbol{w}_t^\top\Sigma\boldsymbol{w}_t\le\min\left\{2\boldsymbol{w}^{*\top}\Sigma\boldsymbol{w}^*+\frac{2}{\rho}\left(\theta_{\max}^*-\theta_{\min}^*\right)+\frac{2}{\rho}\sqrt{\frac{2\beta(\delta_t)}{t-1}}\left(2\sqrt{\lambda\Sigma_{\max}}+\sqrt{\boldsymbol{w}^{*\top}\Sigma\boldsymbol{w}^*}\right)\right.$$

$$\left.+164\frac{\ln t}{\sqrt{t-1}}+\frac{1}{\rho^2}\frac{2\beta(\delta_t)}{t-1},\ \boldsymbol{w}_{\max}^\top\Sigma\boldsymbol{w}_{\max}\right\},\tag{12}$$

where $\boldsymbol{w}_{\max}^\top\triangleq\mathrm{argmax}_{\boldsymbol{w}\in\triangle_d^c}\boldsymbol{w}^\top\Sigma\boldsymbol{w}$.

Below we discuss the two terms in Eq. (12) separately.

Case (i): Plugging the first term of the upper bound of $\boldsymbol{w}_t^\top\Sigma\boldsymbol{w}_t$ in Eq. (12) into Eq. (9), we have that for $t\ge t_0$,

$$\Delta_t\le\sqrt{\frac{2\beta(\delta_t)}{t-1}}\left(2\sqrt{\lambda\Sigma_{\max}}+\sqrt{\boldsymbol{w}^{*\top}\Sigma\boldsymbol{w}^*}\right)+82\rho\frac{\ln t}{\sqrt{t-1}}+\sqrt{\frac{2\beta(\delta_t)}{t-1}}\cdot\sqrt{\boldsymbol{w}_t^\top\Sigma\boldsymbol{w}_t}$$

$$\le\sqrt{\frac{2\beta(\delta_t)}{t-1}}\left(2\sqrt{\lambda\Sigma_{\max}}+\sqrt{\boldsymbol{w}^{*\top}\Sigma\boldsymbol{w}^*}\right)+82\rho\frac{\ln t}{\sqrt{t-1}}+\sqrt{\frac{2\beta(\delta_t)}{t-1}}\cdot$$

$$\sqrt{2\boldsymbol{w}^{*\top}\Sigma\boldsymbol{w}^*+\frac{2}{\rho}\left(\theta_{\max}^*-\theta_{\min}^*\right)+\frac{2}{\rho}\sqrt{\frac{2\beta(\delta_t)}{t-1}}\left(2\sqrt{\lambda\Sigma_{\max}}+\sqrt{\boldsymbol{w}^{*\top}\Sigma\boldsymbol{w}^*}\right)+164\frac{\ln t}{\sqrt{t-1}}+\frac{1}{\rho^2}\frac{2\beta(\delta_t)}{t-1}}$$

$$\le\sqrt{\frac{2\beta(\delta_t)}{t-1}}\left(2\sqrt{\lambda\Sigma_{\max}}+\sqrt{\boldsymbol{w}^{*\top}\Sigma\boldsymbol{w}^*}\right)+82\rho\frac{\ln t}{\sqrt{t-1}}+\sqrt{\frac{2\beta(\delta_t)}{t-1}}\cdot$$

$$\left( \sqrt{2\boldsymbol{w}^{*\top}\Sigma\boldsymbol{w}^*} + \frac{\sqrt{2}}{\sqrt{\rho}}\sqrt{\theta^*_{\max} - \theta^*_{\min}} + \frac{\sqrt{2}}{\sqrt{\rho}}\left(\frac{2\beta(\delta_t)}{t-1}\right)^{\frac{1}{4}}\left(\sqrt{2}\left(\lambda\Sigma_{\max}\right)^{\frac{1}{4}} + \left(\boldsymbol{w}^{*\top}\Sigma\boldsymbol{w}^*\right)^{\frac{1}{4}}\right) \right.$$

$$\left. + 13\frac{\sqrt{\ln t}}{(t-1)^{\frac{1}{4}}} + \frac{1}{\rho}\sqrt{\frac{2\beta(\delta_t)}{t-1}}\right)$$

$$\leq \sqrt{\frac{2\beta(\delta_t)}{t-1}}\left(2\sqrt{\lambda\Sigma_{\max}} + \sqrt{\boldsymbol{w}^{*\top}\Sigma\boldsymbol{w}^*} + \sqrt{2\boldsymbol{w}^{*\top}\Sigma\boldsymbol{w}^*} + \frac{\sqrt{2}}{\sqrt{\rho}}\sqrt{\theta^*_{\max} - \theta^*_{\min}}\right) + 82\rho\frac{\ln t}{\sqrt{t-1}}$$

$$+ \frac{\sqrt{2}}{\sqrt{\rho}}\left(\frac{2\beta(\delta_t)}{t-1}\right)^{\frac{3}{4}}\left(\sqrt{2}\left(\lambda\Sigma_{\max}\right)^{\frac{1}{4}} + \left(\boldsymbol{w}^{*\top}\Sigma\boldsymbol{w}^*\right)^{\frac{1}{4}}\right) + 42\frac{\ln t}{(t-1)^{\frac{3}{4}}} + \frac{1}{\rho}\frac{2\beta(\delta_t)}{t-1}$$

$$\leq \sqrt{\frac{2\beta(\delta_t)}{t-1}}\left(2\sqrt{\lambda\Sigma_{\max}} + \sqrt{\boldsymbol{w}^{*\top}\Sigma\boldsymbol{w}^*} + \sqrt{2\boldsymbol{w}^{*\top}\Sigma\boldsymbol{w}^*} + \frac{\sqrt{2}}{\sqrt{\rho}}\sqrt{\theta^*_{\max} - \theta^*_{\min}}\right) + 82\rho\frac{\ln t}{\sqrt{t-1}}$$

$$+ \frac{\sqrt{2}}{\sqrt{\rho}}\left(\frac{2\beta(\delta_t)}{t-1}\right)^{\frac{3}{4}}\left(\sqrt{2}\left(\lambda\Sigma_{\max}\right)^{\frac{1}{4}} + \left(\boldsymbol{w}^{*\top}\Sigma\boldsymbol{w}^*\right)^{\frac{1}{4}}\right) + 42\frac{\ln t}{(t-1)^{\frac{3}{4}}} + \frac{1}{\rho}\frac{2\beta(\delta_t)}{t-1}$$

$$\leq \sqrt{\frac{\ln t}{t-1}}\left(2\sqrt{\lambda\Sigma_{\max}} + \sqrt{\boldsymbol{w}^{*\top}\Sigma\boldsymbol{w}^*} + \sqrt{2\boldsymbol{w}^{*\top}\Sigma\boldsymbol{w}^*} + \frac{\sqrt{2}}{\sqrt{\rho}}\sqrt{\theta^*_{\max} - \theta^*_{\min}}\right) + 82\rho\frac{\ln t}{\sqrt{t-1}}$$

$$+ 42\frac{\ln t}{(t-1)^{\frac{3}{4}}}\left(\frac{1}{\sqrt{\rho}} + 1\right) + \frac{1}{\rho}\frac{10\ln t}{t-1}$$

According to Lemmas 3 and 4, for any $t \geq 2$, we bound the probability of event $\neg(\mathcal{E}_t \cap \mathcal{F}_t)$ as follows.

$$\Pr\left[\neg(\mathcal{E}_t \cap \mathcal{F}_t)\right] \leq \frac{10d^2}{t^2} + \frac{1}{t\ln^2 t}$$

$$\leq \frac{10d^2}{t\ln^2 t} + \frac{1}{t\ln^2 t}$$

$$= \frac{11d^2}{t\ln^2 t}$$

Recall that $t_0 = \max\left\{(1 + e/\lambda)^{\frac{d}{2}}, 7\right\}$. For any horizon $T$, summing over $t = 1, \ldots, t_0$ and $t = t_0, \ldots, T$, we obtain the regret upper bound

$$\mathbb{E}[\mathcal{R}(T)] = O(t_0\Delta_{\max}) + \sum_{t=t_0}^{T} O\left(\Delta_{\max} \cdot \Pr\left[\neg(\mathcal{E}_t \cap \mathcal{F}_t)\right] + \Delta_t \cdot \mathbb{I}\{\mathcal{E}_t \cap \mathcal{F}_t\}\right)$$

$$= O(\lambda^{-\frac{d}{2}}\Delta_{\max}) + \sum_{t=t_0}^{T} O\left(\Delta_{\max} \cdot \frac{d^2}{t\ln^2 t}\right) + \sum_{t=t_0}^{T} O\left(\Delta_t \cdot \mathbb{I}\{\mathcal{E}_t \cap \mathcal{F}_t\}\right)$$

$$= O(\lambda^{-\frac{d}{2}}\Delta_{\max}) + \sum_{t=t_0}^{T} O\left(\sqrt{\frac{\ln t}{t-1}}\left(\sqrt{\lambda\Sigma_{\max}} + \sqrt{\boldsymbol{w}^{*\top}\Sigma\boldsymbol{w}^*} + \frac{1}{\sqrt{\rho}}\sqrt{\theta^*_{\max} - \theta^*_{\min}}\right)\right.$$

$$\left. + \rho\frac{\ln t}{\sqrt{t-1}} + \frac{\ln t}{(t-1)^{\frac{3}{4}}}\left(\frac{1}{\sqrt{\rho}} + 1\right) + \frac{1}{\rho}\frac{\ln t}{t-1}\right)$$

$$=O(\lambda^{-\frac{d}{2}}\Delta_{\max}) + O\left(\ln T\sqrt{T}\left(\sqrt{\lambda\Sigma_{\max}} + \sqrt{\boldsymbol{w}^{*\top}\Sigma\boldsymbol{w}^*} + \frac{1}{\sqrt{\rho}}\sqrt{\theta^*_{\max} - \theta^*_{\min}} + \rho\right)\right.$$

$$\left. + \ln T \cdot T^{\frac{1}{4}}\left(\frac{1}{\sqrt{\rho}} + 1\right) + \ln^2 T \cdot \frac{1}{\rho}\right)$$

Case (ii): Plugging the second term of the upper bound of $\boldsymbol{w}_t^\top \Sigma \boldsymbol{w}_t$ in Eq. (12) into Eq. (9), we have that for $t \geq t_0$,

$$\Delta_t \leq \sqrt{\frac{2\beta(\delta_t)}{t-1}}\left(2\sqrt{\lambda\Sigma_{\max}} + \sqrt{\boldsymbol{w}_{\max}^\top\Sigma\boldsymbol{w}_{\max}} + \sqrt{\boldsymbol{w}^{*\top}\Sigma\boldsymbol{w}^*}\right) + 82\rho\frac{\ln t}{\sqrt{t-1}}$$

$$\leq 2\sqrt{\frac{2\beta(\delta_t)}{t-1}}\left(\sqrt{\lambda\Sigma_{\max}} + \sqrt{\boldsymbol{w}_{\max}^\top\Sigma\boldsymbol{w}_{\max}}\right) + 82\rho\frac{\ln t}{\sqrt{t-1}}$$

For any horizon $T$, summing over $t = 1, \ldots, t_0$ and $t = t_0, \ldots, T$, we obtain the regret upper bound

$$\mathbb{E}[\mathcal{R}(T)] = O(t_0\Delta_{\max}) + \sum_{t=t_0}^T O\left(\Delta_{\max} \cdot \Pr\left[\neg(\mathcal{E}_t \cap \mathcal{F}_t)\right] + \Delta_t \cdot \mathbb{I}\{\mathcal{E}_t \cap \mathcal{F}_t\}\right)$$

$$= O(\lambda^{-\frac{d}{2}}\Delta_{\max}) + \sum_{t=t_0}^T O\left(\Delta_{\max} \cdot \frac{d^2}{t\ln^2 t}\right) + \sum_{t=t_0}^T O\left(\Delta_t \cdot \mathbb{I}\{\mathcal{E}_t \cap \mathcal{F}_t\}\right)$$

$$= O(\lambda^{-\frac{d}{2}}\Delta_{\max}) + \sum_{t=t_0}^T O\left(\sqrt{\frac{\beta(\delta_t)}{t-1}}\left(\sqrt{\lambda\Sigma_{\max}} + \sqrt{\boldsymbol{w}_{\max}^\top\Sigma\boldsymbol{w}_{\max}}\right) + \rho\frac{\ln t}{\sqrt{t-1}}\right)$$

$$= O(\lambda^{-\frac{d}{2}}\Delta_{\max}) + \sum_{t=t_0}^T O\left(\sqrt{\frac{\ln t}{t-1}}\left(\sqrt{\lambda\Sigma_{\max}} + \sqrt{\boldsymbol{w}_{\max}^\top\Sigma\boldsymbol{w}_{\max}}\right) + \rho\frac{\ln t}{\sqrt{t-1}}\right)$$

$$= O(\lambda^{-\frac{d}{2}}\Delta_{\max}) + O\left(\ln T\sqrt{T}\left(\sqrt{\lambda\Sigma_{\max}} + \sqrt{\boldsymbol{w}_{\max}^\top\Sigma\boldsymbol{w}_{\max}} + \rho\right)\right)$$

Combining both cases (i) and (ii), we can obtain

$$\mathbb{E}[\mathcal{R}(T)] = O\left(\min\left\{\ln T\sqrt{T}\left(\sqrt{\boldsymbol{w}^{*\top}\Sigma\boldsymbol{w}^*} + \rho^{-\frac{1}{2}}\sqrt{\theta^*_{\max} - \theta^*_{\min}}\right) + \ln T \cdot T^{\frac{1}{4}}\left(\rho^{-\frac{1}{2}} + 1\right)\right.\right.$$

$$\left.\left. + \ln^2 T \cdot \rho^{-1}, \ln T\sqrt{T}\sqrt{\boldsymbol{w}_{\max}^\top\Sigma\boldsymbol{w}_{\max}}\right\} + \ln T\sqrt{T}\left(\sqrt{\lambda\Sigma_{\max}} + \rho\right) + \lambda^{-\frac{d}{2}}\Delta_{\max}\right)$$

Let $\Sigma^*_{\max} = \max_{i\in[d]}\Sigma^*_{ii}$. Setting $\lambda = \frac{\boldsymbol{w}^{*\top}\Sigma\boldsymbol{w}^*}{\Sigma_{\max}}$ and ignoring the terms of $o(\ln T\sqrt{T})$ order, we obtain

$$\mathbb{E}[\mathcal{R}(T)] = O\left(\left(\min\left\{\sqrt{\boldsymbol{w}^{*\top}\Sigma\boldsymbol{w}^*} + \rho^{-\frac{1}{2}}\sqrt{\theta^*_{\max} - \theta^*_{\min}}, \sqrt{\boldsymbol{w}_{\max}^\top\Sigma\boldsymbol{w}_{\max}}\right\} + \rho\right)\ln T\sqrt{T}\right)$$

$$= O\left(\left(\min\left\{\sqrt{\boldsymbol{w}^{*\top}\Sigma\boldsymbol{w}^*} + \rho^{-\frac{1}{2}}\sqrt{\theta^*_{\max} - \theta^*_{\min}}, \sqrt{\Sigma_{\max}}\right\} + \rho\right)\ln T\sqrt{T}\right)$$

This completes the proof. $\qquad\square$

## B.2 Proof of Theorem 2

In order to prove Theorem 2, we first analyze the offline problem of CMCB-FI. Suppose that the covariance matrix $\Sigma$ is positive definite.

**(Offline Problem of CMCB-FI)** We define the quadratic optimization $\mathsf{QuadOpt}(\boldsymbol{\theta}^*, \Sigma)$ as

$$\min_{\boldsymbol{w}} \quad f(\boldsymbol{w}) = \rho \boldsymbol{w}^\top \Sigma \boldsymbol{w} - \boldsymbol{w}^\top \boldsymbol{\theta}^*$$

$$s.t. \quad w_i \geq 0, \quad \forall i \in [d]$$

$$\sum_{i=1}^{d} w_i = 1$$

and $\boldsymbol{w}^*$ as the optimal solution to $\mathsf{QuadOpt}(\boldsymbol{\theta}^*, \Sigma)$. We consider the KKT condition for this quadratic optimization as follows:

$$2\rho \Sigma \boldsymbol{w} - \boldsymbol{\theta}^* - \boldsymbol{u} - v\mathbf{1} = 0$$

$$w_i u_i = 0, \quad \forall i \in [d]$$
$$u_i \geq 0, \quad \forall i \in [d]$$
$$w_i \geq 0, \quad \forall i \in [d]$$

$$\sum_{i=1}^{d} w_i = 1$$

Let $S \subseteq [d]$ be a subset of indexes for $\boldsymbol{w}$ such that $S = \{i \in [d] : w_i > 0\}$. Let $\bar{S} = [d] \setminus S$ and we have $\bar{S} = \{i \in [d] : w_i = 0\}$. Then, from the KKT condition, we have

$$\boldsymbol{w}_S = \frac{1}{2\rho} \Sigma_S^{-1} \boldsymbol{\theta}_S^* + \frac{1 - \|\frac{1}{2\rho}\Sigma_S^{-1}\boldsymbol{\theta}_S^*\|}{\|\Sigma_S^{-1}\mathbf{1}\|} \Sigma_S^{-1}\mathbf{1} \succ \mathbf{0} \tag{13}$$

$$\boldsymbol{w}_{\bar{S}} = \mathbf{0}$$

$$v = \frac{2\rho(1 - \|\frac{1}{2\rho}\Sigma_S^{-1}\boldsymbol{\theta}_S^*\|)}{\|\Sigma_S^{-1}\mathbf{1}\|}$$

$$\boldsymbol{u} = 2\rho\Sigma\boldsymbol{w} - \boldsymbol{\theta}^* - v\mathbf{1} \succeq \mathbf{0}$$

Since this problem is a quadratic optimization and the covariance matrix $\Sigma$ is positive-definite, there is a unique feasible $S$ satisfying the above inequalities and the solution $\boldsymbol{w}_S, \boldsymbol{w}_{\bar{S}}$ is the optimal solution $\boldsymbol{w}^*$.

**Main Proof.** Now, we give the proof of Theorem 2.

*Proof.* (Theorem 2) First, we choose prior distributions for $\boldsymbol{\theta}^*$ and $\Sigma$. We assume that $\boldsymbol{\theta}^* \sim \mathcal{N}(0, \frac{1}{\omega}I)$ and $\Sigma \sim \pi_I$, where $\omega > 0$ and $\pi_I$ takes probability 1 at the support $I$ and probability 0 anywhere else. Define $\hat{\boldsymbol{\theta}}_t \triangleq \frac{1}{t}\sum_{i=1}^{t} \theta_i$ and $\boldsymbol{\mu}_t = \frac{t}{t+\omega}\hat{\boldsymbol{\theta}}_t$. Then, we see that $\hat{\boldsymbol{\theta}}_t \sim \mathcal{N}(0, \frac{\omega+t}{t\omega}I)$ and $\boldsymbol{\mu}_t \sim \mathcal{N}(0, \frac{t}{(t+\omega)\omega}I)$.

Thus, the posterior of $\boldsymbol{\theta}^*$ is given by

$$\boldsymbol{\theta}^*|\theta_1, \ldots, \theta_t, \Sigma \sim \mathcal{N}\left(\frac{t}{t+\omega}\hat{\boldsymbol{\theta}}_t, \frac{1}{t+\omega}I\right) = \mathcal{N}\left(\boldsymbol{\mu}_t, \frac{1}{t+\omega}I\right).$$

The posterior of $\Sigma$ is still $\Sigma \sim \pi_I$, i.e., $\Sigma$ is always a fixed identity matrix. Under the Bayesian setting, the expected regret is givenn by

$$\sum_{t=1}^{T} \mathbb{E}_{\boldsymbol{\mu}_t \sim \mathcal{N}(0, \frac{t}{(t+\omega)\omega}I)}\left[\mathbb{E}_{\boldsymbol{\theta}^*|\boldsymbol{\mu}_t \sim \mathcal{N}(\boldsymbol{\mu}_t, \frac{1}{t+\omega}I)}\left[f(\boldsymbol{w}^*) - f(\boldsymbol{w}_t)\right]\right].$$

Recall that $\boldsymbol{w}^*$ is the optimal solution to $\mathsf{QuadOpt}(\boldsymbol{\theta}^*, \Sigma)$. It can be seen that the best strategy of $\boldsymbol{w}_t$ at timestep $t$ is to select the optimal solution to $\mathsf{QuadOpt}(\boldsymbol{\mu}_t, \Sigma)$ and we use algorithm $\mathcal{A}$ to denote this strategy. Thus, to obtain a regret lower bound for the problem, it suffices to prove a regret lower bound of algorithm $\mathcal{A}$ for the problem.

Below we prove a regret lower bound of algorithm $\mathcal{A}$ for the problem.

**Step (i).** We consider the case when $\boldsymbol{w}^*$ and $\boldsymbol{w}_t$ both lie in the interior of the $d$-dimensional probability simplex $\triangle_d$, i.e., $w_i^* > 0, \forall i \in [d]$ and $w_{t,i} > 0, \forall i \in [d]$. From Eq. (13), $\boldsymbol{w}^*$ satisfies

$$\frac{1}{2\rho} I^{-1} \boldsymbol{\theta}^* + \frac{1 - \|\frac{1}{2\rho} I^{-1} \boldsymbol{\theta}^*\|}{\|I^{-1}\mathbf{1}\|} I^{-1}\mathbf{1} \succ \mathbf{0}.$$

Rearranging the terms, we have

$$\|\boldsymbol{\theta}^*\|\mathbf{1} - d\boldsymbol{\theta}^* \prec 2\rho\mathbf{1},$$

which is equivalent to

$$\begin{cases} \theta_2^* + \cdots + \theta_d^* - (d-1)\theta_1^* & < 2\rho \\ \qquad\qquad \vdots \\ \theta_1^* + \cdots + \theta_{d-1}^* - (d-1)\theta_d^* & < 2\rho \end{cases} \tag{14}$$

Similarly, $\boldsymbol{w}_t$ satisfies

$$\begin{cases} \mu_2 + \cdots + \mu_d - (d-1)\mu_1 & < 2\rho \\ \qquad\qquad \vdots \\ \mu_1 + \cdots + \mu_{d-1} - (d-1)\mu_d & < 2\rho \end{cases} \tag{15}$$

We first derive a condition that makes $\boldsymbol{\mu}_t$ lie in the interior of $\triangle_d$. Recall that $\boldsymbol{\mu}_t \sim \mathcal{N}(0, \frac{t}{(t+\omega)\omega} I)$. Define event

$$\mathcal{E}_t \triangleq \left\{ -3\sqrt{\frac{t}{(t+\omega)\omega}} \leq \mu_t \leq 3\sqrt{\frac{t}{(t+\omega)\omega}} \right\}.$$

According to the $3 - \sigma$ principle for Gaussian distributions, we have

$$\Pr[\mathcal{E}_t] \geq (99.7\%)^d.$$

Conditioning on $\mathcal{E}_t$, under which Eq. (15) hold, it suffices to let

$$3(d-1)\sqrt{\frac{t}{(t+\omega)\omega}} - \left( -3(d-1)\sqrt{\frac{t}{(t+\omega)\omega}} \right) < \rho,$$

which is equivalent to

$$\left(1 + \frac{\omega}{t}\right)\omega > \frac{36(d-1)^2}{\rho^2}, \tag{16}$$

when $t > 0, m > 0, t + \omega > 0$. Let $t_1 > 0$ be the smallest timestep that satisfies Eq. (16). Thus, when $\mathcal{E}_t$ occurs and $t \geq t_1$, $\boldsymbol{\mu}_t$ lie in the interior of $\triangle_d$.

Next, we derive some condition that make $\boldsymbol{\theta}^*$ lie in the interior of $\triangle_d$. Recall that $\boldsymbol{\theta}^* | \boldsymbol{\mu}_t \sim \mathcal{N}(\boldsymbol{\mu}_t, \frac{1}{t+\omega} I)$.

Fix $\boldsymbol{\mu}_t$, and then we define event

$$\mathcal{F}_t \triangleq \left\{ -3\sqrt{\frac{t}{(t+\omega)\omega}} \leq \boldsymbol{\theta}^* - \boldsymbol{\mu}_t \leq 3\sqrt{\frac{t}{(t+\omega)\omega}} \right\}.$$

According to the $3-\sigma$ principle for Gaussian distributions, we have

$$\Pr\left[\mathcal{F}_t\right] \geq (99.7\%)^d.$$

Conditioning on $\mathcal{F}_t$, in order to let Eq. (14) hold, it suffices to let

$$3(d-1)\frac{1}{\sqrt{t+\omega}} - \left(-3(d-1)\frac{1}{\sqrt{t+\omega}}\right) < \rho,$$

which is equivalent to

$$t > \frac{36(d-1)^2}{\rho^2} - \omega, \tag{17}$$

when $t+\omega > 0$. Let $t_2 > 0$ be the smallest timestep that satisfies Eq. (17). Thus, when $\mathcal{F}_t$ occurs and $t \geq t_2$, $\boldsymbol{\theta}^*$ lie in the interior of $\triangle_d$.

**Step (ii).** Suppose that $\mathcal{E}_t \cap \mathcal{F}_t \cap \mathcal{G}_t$ occurs and consider $t \geq \tilde{t} \triangleq \max\{t_1, t_2\}$. Then, $\boldsymbol{w}^*$ and $\boldsymbol{w}_t$ both lie in the interior of $\triangle_d$, i.e., $w_i^* > 0, \forall i \in [d]$ and $w_{t,i} > 0, \forall i \in [d]$. We have

$$\boldsymbol{w}^* = \frac{1}{2\rho}\Sigma^{-1}\boldsymbol{\theta}^* + \frac{1 - \|\frac{1}{2\rho}\Sigma^{-1}\boldsymbol{\theta}^*\|}{\|\Sigma^{-1}\mathbf{1}\|}\Sigma^{-1}\mathbf{1}$$

$$\boldsymbol{w}_t = \frac{1}{2\rho}\Sigma^{-1}\boldsymbol{\mu}_t + \frac{1 - \|\frac{1}{2\rho}\Sigma^{-1}\boldsymbol{\mu}_t\|}{\|\Sigma^{-1}\mathbf{1}\|}\Sigma^{-1}\mathbf{1}$$

Let $\Delta\boldsymbol{\theta}_t \triangleq \boldsymbol{\mu}_t - \boldsymbol{\theta}^*$ and thus $\Delta\boldsymbol{\theta}_t|\boldsymbol{\mu}_t \sim \mathcal{N}(0, \frac{1}{t+\omega}I)$. Let $\Delta\boldsymbol{w}_t \triangleq \boldsymbol{w}_t - \boldsymbol{w}^* = \frac{1}{2\rho}\Sigma^{-1}\Delta\boldsymbol{\theta}_t - \frac{\|\frac{1}{2\rho}\Sigma^{-1}\Delta\boldsymbol{\theta}_t\|}{\|\Sigma^{-1}\mathbf{1}\|}\Sigma^{-1}\mathbf{1} = \frac{1}{2\rho}\Delta\boldsymbol{\theta}_t - \frac{1}{2\rho d}\|\Delta\boldsymbol{\theta}_t\|\mathbf{1}$. Then, we have

$$
\begin{aligned}
f(\boldsymbol{w}^*) - f(\boldsymbol{w}_t) =& f(\boldsymbol{w}^*) - f(\boldsymbol{w}^* + \Delta\boldsymbol{w}_t) \\
=& \left((\boldsymbol{w}^*)^\top\boldsymbol{\theta}^* - \rho(\boldsymbol{w}^*)^\top\Sigma\boldsymbol{w}^*\right) \\
& - \left((\boldsymbol{w}^*)^\top\boldsymbol{\theta}^* + (\Delta\boldsymbol{w}_t)^\top\boldsymbol{\theta}^* - \rho(\boldsymbol{w}^*)^\top\Sigma\boldsymbol{w}^* - 2\rho(\Delta\boldsymbol{w}_t)^\top\Sigma\boldsymbol{w}^* - \rho(\Delta\boldsymbol{w}_t)^\top\Sigma\Delta\boldsymbol{w}_t\right) \\
=& -(\Delta\boldsymbol{w}_t)^\top\boldsymbol{\theta}^* + 2\rho(\Delta\boldsymbol{w}_t)^\top\Sigma\boldsymbol{w}^* + \rho(\Delta\boldsymbol{w}_t)^\top\Sigma\Delta\boldsymbol{w}_t \\
=& (\Delta\boldsymbol{w}_t)^\top\nabla f(\boldsymbol{\theta}^*) + \rho(\Delta\boldsymbol{w}_t)^\top\Sigma\Delta\boldsymbol{w}_t \\
\geq& \rho(\Delta\boldsymbol{w}_t)^\top\Sigma\Delta\boldsymbol{w}_t \\
=& \rho(\Delta\boldsymbol{w}_t)^\top\Delta\boldsymbol{w}_t \\
=& \rho\left(\frac{1}{4\rho^2}(\Delta\boldsymbol{\theta}_t)^\top\Delta\boldsymbol{\theta}_t - \frac{1}{4\rho^2 d^2}\|\Delta\boldsymbol{\theta}_t\|^2 d\right) \\
=& \frac{1}{4\rho}\left(\sum_{i=1}^d \Delta\theta_{t,i}^2 - \frac{\left(\sum_{i=1}^d \Delta\theta_{t,i}\right)^2}{d}\right) \\
=& \frac{1}{4\rho d}\sum_{1\leq i<j\leq d}(\Delta\theta_{t,i} - \Delta\theta_{t,j})^2
\end{aligned}
$$

Let $i_1 = 1, j_1 = 2, i_2 = 3, j_2 = 4, \ldots, i_{\lceil \frac{d}{2} \rceil} = 2 \lceil \frac{d}{2} \rceil - 1, j_{\lceil \frac{d}{2} \rceil} = 2 \lceil \frac{d}{2} \rceil$. For any $i_k, j_k$ ($k \in [[\frac{d}{2}]]$), $\Delta\theta_{t,i_k} - \Delta\theta_{t,j_k} | \boldsymbol{\mu}_t \sim \mathcal{N}(0, \frac{2}{t+\omega})$ and they are mutually independent among $k$.

Fix $\boldsymbol{\mu}_t$, and then we define event

$$\mathcal{G}_t \triangleq \left\{ |\Delta\theta_{t,i_k} - \Delta\theta_{t,j_k}| \geq 0.3\sqrt{\frac{2}{t+\omega}}, \forall k \in \left[\left[\frac{d}{2}\right]\right] \right\}.$$

From the c.d.f. of Gaussian distributions, we have

$$\Pr[\mathcal{G}_t] \geq (75\%)^{\lceil \frac{d}{2} \rceil}.$$

From this, we get

$$
\begin{aligned}
\Pr[\mathcal{F}_t \cap \mathcal{G}_t] &\geq 1 - \Pr[\bar{\mathcal{F}}_t] - \Pr[\bar{\mathcal{G}}_t] \\
&\geq 1 - \left(1 - (99.7\%)^d\right) - \left(1 - (75\%)^{\lceil \frac{d}{2} \rceil}\right) \\
&\geq (99.7\%)^d + (75\%)^{\lceil \frac{d}{2} \rceil} - 1.
\end{aligned}
$$

When $d \leq 18$, $\Pr[\mathcal{F}_t \cap \mathcal{G}_t] \geq (99.7\%)^d + (75\%)^{\lceil \frac{d}{2} \rceil} - 1 > 0$.

**Step (iii).** We bound the expected regret by considering the event $\mathcal{E}_t \cap \mathcal{F}_t \cap \mathcal{G}_t$ and $t \geq \tilde{t}$. Specifically,

$$
\sum_{t=1}^{T} \mathbb{E}_{\boldsymbol{\mu}_t \sim \mathcal{N}(0, \frac{t}{(t+\omega)\omega} I)} \left[ \mathbb{E}_{\boldsymbol{\theta}^* | \boldsymbol{\mu}_t \sim \mathcal{N}(\boldsymbol{\mu}_t, \frac{1}{t+\omega} I)} [f(\boldsymbol{w}^*) - f(\boldsymbol{w}_t)] \right]
$$

$$
\geq \sum_{t=\tilde{t}}^{T} \mathbb{E}_{\boldsymbol{\mu}_t \sim \mathcal{N}(0, \frac{t}{(t+\omega)\omega} I)} \left[ \mathbb{E}_{\boldsymbol{\theta}^* | \boldsymbol{\mu}_t \sim \mathcal{N}(\boldsymbol{\mu}_t, \frac{1}{t+\omega} I)} [f(\boldsymbol{w}^*) - f(\boldsymbol{w}_t)] | \mathcal{E}_t \right] \Pr[\mathcal{E}_t]
$$

$$
\geq \sum_{t=\tilde{t}}^{T} \mathbb{E}_{\boldsymbol{\mu}_t \sim \mathcal{N}(0, \frac{t}{(t+\omega)\omega} I)} \left[ \mathbb{E}_{\boldsymbol{\theta}^* | \boldsymbol{\mu}_t \sim \mathcal{N}(\boldsymbol{\mu}_t, \frac{1}{t+\omega} I)} [f(\boldsymbol{w}^*) - f(\boldsymbol{w}_t) | \mathcal{F}_t \cap \mathcal{G}_t] \Pr[\mathcal{F}_t \cap \mathcal{G}_t] | \mathcal{E}_t \right] \Pr[\mathcal{E}_t]
$$

$$
\geq \sum_{t=\tilde{t}}^{T} \mathbb{E}_{\boldsymbol{\mu}_t \sim \mathcal{N}(0, \frac{t}{(t+\omega)\omega} I)} \left[ \mathbb{E}_{\boldsymbol{\theta}^* | \boldsymbol{\mu}_t \sim \mathcal{N}(\boldsymbol{\mu}_t, \frac{1}{t+\omega} I)} \left[ \frac{1}{4\rho d} \sum_{1 \leq i < j \leq d} (\Delta\theta_{t,i} - \Delta\theta_{t,j})^2 | \mathcal{F}_t \cap \mathcal{G}_t \cap \mathcal{E}_t \right] \cdot \Pr[\mathcal{F}_t \cap \mathcal{G}_t] \right] \Pr[\mathcal{E}_t]
$$

$$
\geq \sum_{t=\tilde{t}}^{T} \mathbb{E}_{\boldsymbol{\mu}_t \sim \mathcal{N}(0, \frac{t}{(t+\omega)\omega} I)} \left[ \mathbb{E}_{\boldsymbol{\theta}^* | \boldsymbol{\mu}_t \sim \mathcal{N}(\boldsymbol{\mu}_t, \frac{1}{t+\omega} I)} \left[ \frac{\lceil \frac{d}{2} \rceil}{4\rho d} \cdot \frac{0.3^2 \cdot 2}{t+\omega} | \mathcal{F}_t \cap \mathcal{G}_t \cap \mathcal{E}_t \right] \cdot \left( (99.7\%)^d + (75\%)^{\lceil \frac{d}{2} \rceil} - 1 \right) \right] (99.7\%)^d
$$

$$
= \sum_{t=\tilde{t}}^{T} \frac{0.01125}{\rho(t+\omega)} \left( (99.7\%)^d + (75\%)^{\lceil \frac{d}{2} \rceil} - 1 \right) (99.7\%)^d
$$

$$
= \frac{0.01125 \left( (99.7\%)^d + (75\%)^{\lceil \frac{d}{2} \rceil} - 1 \right) (99.7\%)^d}{\rho} \ln\left( \frac{t+\omega}{\tilde{t}+\omega} \right).
$$

In the following, we consider an intrinsic bound of the expected regret and set the problem parameters to proper quantities. Since the expected regret is upper bounded by $\Delta_{\max} T$ and

$$\Delta_{\max} = f_{\max} - f_{\min}$$

$$\geq f\left(\frac{1}{d}\mathbf{1}\right) - (\theta^*_{\min} - \rho)$$

$$= \frac{1}{d}\sum_{i=1}^{d}\theta^*_i - \rho\frac{1}{d} - (\theta^*_{\min} - \rho)$$

$$= \frac{1}{d}\sum_{i=1}^{d}\theta^*_i - \theta^*_{\min} + \frac{d-1}{d}\rho$$

$$>0,$$

we conclude that the expected regret is lower bounded by

$$\min\left\{\frac{0.01125\left((99.7\%)^d + (75\%)^{\lceil\frac{d}{2}\rceil} - 1\right)(99.7\%)^d}{\rho}\ln\left(\frac{t+\omega}{\tilde{t}+m}\right),\left(\frac{1}{d}\sum_{i=1}^{d}\theta^*_i - \theta^*_{\min} + \frac{d-1}{d}\rho\right)T\right\}.$$

Choose $\omega = 36(d-1)^2 T$ and $\rho = \frac{1}{\sqrt{T}}$. According to Eqs. (16) and (17), we have $t_1 = t_2 = \tilde{t} = 1$. Therefore, for $d \leq 18$, the expected regret is lower bounded by $\Omega(\sqrt{T})$.

$\square$

## C  Proof for CMCB-SB

### C.1  Proof of Theorem 3

*Proof.* (Theorem 3) Denote $\Delta_t = f(\boldsymbol{w}^*) - f(\boldsymbol{w}_t)$, $\bar{f}_t(\boldsymbol{w}) = \boldsymbol{w}^\top\hat{\boldsymbol{\theta}}_{t-1} + E_t(\boldsymbol{w}) - \rho\boldsymbol{w}^\top\underline{\Sigma}_{t-1}\boldsymbol{w}$ and $G_t$ the matrix whose $ij$-th entry is $g_{ij}(t)$.

For $t \geq d^2 + 1$, suppose that event $\mathcal{G}_t \cap \mathcal{H}_t$ occurs. Define $h_t(\boldsymbol{w}) \triangleq E_t(\boldsymbol{w}) + \rho\boldsymbol{w}^\top G_t\boldsymbol{w}$ for any $\boldsymbol{w} \in \triangle_d^c$. Then, we have

$$0 \leq \bar{f}_t(\boldsymbol{w}) - f(\boldsymbol{w}) \leq 2h_t(\boldsymbol{w}).$$

According to the selection strategy of $\boldsymbol{w}_t$, we obtain

$$\begin{aligned}
f(\boldsymbol{w}^*) - f(\boldsymbol{w}_t) &\leq \bar{f}_t(\boldsymbol{w}^*) - f(\boldsymbol{w}_t) \\
&\leq \bar{f}_t(\boldsymbol{w}_t) - f(\boldsymbol{w}_t) \\
&\leq 2h_t(\boldsymbol{w}_t)
\end{aligned}$$

Thus, for any $T \geq d^2 + 1$, we have that

$$\sum_{t=d^2+1}^{T}\left(f(\boldsymbol{w}^*) - f(\boldsymbol{w}_t)\right)$$

$$\leq 2\sum_{t=d^2+1}^{T}h_t(\boldsymbol{w}_t)$$

$$= 2\sum_{t=d^2+1}^{T}\left(\sqrt{2\beta(\delta_t)}\sqrt{\boldsymbol{w}^\top D_{t-1}^{-1}(\lambda\Lambda_{\bar{\Sigma}_t}D_{t-1} + \sum_{s=1}^{t-1}\bar{\Sigma}_{s,\boldsymbol{w}_s})D_{t-1}^{-1}\boldsymbol{w} + \rho\boldsymbol{w}^\top G_t\boldsymbol{w}}\right)$$

$$\leq 2\sqrt{2\beta(\delta_T)} \sum_{t=d^2+1}^{T} \sqrt{\boldsymbol{w}^\top D_{t-1}^{-1}(\lambda\Lambda_{\bar{\Sigma}_t} D_{t-1} + \sum_{s=1}^{t-1} \bar{\Sigma}_{s,\boldsymbol{w}_s})D_{t-1}^{-1}\boldsymbol{w}} + 2\rho \sum_{t=d^2+1}^{T} \boldsymbol{w}^\top G_t \boldsymbol{w}$$

$$\leq 2\sqrt{2\beta(\delta_T)} \sqrt{T \cdot \sum_{t=d^2+1}^{T} \left( \boldsymbol{w}^\top D_{t-1}^{-1}(\lambda\Lambda_{\bar{\Sigma}_t} D_{t-1} + \sum_{s=1}^{t-1} \bar{\Sigma}_{s,\boldsymbol{w}_s})D_{t-1}^{-1}\boldsymbol{w} \right)}$$

$$+ 2\rho \sum_{t=d^2+1}^{T} \boldsymbol{w}^\top G_t \boldsymbol{w}$$

$$\leq 2\sqrt{2\beta(\delta_T)}\sqrt{T} \sqrt{\lambda \underbrace{\sum_{t=d^2+1}^{T} \left( \boldsymbol{w}^\top D_{t-1}^{-1}\Lambda_{\bar{\Sigma}_t} \boldsymbol{w} \right)}_{\Gamma_1} + \underbrace{\sum_{t=d^2+1}^{T} \left( \boldsymbol{w}^\top D_{t-1}^{-1} \sum_{s=1}^{t-1} \bar{\Sigma}_{s,\boldsymbol{w}_s} D_{t-1}^{-1}\boldsymbol{w} \right)}_{\Gamma_2}}$$

$$+ 2\rho \underbrace{\sum_{t=d^2+1}^{T} \boldsymbol{w}^\top G_t \boldsymbol{w}}_{\Gamma_3}$$

Let $\Sigma_{ij}^+ = \Sigma_{ij} \vee 0$ for any $i, j, \in [d]$. We first address $\Gamma_3$. Since for any $t \geq d^2 + 1$ and $i, j \in [d]$,

$$g_{ij}(t) = 16 \left( \frac{3\ln t}{N_{ij}(t-1)} \vee \sqrt{\frac{3\ln t}{N_{ij}(t-1)}} \right) + \sqrt{\frac{48\ln^2 t}{N_{ij}(t-1)N_i(t-1)}} + \sqrt{\frac{36\ln^2 t}{N_{ij}(t-1)N_j(t-1)}}$$

$$\leq 48 \frac{\ln t}{\sqrt{N_{ij}(t-1)}} + \sqrt{\frac{48\ln^2 t}{N_{ij}(t-1)}} + \sqrt{\frac{36\ln^2 t}{N_{ij}(t-1)}}$$

$$\leq 61 \frac{\ln t}{\sqrt{N_{ij}(t-1)}},$$

we can bound $\Gamma_3$ as follows

$$\Gamma_3 = \sum_{t=d^2+1}^{T} \boldsymbol{w}^\top G_t \boldsymbol{w}$$

$$= \sum_{t=d^2+1}^{T} \sum_{i,j\in[d]} g_{ij}(t) w_{t,i} w_{t,j}$$

$$\leq 61 \sum_{i,j\in[d]} \sum_{t=d^2+1}^{T} \frac{\ln t}{\sqrt{N_{ij}(t-1)}} w_{t,i} w_{t,j}$$

$$\leq 61 \ln T \sum_{i,j\in[d]} \sum_{t=d^2+1}^{T} \frac{w_{t,i} w_{t,j}}{\sqrt{\sum_{s=1}^{t-1} w_{s,i} w_{s,j}}}$$

$$\leq 61 \ln T \sum_{i,j\in[d]} \sum_{t=d^2+1}^{T} \frac{w_{t,i} w_{t,j}}{\sqrt{\sum_{s=1}^{t-1} w_{s,i} w_{s,j}}}$$

$$\leq 122 \ln T \sum_{i,j \in [d]} \sqrt{\sum_{t=1}^{T} w_{t,i} w_{t,j}}$$

$$\leq 122 \ln T \sqrt{d^2 \sum_{i,j \in [d]} \sum_{t=1}^{T} w_{t,i} w_{t,j}}$$

$$= 122 \ln T \sqrt{d^2 \sum_{t=1}^{T} \sum_{i,j \in [d]} w_{t,i} w_{t,j}}$$

$$= 122 \ln T \sqrt{d^2 \sum_{t=1}^{T} \left( \sum_{i \in [d]} w_{t,i} \right)^2}$$

$$\leq 122 d \ln T \sqrt{T}$$

Next, we obtain a bound for $\Gamma_1$.

$$\Gamma_1 = \sum_{t=d^2+1}^{T} \left( \boldsymbol{w}^{\top} D_{t-1}^{-1} \Lambda_{\bar{\Sigma}_t} \boldsymbol{w} \right)$$

$$= \sum_{t=d^2+1}^{T} \sum_{i \in [d]} \frac{\bar{\Sigma}_{t,ii}}{N_i(t-1)} w_{t,i}^2$$

$$\leq \sum_{t=d^2+1}^{T} \sum_{i \in [d]} \frac{\bar{\Sigma}_{t,ii}}{\sum_{s=1}^{t-1} w_{s,i}} w_{t,i}^2$$

$$\leq \sum_{i \in [d]} \sum_{t=d^2+1}^{T} \frac{\Sigma_{ii} + 2g_{ii}(t)}{\sum_{s=1}^{t-1} w_{s,i}} w_{t,i}^2$$

$$\leq \sum_{i \in [d]} \left( \sum_{t=d^2+1}^{T} \frac{\Sigma_{ii}}{\sum_{s=1}^{t-1} w_{s,i}} w_{t,i} + 122 \sum_{t=d^2+1}^{T} \frac{\frac{\ln t}{\sqrt{N_i(t-1)}}}{\sum_{s=1}^{t-1} w_{s,i}} w_{t,i} \right)$$

$$\leq \sum_{i \in [d]} \left( \Sigma_{,ii} \sum_{t=d^2+1}^{T} \frac{1}{\sum_{s=1}^{t-1} w_{s,i}} w_{t,i} + 122 \ln T \sum_{t=d^2+1}^{T} \frac{1}{\left( \sum_{s=1}^{t-1} w_{s,i} \right)^{\frac{3}{2}}} w_{t,i} \right)$$

$$\leq \sum_{i \in [d]} \left( \Sigma_{ii}^+ \ln \left( \sum_{t=1}^{T} w_{t,i} \right) + 244 \ln T \frac{1}{\sqrt{\sum_{t=1}^{d^2} w_{t,i}}} \right)$$

$$\leq \sum_{i \in [d]} \left( \Sigma_{ii}^+ \ln T + 244 \ln T \frac{1}{\sqrt{\sum_{t=1}^{d^2} w_{t,i}}} \right)$$

Finally, we bound $\Gamma_2$.

$$\Gamma_2$$

$$= \sum_{t=d^2+1}^{T} \left( \boldsymbol{w}_t^\top D_{t-1}^{-1} \left( \sum_{s=1}^{t-1} \bar{\Sigma}_{s,\boldsymbol{w}_s} \right) D_{t-1}^{-1} \boldsymbol{w}_t \right)$$

$$= \sum_{i,j\in[d]} \sum_{t=d^2+1}^{T} \frac{\sum_{s=1}^{t-1} \bar{\Sigma}_{s,ij} \mathbb{I}\{w_{s,i}, w_{s,j} > 0\}}{N_i(t-1)N_j(t-1)} w_{t,i} w_{t,j}$$

$$\leq \sum_{i,j\in[d]} \sum_{t=d^2+1}^{T} \frac{\sum_{s=1}^{t-1} \bar{\Sigma}_{s,ij} \mathbb{I}\{w_{s,i}, w_{s,j} > 0\}}{N_{ij}^2(t-1)} w_{t,i} w_{t,j}$$

$$\leq \sum_{t=d^2+1}^{T} \sum_{i,j\in[d]} \frac{\sum_{s=1}^{t-1} (\Sigma_{ij} + 2g_{ij}(s)) \mathbb{I}\{w_{s,i}, w_{s,j} > 0\}}{N_{ij}^2(t-1)} w_{t,i} w_{t,j}$$

$$= \sum_{i,j\in[d]} \left( \sum_{t=d^2+1}^{T} \frac{\Sigma_{ij} \sum_{s=1}^{t-1} \mathbb{I}\{w_{s,i}, w_{s,j} > 0\}}{N_{ij}^2(t-1)} w_{t,i} w_{t,j} + \sum_{t=d^2+1}^{T} \frac{2\sum_{s=1}^{t-1} g_{ij}(s) \mathbb{I}\{w_{s,i}, w_{s,j} > 0\}}{N_{ij}^2(t-1)} w_{t,i} w_{t,j} \right)$$

$$\leq \sum_{i,j\in[d]} \left( \Sigma_{ij} \sum_{t=d^2+1}^{T} \frac{1}{N_{ij}(t-1)} w_{t,i} w_{t,j} + 122 \sum_{t=d^2+1}^{T} \frac{\sum_{s=1}^{t-1} \frac{\ln s}{\sqrt{N_{ij}(s-1)}} \mathbb{I}\{w_{s,i}, w_{s,j} > 0\}}{N_{ij}^2(t-1)} w_{t,i} w_{t,j} \right)$$

$$\leq \sum_{i,j\in[d]} \left( \Sigma_{ij} \sum_{t=d^2+1}^{T} \frac{1}{\sum_{s=1}^{t-1} w_{s,i} w_{s,j}} w_{t,i} w_{t,j} + 122 \sum_{t=d^2+1}^{T} \frac{\ln t \sum_{s=1}^{t-1} \frac{1}{\sqrt{\sum_{\ell=1}^{s-1} \mathbb{I}\{w_{\ell,i}, w_{\ell,j} > 0\}}} \mathbb{I}\{w_{s,i}, w_{s,j} > 0\}}{N_{ij}^2(t-1)} w_{t,i} w_{t,j} \right)$$

$$\leq \sum_{i,j\in[d]} \left( \Sigma_{ij} \sum_{t=d^2+1}^{T} \frac{1}{\sum_{s=1}^{t-1} w_{s,i} w_{s,j}} w_{t,i} w_{t,j} + 244 \sum_{t=d^2+1}^{T} \frac{\ln t \sqrt{\sum_{s=1}^{t-1} \mathbb{I}\{w_{s,i}, w_{s,j} > 0\}}}{N_{ij}^2(t-1)} w_{t,i} w_{t,j} \right)$$

$$= \sum_{i,j\in[d]} \left( \Sigma_{ij} \sum_{t=d^2+1}^{T} \frac{1}{\sum_{s=1}^{t-1} w_{s,i} w_{s,j}} w_{t,i} w_{t,j} + 244 \ln T \sum_{t=d^2+1}^{T} \frac{1}{N_{ij}^{\frac{3}{2}}(t-1)} w_{t,i} w_{t,j} \right)$$

$$\leq \sum_{i,j\in[d]} \left( \Sigma_{ij} \sum_{t=d^2+1}^{T} \frac{1}{\sum_{s=1}^{t-1} w_{s,i} w_{s,j}} w_{t,i} w_{t,j} + 244 \ln T \sum_{t=d^2+1}^{T} \frac{1}{\left( \sum_{s=1}^{t-1} w_{s,i} w_{s,j} \right)^{\frac{3}{2}}} w_{t,i} w_{t,j} \right)$$

$$\leq \sum_{i,j\in[d]} \left( \Sigma_{ij}^{+} \ln \left( \sum_{t=1}^{T} w_{t,i} w_{t,j} \right) + 488 \ln T \frac{1}{\sqrt{\sum_{t=1}^{d^2} w_{t,i} w_{t,j}}} \right)$$

$$\leq \sum_{i,j\in[d]} \left( \Sigma_{ij}^{+} \ln T + 488 \ln T \frac{1}{\sqrt{\sum_{t=1}^{d^2} w_{t,i} w_{t,j}}} \right)$$

Recall that $\beta(\delta_T) = \ln(T \ln^2 T) + d \ln \ln T + \frac{d}{2} \ln(1 + e/\lambda) = O(\ln T + d \ln \ln T + d \ln(1 + \lambda^{-1}))$. Combining the bounds of $\Gamma_1, \Gamma_2$ and $\Gamma_3$, we obtain

$$\sum_{t=d^2+1}^{T} (f(\boldsymbol{w}^*) - f(\boldsymbol{w}_t))$$

$$\leq 2\sqrt{2\beta(\delta_T)}\sqrt{T}\sqrt{\underbrace{\lambda\sum_{t=d^2+1}^{T}\left(\boldsymbol{w}^\top D_{t-1}^{-1}\Lambda_{\bar{\Sigma}_t}\boldsymbol{w}\right)}_{\Gamma_1} + \underbrace{\sum_{t=d^2+1}^{T}\left(\boldsymbol{w}^\top D_{t-1}^{-1}\sum_{s=1}^{t-1}\bar{\Sigma}_{s,\boldsymbol{w}_s}D_{t-1}^{-1}\boldsymbol{w}\right)}_{\Gamma_2} + 2\rho\underbrace{\sum_{t=d^2+1}^{T}\boldsymbol{w}^\top G_t\boldsymbol{w}}_{\Gamma_3}}$$

$$\leq 2\sqrt{2\beta(\delta_T)}\sqrt{T}\sqrt{\lambda\sum_{i\in[d]}\left(\Sigma_{ii}^+\ln T + 244\ln T\frac{1}{\sqrt{\sum_{t=1}^{d^2}w_{t,i}}}\right) + \sum_{i,j\in[d]}\left(\Sigma_{ij}^+\ln T + 488\ln T\frac{1}{\sqrt{\sum_{t=1}^{d^2}w_{t,i}w_{t,j}}}\right)}$$
$$+ 244\rho d\ln T\sqrt{T}$$

$$\leq 2\sqrt{2\beta(\delta_T)}\sqrt{T}\sqrt{\lambda\ln T\sum_{i\in[d]}\Sigma_{ii}^+ + \ln T\sum_{i,j\in[d]}\Sigma_{ij}^+ + (244\lambda+488)d^2\ln T} + 244\rho d\ln T\sqrt{T}$$

$$= O\left(\sqrt{(\ln T + d\ln\ln T + d\ln(1+\lambda^{-1}))\cdot T}\sqrt{\lambda\ln T\sum_{i\in[d]}\Sigma_{ii}^+ + \ln T\sum_{i,j\in[d]}\Sigma_{ij}^+ + (\lambda+1)d^2\ln T} + \rho d\ln T\sqrt{T}\right)$$

$$= O\left(\sqrt{d(\ln T + \ln(1+\lambda^{-1}))\cdot T}\sqrt{\lambda\ln T\sum_{i\in[d]}\Sigma_{ii}^+ + \ln T\sum_{i,j\in[d]}\Sigma_{ij}^+ + (\lambda+1)d^2\ln T} + \rho d\ln T\sqrt{T}\right)$$

$$= O\left(\sqrt{d\ln T\left(\ln T + \ln(1+\lambda^{-1})\right)\cdot T}\sqrt{\lambda\sum_{i\in[d]}\Sigma_{ii}^+ + \sum_{i,j\in[d]}\Sigma_{ij}^+ + (\lambda+1)d^2} + \rho d\ln T\sqrt{T}\right)$$

According to Lemmas 1 and 2, for any $t\geq 2$, the probability of event $\neg(\mathcal{G}_t\cap\mathcal{H}_t)$ satisfies

$$\Pr\left[\neg(\mathcal{G}_t\cap\mathcal{H}_t)\right]\leq\frac{10d^2}{t^2} + \frac{1}{t\ln^2 t}$$
$$\leq\frac{10d^2}{t\ln^2 t} + \frac{1}{t\ln^2 t}$$
$$= \frac{11d^2}{t\ln^2 t}$$

Therefore, for any horizon $T$, we obtain the regret upper bound

$$\mathbb{E}[\mathcal{R}(T)] = O(\Delta_{\max}) + \sum_{t=2}^{T}O\left(\Delta_{\max}\cdot\Pr\left[\neg(\mathcal{G}_t\cap\mathcal{H}_t)\right] + \Delta_t\cdot\mathbb{I}\{\mathcal{G}_t\cap\mathcal{H}_t\}\right)$$

$$= O(\Delta_{\max}) + \sum_{t=2}^{T}O\left(\Delta_{\max}\cdot\frac{d^2}{t\ln^2 t}\right) + O\left(\sqrt{d\ln T\left(\ln T + \ln(1+\lambda^{-1})\right)T}\cdot\right.$$
$$\left.\sqrt{\lambda\sum_{i\in[d]}\Sigma_{ii}^+ + \sum_{i,j\in[d]}\Sigma_{ij}^+ + (\lambda+1)d^2} + \rho d\ln T\sqrt{T}\right)$$

$$= O\left(\sqrt{d\ln T\left(\ln T + \ln(1+\lambda^{-1})\right)T}\sqrt{\lambda\sum_{i\in[d]}\Sigma_{ii}^+ + \sum_{i,j\in[d]}\Sigma_{ij}^+ + (\lambda+1)d^2} + \rho d\ln T\sqrt{T} + d^2\Delta_{\max}\right)$$

$$=O\left(\sqrt{d\left(\ln(1+\lambda^{-1})+1\right)\ln^2 T\cdot T}\sqrt{(\lambda+1)\left(\sum_{i,j\in[d]}\Sigma_{ij}^{+}+d^2\right)}+\rho d\ln T\sqrt{T}\right)$$

$$=O\left(\sqrt{(\lambda+1)\left(\ln(1+\lambda^{-1})+1\right)(\|\Sigma\|_{+}+d^2)d\ln^2 T\cdot T}+\rho d\ln T\sqrt{T}\right)$$

$$=O\left(\sqrt{L(\lambda)(\|\Sigma\|_{+}+d^2)d\ln^2 T\cdot T}+\rho d\ln T\sqrt{T}\right)$$

where $L(\lambda)=(\lambda+1)\left(\ln(1+\lambda^{-1})+1\right)$ and $\|\Sigma\|_{+}=\sum_{i,j\in[d]}(\Sigma_{ij}\vee 0)$ for any $i,j\in[d]$. $\qquad\square$

## C.2   Proof of Theorem 4

*Proof.* First, we construct some instances with $d\geq 4$, $\frac{2}{d}\leq c\leq\frac{1}{2}$, $\Sigma_*=I$ and $\boldsymbol{\theta}_t\sim N(\boldsymbol{\theta}^*,I)$.

Let $I_J$ be a random instance constructed as follows: we uniformly choose a dimension $J$ from $[d]$, and the expected reward vector $\boldsymbol{\theta}_J^*$ has $\frac{1}{2}+\varepsilon$ on its $J$-th entry and $\frac{1}{2}$ elsewhere, where $\varepsilon\in(0,\frac{1}{2}]$ will be specified later. Let $I_u$ be a uniform instance, where $\boldsymbol{\theta}_u^*$ has all its entries to be $\frac{1}{2}$. Let $\Pr_J[\cdot]$ and $\Pr_u[\cdot]$ denote the probabilities under instances $I_J$ and $I_u$, respectively, and let $\Pr_j[\cdot]=\Pr_J[\cdot|J=j]$. Analogously, $E_J[\cdot]$, $E_u[\cdot]$ and $E_j[\cdot]=E_J[\cdot|J=j]$ denote the expectation operations.

Fix an algorithm $\mathcal{A}$. Let $S_t\in\{\mathbb{R}\cup\{\bot\}\}^d$ be a random variable vector denoting the observations at timestep $t$, obtained by running $\mathcal{A}$. Here $\bot$ denotes no observation on this dimension. Let $Q_{\bot}$ denote the distribution on support $\{\bot\}$ which takes value $\bot$ with probability 1.

In CMCB-SB, if $w_{t,i}>0$, we can observe the reward on the $i$-th dimension, i.e., $S_{t,i}=\theta_{t,i}$; otherwise, if $w_{t,i}=0$, we cannot get observation on the $i$-th dimension, i.e., $S_{t,i}=\bot$. Let $D_J$ be the distribution of observation sequence $S_1,\ldots S_t$ under instance $I_J$, and $D_j=D_{J|J=j}$ is the distribution conditioned on $J=j$. Let $D_u$ be the distribution of observation sequence $S_1,\ldots S_t$ under instance $I_u$. For any $i\in[d]$, let $N_i=\sum_{t=1}^T\mathbb{I}\{w_{t,i}>0\}$ be the number of pulls that has a positive weight on the $i$-th dimension, i.e., the number of observations on the $i$-th dimension.

Following the analysis procedure of Lemma A.1 in [4], we have

$$
\begin{aligned}
KL(D_j\|D_u) &=\sum_{t=1}^T KL(D_u[S_t|S_1,\ldots,S_{t-1}]\|D_j[S_t|S_1,\ldots,S_{t-1}])\\
&=\sum_{t=1}^T\sum_{i=1}^d\left(\Pr[w_{t,i}>0]\cdot KL\left(N(\theta_{u,i}^*,1)\|N(\theta_{j,i}^*,1)\right)+\Pr[w_{t,i}=0]\cdot KL(Q_{\bot}\|Q_{\bot})\right)\\
&=\sum_{t=1}^T\left(\Pr[w_{t,j}>0]\cdot KL\left(N(\frac{1}{2},1)\|N(\frac{1}{2}+\varepsilon,1)\right)+\sum_{i\neq j}^d\Pr[w_{t,i}>0]\cdot KL\left(N(\frac{1}{2},1)\|N(\frac{1}{2},1)\right)\right)\\
&=\frac{1}{2}\varepsilon^2\cdot\sum_{t=1}^T\Pr[w_{t,j}>0]\\
&=\frac{1}{2}\varepsilon^2 E_u[N_j]
\end{aligned}
$$

Here the first equality comes from the chain rule of entropy [9]. The second equality is due to that given $S_1, \ldots, S_{t-1}$, if $w_{t,i} > 0$, the conditional distribution of $S_t$ is $N(\theta^*_{\cdot,i}, 1)$, where "$\cdot$" refers to the subscript of instances; otherwise, if $w_{t,i} = 0$, $S_t$ is $\perp$ deterministically. The third equality is due to that $\theta^*_u$ and $\theta^*_j$ only have one different entry on the $j$-th dimension.

Let $\| \cdot \|$ with subscript $TV$ denote the total variance distance, and $KL(\cdot\|\cdot)$ denote the Kullback–Leibler divergence. Using Eq. (28) in the analysis of Lemma A.1 in [4] and Pinsker's inequality, we have

$$
\begin{aligned}
E_j[N_j] &\leq E_u[N_j] + T\|D_j - D_u\|_{TV} \\
&\leq E_u[N_j] + T\sqrt{\frac{1}{2}KL(D_j\|D_u)} \\
&= E_u[N_j] + \frac{T\varepsilon}{2}\sqrt{E_u[N_j]}
\end{aligned}
$$

Let $m = \lfloor \frac{1}{c} \rfloor \leq \frac{d}{2}$ denote the maximum number of positive entries for a feasible action, i.e., the maximum number of observations for a pull. Performing the above argument for all $j \in [d]$ and using $\sum_{j\in[d]} E_u[N_j] \leq mT$, we have

$$
\begin{aligned}
\sum_{j\in[d]} E_j[N_j] &\leq \sum_{j\in[d]} E_u[N_j] + \frac{T\varepsilon}{2}\sum_{j\in[d]}\sqrt{E_u[N_j]} \\
&\leq mT + \frac{T\varepsilon}{2}\sqrt{d\sum_{j\in[d]} E_u[N_j]} \\
&\leq mT + \frac{T\varepsilon}{2}\sqrt{dmT}
\end{aligned}
$$

and thus

$$
E_J[N_J] = \frac{1}{d}\sum_{j\in[d]} E_j[N_j] \leq \frac{mT}{d} + \frac{T\varepsilon}{2}\sqrt{\frac{mT}{d}}
$$

Letting $\rho \leq \frac{\varepsilon}{2(1-c)}$, the expected reward (linear) term dominates $f(w)$, and the best action $w^*$ under $I_J$ has the weight 1 on the $J$-th entry and 0 elsewhere.

Recall that $m \leq \frac{1}{c}$. For each pull that has no weight on the $J$-th entry, algorithm $\mathcal{A}$ must suffer a regret at least

$$
\begin{aligned}
(\frac{1}{2} + \varepsilon - \rho) &- (\frac{1}{2} - \rho \cdot \frac{1}{m}) \\
&\geq \varepsilon - \frac{m-1}{m}\rho \\
&\geq \varepsilon - \frac{m-1}{m}\cdot\frac{\varepsilon}{2(1-c)} \\
&\geq \varepsilon - \frac{m-1}{m}\cdot\frac{\varepsilon}{2(1-\frac{1}{m})} \\
&= \frac{\varepsilon}{2}
\end{aligned}
$$

Thus, the regret is lower bounded by

$$
\begin{aligned}
E[R(T)] \geq & (T - E_J[N_J]) \cdot \frac{\varepsilon}{2} \\
\geq & \left( T - \frac{mT}{d} - \frac{T\varepsilon}{2}\sqrt{\frac{mT}{d}} \right) \cdot \frac{\varepsilon}{2} \\
= & \Omega\left( T\varepsilon - T\varepsilon^2\sqrt{\frac{mT}{d}} \right),
\end{aligned}
$$

where the last equality is due to $m \leq \frac{d}{2}$.

Letting $\varepsilon = a_0\sqrt{\frac{d}{Tm}}$ for small enough constant $a_0$, we obtain the regret lower bound $\Omega(\sqrt{\frac{dT}{m}}) = \Omega(\sqrt{cdT})$.

$\square$

## D  Proof for CMCB-FB

### D.1  Proof of Theorem 5

In order to prove Theorem 5, we first prove Lemmas 5 and 6, which give the concentrations of covariance and means for CMCB-FB, using different techniques than those for CMCB-SB (Lemmas 1 and 2) and CMCB-FI (Lemmas 3 and 4).

**Lemma 5** (Concentration of Covariance for CMCB-FB). *Consider the CMCB-FB problem and algorithm* MC-ETE *(Algorithm 3). For any $t > 0$, the event*

$$
\mathcal{M}_t \triangleq \left\{ |\Sigma_{ij} - \hat{\Sigma}_{ij,t-1}| \leq 5\|C_\pi^+\|\sqrt{\frac{3\ln t}{2N_\pi(t)}} \right\}
$$

*satisfies*

$$
Pr[\mathcal{M}_t] \geq 1 - \frac{6d^2}{t^2},
$$

*where $\|C_\pi^+\| \triangleq \max_{i\in[\tilde{d}]}\left\{ \sum_{j\in[\tilde{d}]} |C_{\pi,ij}^+| \right\}$.*

*Proof.* Let $\boldsymbol{\sigma} = (\Sigma_{11}, \ldots, \Sigma_{dd}, \Sigma_{12}, \ldots, \Sigma_{1d}, \Sigma_{23}, \ldots, \Sigma_{d,d-1})^\top \in \mathbb{R}^{\tilde{d}}$ denote the column vector that stacks the $\tilde{d}$ distinct entries in the covariance matrix $\Sigma$.

Recall that the $\tilde{d} \times \tilde{d}$ matrix

$$
C_\pi = \begin{bmatrix}
w_{1,1}^2 & \cdots & w_{1,d}^2 & 2w_{1,1}w_{1,2} & 2w_{1,1}w_{1,3} & \cdots & 2w_{1,d-1}w_{1,d} \\
w_{2,1}^2 & \cdots & w_{2,d}^2 & 2w_{2,1}w_{2,2} & 2w_{2,1}w_{2,3} & \cdots & 2w_{2,d-1}w_{2,d} \\
& \cdots & & & & \cdots & \\
& & & & & & \\
& \cdots & & & & \cdots & \\
w_{\tilde{d},1}^2 & \cdots & w_{\tilde{d},d}^2 & 2w_{\tilde{d},1}w_{\tilde{d},2} & 2w_{\tilde{d},1}w_{\tilde{d},3} & \cdots & 2w_{\tilde{d},d-1}w_{\tilde{d},d}
\end{bmatrix},
$$

where $w_{i,j}$ denotes the $j$-th entry of portfolio vector $\boldsymbol{w}_i$ in design set $W$. We use $C_{\pi,k}$ to denote the $k$-th row in matrix $C_\pi$.

We recall the feedback structure in algorithm MC-ETE as follows. At each timestep $t$, the learner plays an action $\boldsymbol{w}_t \in \triangle_d$, and observes the full-bandit feedback $y_t = \boldsymbol{w}_t^\top \boldsymbol{\theta}_t$ with $\mathbb{E}[y_t] = \boldsymbol{w}_t^\top \boldsymbol{\theta}^*$ and $\text{Var}[y_t] = \boldsymbol{w}_t^\top \Sigma \boldsymbol{w}_t = \sum_{i\in[d]} w_{t,i}^2 \Sigma_{ii} +$

$\sum_{i,j\in[d],i<j} 2w_{t,i}w_{t,j}\Sigma_{ij}$. Then, during exploration round $s$, where each action in design set $\pi = \{\boldsymbol{v}_1,\ldots,\boldsymbol{v}_{\tilde{d}}\}$ is pulled once, the full-bandit feedback $\boldsymbol{y}_s$ has mean $\boldsymbol{y}(\pi) \triangleq (\boldsymbol{v}_1^\top\boldsymbol{\theta}^*,\ldots,\boldsymbol{v}_{\tilde{d}}^\top\boldsymbol{\theta}^*)^\top$ and variance $\boldsymbol{z}(\pi) \triangleq (\boldsymbol{v}_1^\top\Sigma\boldsymbol{v}_1,\ldots,\boldsymbol{v}_{\tilde{d}}^\top\Sigma\boldsymbol{v}_{\tilde{d}})^\top = (C_{\pi,1}^\top\boldsymbol{\sigma},\ldots,C_{\pi,\tilde{d}}^\top\boldsymbol{\sigma})^\top = C_\pi\boldsymbol{\sigma}$. For any $t > 0$, denote $\hat{\boldsymbol{y}}_t$ the empirical mean of $\boldsymbol{y}(\pi) \triangleq (\boldsymbol{v}_1^\top\boldsymbol{\theta}^*,\ldots,\boldsymbol{v}_{\tilde{d}}^\top\boldsymbol{\theta}^*)^\top$ and $\hat{\boldsymbol{z}}_t$ the empirical variance of $\boldsymbol{z}(\pi) \triangleq (C_{\pi,1}^\top\boldsymbol{\sigma},\ldots,C_{\pi,\tilde{d}}^\top\boldsymbol{\sigma})^\top$.

Using the Chernoff-Hoeffding inequality for empirical variances (Lemma 1 in [28]), we have that for any $t > 0$ and $k \in \tilde{d}$, with probability at least $1 - \frac{6d^2}{t^2}$,

$$|\hat{z}_{t-1,k} - z_k| = \left|\hat{z}_{t-1,k} - C_{\pi,k}^\top\boldsymbol{\sigma}\right| \le 5\sqrt{\frac{3\ln t}{2N_\pi(t-1)}}.$$

Let $C_{\pi,i}^+$ denote the $i$-th row of matrix $C_\pi^+$ and $C_{\pi,ik}^+$ denote the $ik$-th entry of matrix $C_\pi^+$. Since $C_\pi^+ C_\pi = I$, for any $i \in [\tilde{d}]$, we have

$$
\begin{aligned}
|\hat{\sigma}_{t-1,i} - \sigma_i| &= \left|(C_{\pi,i}^+)^\top\hat{\boldsymbol{z}}_t - (C_{\pi,i}^+)^\top C_\pi\boldsymbol{\sigma}\right| \\
&\le \sum_{k\in[\tilde{d}]}\left|C_{\pi,ik}^+\right|\left|\hat{z}_{t-1,k} - C_{\pi,k}^\top\boldsymbol{\sigma}\right| \\
&\le 5\sum_{k\in[\tilde{d}]}\left|C_{\pi,ik}^+\right|\sqrt{\frac{3\ln t}{2N_\pi(t)}} \\
&\le 5\|C_\pi^+\|\sqrt{\frac{3\ln t}{2N_\pi(t)}}
\end{aligned}
$$

In addition, since $\boldsymbol{\sigma} = (\Sigma_{11},\ldots,\Sigma_{dd},\Sigma_{12},\ldots,\Sigma_{1d},\Sigma_{23},\ldots,\Sigma_{d,d-1})^\top \in \mathbb{R}^{\tilde{d}}$ is the column vector stacking the distinct entries in $\Sigma$, we obtain the lemma. $\qquad\square$

**Lemma 6** (Concentration of Means for CMCB-FB). *Consider the CMCB-FB problem and algorithm* MC-ETE *(Algorithm 3). Let $\delta_t > 0$, $\lambda > 0$, $\beta(\delta_t) = \ln(1/\delta_t) + \ln\ln t + \frac{\tilde{d}}{2}\ln(1 + e/\lambda)$ and $E_t(\boldsymbol{w}) = \sqrt{2\beta(\delta_t)}\sqrt{\boldsymbol{w}^\top B_\pi^+ D_{t-1}^{-1}(\lambda\Lambda_{\Sigma_\pi} D_{t-1} + \sum_{s=1}^{N_\pi(t-1)}\Sigma_\pi)D_{t-1}^{-1}(B_\pi^+)\top\boldsymbol{w}}$, where $\Sigma_\pi = \mathrm{diag}(\boldsymbol{v}_1^\top\Sigma\boldsymbol{v}_1,\ldots,\boldsymbol{v}_{\tilde{d}}^\top\Sigma\boldsymbol{v}_{\tilde{d}})$. Then, the event $\mathcal{N}_t \triangleq \{|\boldsymbol{w}^\top\boldsymbol{\theta}^* - \boldsymbol{w}^\top\hat{\boldsymbol{\theta}}_{t-1}| \le E_t(\boldsymbol{w}), \forall\boldsymbol{w}\in\mathcal{D}\}$ satisfies $\Pr[\mathcal{N}_t] \ge 1 - \delta_t$.*

*Proof.* Recall that the $\tilde{d}\times d$ matrix $B_\pi = [\boldsymbol{v}_1^\top;\ldots;\boldsymbol{v}_{\tilde{d}}^\top]$. and $B_\pi^+$ is the Moore–Penrose pseudoinverse of $B_\pi$. Since $B_\pi$ is of full column rank, $B_\pi^+$ satisfies $B_\pi^+ B_\pi = I$.

We recall the feedback structure in algorithm MC-ETE as follows. At each timestep $t$, the learner plays an action $\boldsymbol{w}_t \in \triangle_d$, and observes the full-bandit feedback $y_t = \boldsymbol{w}_t^\top\boldsymbol{\theta}_t$ such that $\mathbb{E}[y_t] = \boldsymbol{w}_t^\top\boldsymbol{\theta}^*$ and $\mathrm{Var}[y_t] = \boldsymbol{w}_t^\top\Sigma\boldsymbol{w}_t = \sum_{i\in[d]}w_{t,i}^2\Sigma_{ii} + \sum_{i,j\in[d],i<j} 2w_{t,i}w_{t,j}\Sigma_{ij}$. Then, during exploration round $s$, where each action in design set $\pi = \{\boldsymbol{v}_1,\ldots,\boldsymbol{v}_{\tilde{d}}\}$ is pulled once, the full-bandit feedback $\boldsymbol{y}_s$ has mean $\boldsymbol{y}(\pi) \triangleq (\boldsymbol{v}_1^\top\boldsymbol{\theta}^*,\ldots,\boldsymbol{v}_{\tilde{d}}^\top\boldsymbol{\theta}^*)^\top$ and variance $\boldsymbol{z}(\pi) \triangleq (\boldsymbol{v}_1^\top\Sigma\boldsymbol{v}_1,\ldots,\boldsymbol{v}_{\tilde{d}}^\top\Sigma\boldsymbol{v}_{\tilde{d}})^\top = (C_{\pi,1}^\top\boldsymbol{\sigma},\ldots,C_{\pi,\tilde{d}}^\top\boldsymbol{\sigma})^\top = C_\pi\boldsymbol{\sigma}$. For any $t > 0$, $\hat{\boldsymbol{y}}_t$ is the empirical mean of $\boldsymbol{y}(\pi) \triangleq (\boldsymbol{v}_1^\top\boldsymbol{\theta}^*,\ldots,\boldsymbol{v}_{\tilde{d}}^\top\boldsymbol{\theta}^*)^\top$ and $\hat{\boldsymbol{z}}_t$ is the empirical variance of $\boldsymbol{z}(\pi) \triangleq (C_{\pi,1}^\top\boldsymbol{\sigma},\ldots,C_{\pi,\tilde{d}}^\top\boldsymbol{\sigma})^\top$.

Let $D_t$ be the $\tilde{d}\times\tilde{d}$ diagonal matrix such that $D_{t,ii} = N_\pi(t)$ for any $i \in [\tilde{d}]$ and $t > 0$. Let $\Sigma_\pi$ be a $\tilde{d}\times\tilde{d}$ diagonal matrix such that $\Sigma_{\pi,ii} = \boldsymbol{w}_i^\top\Sigma\boldsymbol{w}_i$ for any $i \in [\tilde{d}]$, and thus $\Lambda_{\Sigma_\pi} = \Sigma_\pi$. Let $\boldsymbol{\varepsilon}_t$ be the vector such that $\boldsymbol{\eta}_t = \Sigma^{\frac{1}{2}}\boldsymbol{\varepsilon}_t$ for any timestep $t > 0$. Let $\zeta_s = (\boldsymbol{v}_1\top\Sigma^{\frac{1}{2}}\boldsymbol{\varepsilon}_{s,1},\ldots,\boldsymbol{v}_{\tilde{d}}\top\Sigma^{\frac{1}{2}}\boldsymbol{\varepsilon}_{s,\tilde{d}})^\top$, where $\Sigma^{\frac{1}{2}}\boldsymbol{\varepsilon}_{s,k}$ denotes the noise of the $k$-th sample in the $s$-th exploration round.

Note that, the following analysis of $|\boldsymbol{w}^\top(\boldsymbol{\theta}^* - \hat{\boldsymbol{\theta}}_{t-1})|$ and the constructions (definitions) of the noise $\zeta_s$, matrices $S_t, V_t$ and super-martingale $M_t^{\boldsymbol{u}}$ are different from those in CMCB-SB (Lemmas 1 and 2) and CMCB-FI (Lemmas 3 and 4).

For any $\boldsymbol{w} \in \triangle_d$, we have

$$
\begin{aligned}
\left|\boldsymbol{w}^\top\left(\boldsymbol{\theta}^* - \hat{\boldsymbol{\theta}}_{t-1}\right)\right| &= \left|\boldsymbol{w}^\top\left(B_\pi^+ B_\pi \boldsymbol{\theta}^* - B_\pi^+ \hat{\boldsymbol{y}}_{t-1}\right)\right| \\
&= \left|\boldsymbol{w}^\top B_\pi^+ \left(\boldsymbol{y}(\pi) - \hat{\boldsymbol{y}}_{t-1}\right)\right| \\
&= \left|-\boldsymbol{w}^\top B_\pi^+ D_{t-1}^{-1} \sum_{s=1}^{N_\pi(t-1)} \zeta_s\right| \\
&= \left|-\boldsymbol{w}^\top B_\pi^+ D_{t-1}^{-1} \left(D + \sum_{s=1}^{N_\pi(t-1)} \Sigma_\pi\right)^{\frac{1}{2}} \left(D + \sum_{s=1}^{N_\pi(t-1)} \Sigma_\pi\right)^{-\frac{1}{2}} \sum_{s=1}^{N_\pi(t-1)} \zeta_s\right| \\
&\leq \sqrt{\boldsymbol{w}^\top B_\pi^+ D_{t-1}^{-1} \left(D + \sum_{s=1}^{N_\pi(t-1)} \Sigma_\pi\right) D_{t-1}^{-1}(B_\pi^+)^\top \boldsymbol{w}} \cdot \left\|\sum_{s=1}^{N_\pi(t-1)} \zeta_s\right\|_{\left(D + \sum_{s=1}^{N_\pi(t-1)} \Sigma_\pi\right)^{-1}}
\end{aligned}
$$

Let $S_t = \sum_{s=1}^{N_\pi(t-1)} \zeta_s$, $V_t = \sum_{s=1}^{N_\pi(t-1)} \Sigma_\pi$ and $I_{D+V_t} = \frac{1}{2}\|S_t\|_{(D+V_t)^{-1}}^2$. Then, we have

$$
\left\|\sum_{s=1}^{N_\pi(t-1)} \zeta_s\right\|_{\left(D + \sum_{s=1}^{N_\pi(t-1)} \Sigma_\pi\right)^{-1}} = \|S_t\|_{(D+V_t)^{-1}} = \sqrt{2I_{D+V_t}}.
$$

Since $D \preceq \lambda\Lambda_{\Sigma_\pi} D_{t-1}$, we get

$$
\begin{aligned}
\left|\boldsymbol{w}^\top\left(\boldsymbol{\theta}^* - \hat{\boldsymbol{\theta}}_{t-1}\right)\right| &\leq \sqrt{\boldsymbol{w}^\top B_\pi^+ D_{t-1}^{-1} D D_{t-1}^{-1}(B_\pi^+)^\top \boldsymbol{w} + \boldsymbol{w}^\top B_\pi^+ D_{t-1}^{-1} \left(\sum_{s=1}^{N_\pi(t-1)} \Sigma_\pi\right) D_{t-1}^{-1}(B_\pi^+)^\top \boldsymbol{w}} \cdot \sqrt{2I_{D+V_t}} \\
&\leq \sqrt{\lambda\boldsymbol{w}^\top B_\pi^+ D_{t-1}^{-1} \Lambda_\Sigma (B_\pi^+)^\top \boldsymbol{w} + \boldsymbol{w}^\top B_\pi^+ D_{t-1}^{-1} \left(\sum_{s=1}^{N_\pi(t-1)} \Sigma_\pi\right) D_{t-1}^{-1}(B_\pi^+)^\top \boldsymbol{w}} \cdot \sqrt{2I_{D+V_t}} \\
&= \sqrt{\boldsymbol{w}^\top B_\pi^+ D_{t-1}^{-1} \left(\lambda\Lambda_\Sigma D_{t-1} + \sum_{s=1}^{N_\pi(t-1)} \Sigma_\pi\right) D_{t-1}^{-1}(B_\pi^+)^\top \boldsymbol{w}} \cdot \sqrt{2I_{D+V_t}}
\end{aligned}
$$

Thus,

$$
\begin{aligned}
&\Pr\left[\left|\boldsymbol{w}^\top\left(\boldsymbol{\theta}^* - \hat{\boldsymbol{\theta}}_{t-1}\right)\right| > \sqrt{2\beta(\delta_t)}\sqrt{\boldsymbol{w}^\top B_\pi^+ D_{t-1}^{-1}(\lambda\Lambda_{\Sigma_\pi} D_{t-1} + \sum_{s=1}^{N_\pi(t-1)} \Sigma_\pi)D_{t-1}^{-1}(B_\pi^+)^\top \boldsymbol{w}}\right] \\
&\leq \Pr\left[\sqrt{\boldsymbol{w}^\top B_\pi^+ D_{t-1}^{-1}\left(\lambda\Lambda_\Sigma D_{t-1} + \sum_{s=1}^{N_\pi(t-1)} \Sigma_\pi\right)D_{t-1}^{-1}(B_\pi^+)^\top \boldsymbol{w}} \cdot \sqrt{2I_{D+V_t}}\right]
\end{aligned}
$$

$$> \sqrt{2\beta(\delta_t)} \sqrt{\boldsymbol{w}^\top B_\pi^+ D_{t-1}^{-1} \left( \lambda \Lambda_{\Sigma_\pi} D_{t-1} + \sum_{s=1}^{N_\pi(t-1)} \Sigma_\pi \right) D_{t-1}^{-1} (B_\pi^+)^\top \boldsymbol{w}}$$

$$= \Pr\left[ I_{D+V_t} > \beta(\delta_t) \right]$$

Hence, to prove Eq. (4), it suffices to prove

$$\Pr\left[ I_{D+V_t} > \beta(\delta_t) \right] \leq \delta_t. \tag{18}$$

To prove Eq. (18), we introduce some notions. Let $\boldsymbol{u} \in \mathbb{R}^d$ be a multivariate Gaussian random variable with mean $\boldsymbol{0}$ and covariance $D^{-1}$, which is independent of all the other random variables and we use $\varphi(\boldsymbol{u})$ denote its probability density function. Let

$$P_s^{\boldsymbol{u}} = \exp\left( \boldsymbol{u}^\top \zeta_s - \frac{1}{2} \boldsymbol{u}^\top \Sigma_\pi \boldsymbol{u} \right),$$

$$M_t^{\boldsymbol{u}} \triangleq \exp\left( \boldsymbol{u}^\top S_t - \frac{1}{2} \|\boldsymbol{u}\|_{V_t}^2 \right),$$

and

$$M_t \triangleq \mathbb{E}_{\boldsymbol{u}}[M_t^{\boldsymbol{u}}] = \int_{\mathbb{R}^d} \exp\left( \boldsymbol{u}^\top S_t - \frac{1}{2} \|\boldsymbol{u}\|_{V_t}^2 \right) \varphi(\boldsymbol{u}) du,$$

where $s = 1, \ldots, N_\pi(t-1)$ is the index of exploration round. We have $M_t^{\boldsymbol{u}} = \Pi_{s=1}^{N_\pi(t-1)} P_s^{\boldsymbol{u}}$. In the following, we prove $\mathbb{E}[M_t] \leq 1$.

For any timestep $t$, $\eta_t = \Sigma^{\frac{1}{2}} \varepsilon_t$ is $\Sigma$-sub-Gaussian and $\eta_t$ is independent among different timestep $t$. Then, $\zeta_s = (\boldsymbol{v}_1 \top \Sigma^{\frac{1}{2}} \boldsymbol{\varepsilon}_{s,1}, \ldots, \boldsymbol{v}_{\tilde{d}} \top \Sigma^{\frac{1}{2}} \boldsymbol{\varepsilon}_{s,\tilde{d}})^\top$ is $\Sigma_\pi$-sub-Gaussian. According to the sub-Gaussian property, $\zeta_s$ satisfies

$$\forall \boldsymbol{v} \in \mathbb{R}^d, \ \mathbb{E}\left[ e^{\boldsymbol{v}^\top \zeta_s} \right] \leq e^{\frac{1}{2} \boldsymbol{v}^\top \Sigma_\pi \boldsymbol{v}},$$

which is equivalent to

$$\forall \boldsymbol{v} \in \mathbb{R}^d, \ \mathbb{E}\left[ e^{\boldsymbol{v}^\top \zeta_s - \frac{1}{2} \boldsymbol{v}^\top \Sigma_\pi \boldsymbol{v}} \right] \leq 1.$$

Let $\mathcal{J}_s$ be the $\sigma$-algebra $\sigma(W, \zeta_1, \ldots, W, \zeta_{s-1}, \pi)$. Thus, we have

$$\mathbb{E}\left[ P_s^{\boldsymbol{u}} | \mathcal{J}_s \right] = \mathbb{E}\left[ \exp\left( \boldsymbol{u}^\top \zeta_s - \frac{1}{2} \boldsymbol{u}^\top \Sigma_\pi \boldsymbol{u} \right) | \mathcal{J}_s \right] \leq 1.$$

Then, we can obtain

$$\begin{aligned} \mathbb{E}[M_t^{\boldsymbol{u}} | \mathcal{J}_{N_\pi(t-1)}] &= \mathbb{E}\left[ \Pi_{s=1}^{N_\pi(t-1)} P_s^{\boldsymbol{u}} | \mathcal{J}_{N_\pi(t-1)} \right] \\ &= \left( \Pi_{s=1}^{N_\pi(t-2)} P_s^{\boldsymbol{u}} \right) \mathbb{E}\left[ P_{N_\pi(t-1)}^{\boldsymbol{u}} | \mathcal{J}_{N_\pi(t-1)} \right] \\ &\leq M_{t-1}^{\boldsymbol{u}}, \end{aligned}$$

which implies that $M_t^{\boldsymbol{u}}$ is a super-martingale and $\mathbb{E}[M_t^{\boldsymbol{u}} | \boldsymbol{u}] \leq 1$. Thus,

$$\mathbb{E}[M_t] = \mathbb{E}_{\boldsymbol{u}}[\mathbb{E}[M_t^{\boldsymbol{u}} | \boldsymbol{u}]] \leq 1.$$

According to Lemma 9 in [1], we have

$$M_t \triangleq \int_{\mathbb{R}^d} \exp\left( \boldsymbol{u}^\top S_t - \frac{1}{2} \|\boldsymbol{u}\|_{V_t}^2 \right) \varphi(\boldsymbol{u}) du = \sqrt{\frac{\det D}{\det(D + V_t)}} \exp\left( I_{D+V_t} \right).$$

Thus,

$$\mathbb{E}\left[\sqrt{\frac{\det D}{\det(D + V_t)}} \exp\left(I_{D+V_t}\right)\right] \le 1.$$

Now we prove Eq. (18). First, we have

$$\Pr\left[I_{D+V_t} > \beta(\delta_t)\right] = \Pr\left[\sqrt{\frac{\det D}{\det(D + V_t)}} \exp\left(I_{D+V_t}\right) > \sqrt{\frac{\det D}{\det(D + V_t)}} \exp\left(\beta(\delta_t)\right)\right]$$

$$= \Pr\left[M_t > \frac{1}{\sqrt{\det(I + D^{-\frac{1}{2}} V_t D^{-\frac{1}{2}})}} \exp\left(\beta(\delta_t)\right)\right]$$

$$\le \frac{\mathbb{E}[M_t]\sqrt{\det(I + D^{-\frac{1}{2}} V_t D^{-\frac{1}{2}})}}{\exp\left(\beta(\delta_t)\right)}$$

$$\le \frac{\sqrt{\det(I + D^{-\frac{1}{2}} V_t D^{-\frac{1}{2}})}}{\exp\left(\beta(\delta_t)\right)} \tag{19}$$

Then, for some constant $\gamma > 0$ and for any $a \in \mathbb{N}$, we define the set of timesteps $\mathcal{K}_a \subseteq [T]$ such that

$$t \in \mathcal{K}_a \Leftrightarrow (1 + \gamma)^a \le N_\pi(t - 1) < (1 + \gamma)^{a+1}.$$

Define $D_a$ as a diagonal matrix such that $D_{a,ii} = (1 + \gamma)^a$, $\forall i \in \tilde{d}$. Suppose $t \in \mathcal{K}_a$ for some fixed $a$. Then, we have

$$\frac{1}{1 + \gamma} D_t \preceq D_a \preceq D_t.$$

Let $D = \lambda \Lambda_{\Sigma_\pi} D_a \succeq \frac{\lambda}{1+\gamma} \Lambda_{\Sigma_\pi} D_t$. Then, we have

$$D^{-\frac{1}{2}} V_t D^{-\frac{1}{2}} \preceq \frac{1 + \gamma}{\lambda} D_t^{-\frac{1}{2}} \Lambda_{\Sigma_\pi}^{-\frac{1}{2}} V_t \Lambda_{\Sigma_\pi}^{-\frac{1}{2}} D_t^{-\frac{1}{2}},$$

where matrix $D_t^{-\frac{1}{2}} \Lambda_{\Sigma_\pi}^{-\frac{1}{2}} V_t \Lambda_{\Sigma_\pi}^{-\frac{1}{2}} D_t^{-\frac{1}{2}}$ has $\tilde{d}$ ones on the diagonal. Since the determinant of a positive definite matrix is smaller than the product of its diagonal terms, we have

$$\det(I + D^{-\frac{1}{2}} V_t D^{-\frac{1}{2}}) \le \det(I + \frac{1 + \gamma}{\lambda} D_t^{-\frac{1}{2}} \Lambda_{\Sigma_\pi}^{-\frac{1}{2}} V_t \Lambda_{\Sigma_\pi}^{-\frac{1}{2}} D_t^{-\frac{1}{2}})$$

$$\le \left(1 + \frac{1 + \gamma}{\lambda}\right)^{\tilde{d}} \tag{20}$$

Let $\gamma = e - 1$. Using Eqs. (19) and (20), $\beta(\delta_t) = \ln(1/\delta_t) + \ln\ln t + \frac{\tilde{d}}{2}\ln(1 + e/\lambda) = \ln(t \ln^2 t) + \ln\ln t + \frac{\tilde{d}}{2}\ln(1 + \frac{e}{\lambda})$ and a union bound over $a$, we have

$$\Pr\left[I_{D+V_t} > \beta(\delta_t)\right] \le \sum_a \Pr\left[I_{D+V_t} > \beta(\delta_t) | t \in \mathcal{K}_a, D = \lambda \Lambda_{\Sigma_\pi} D_a\right]$$

$$\le \sum_a \frac{\sqrt{\det(I + D^{-\frac{1}{2}} V_t D^{-\frac{1}{2}})}}{\exp\left(\beta(\delta_t)\right)}$$

$$\leq \frac{\ln t}{\ln(1+\gamma)} \cdot \frac{\left(1+\frac{1+\gamma}{\lambda}\right)^{\frac{\tilde{d}}{2}}}{\exp\left(\ln(t\ln^2 t) + \ln\ln t + \frac{d}{2}\ln(1+\frac{e}{\lambda})\right)}$$

$$= \ln t \cdot \frac{\left(1+\frac{e}{\lambda}\right)^{\frac{\tilde{d}}{2}}}{t\ln^2 t \cdot \ln t \cdot \left(1+\frac{e}{\lambda}\right)^{\frac{\tilde{d}}{2}}}$$

$$= \frac{1}{t\ln^2 t}$$

$$= \delta_t$$

Thus, Eq. (18) holds and we complete the proof of Lemma 2. $\qquad\square$

Now, we give the proof of Theorem 5.

*Proof.* (Theorem 5) First, we bound the number of exploration rounds up to time $T$. Let $\psi(t) = t^{\frac{2}{3}}/d$. According to the condition of exploitation (Line 4 in Algorithm 3), we have that if at timestep $t$ algorithm MC-ETE starts an exploration round, then $t$ satisfies

$$N_\pi(t-1) \leq \psi(t).$$

Let $t_0$ denote the timestep at which algorithm MC-ETE starts the last exploration round. Then, we have

$$N_\pi(t_0 - 1) \leq \psi(t_0)$$

and thus

$$\begin{aligned}
N_\pi(T) &= N_\pi(t_0) \\
&= N_\pi(t_0 - 1) + 1 \\
&\leq \psi(t_0) + 1 \\
&\leq \psi(T) + 1.
\end{aligned}$$

Next, for each timestep $t$, we bound the estimation error of $f(\boldsymbol{w})$. For any $t > 0$, let $\Delta_t \triangleq f(\boldsymbol{w}^*) - f(\boldsymbol{w}_t)$ and $\hat{f}_t(\boldsymbol{w}) \triangleq \boldsymbol{w}^\top \hat{\boldsymbol{\theta}}_t - \rho \boldsymbol{w}^\top \hat{\Sigma}_t \boldsymbol{w}$. Suppose that event $\mathcal{M}_t \cap \mathcal{N}_t$ occurs. Then, according to Lemmas 5 and 6, we have that for any $\boldsymbol{w} \in \triangle_d$,

$$
\left| f(\boldsymbol{w}) - \hat{f}_{t-1}(\boldsymbol{w}) \right| \leq \sqrt{2\beta(\delta_t)} \sqrt{\boldsymbol{w}^\top B_\pi^+ D_{t-1}^{-1} \left(\lambda\Lambda_{\Sigma_\pi} D_{t-1} + \sum_{s=1}^{N_\pi(t-1)} \Sigma_\pi \right) D_{t-1}^{-1}(B_\pi^+)^\top \boldsymbol{w}}
$$

$$
+ 5\rho\|C_\pi^+\| \sqrt{\frac{3\ln t}{2N_\pi(t-1)}}
$$

$$
\leq \sqrt{2\ln(t\ln^2 t) + \ln\ln t + \frac{\tilde{d}}{2}\ln(1+\frac{e}{\lambda})} \cdot \sqrt{\boldsymbol{w}^\top B_\pi^+ D_{t-1}^{-1} \left(\lambda\Lambda_{\Sigma_\pi} + \Sigma_\pi\right)(B_\pi^+)^\top \boldsymbol{w}}
$$

$$
+ 5\rho\|C_\pi^+\| \sqrt{\frac{3\ln t}{2N_\pi(t-1)}}
$$

$$
\leq 7\sqrt{\ln t + \frac{\tilde{d}}{2}\ln(1+\frac{e}{\lambda})} \cdot \left( \sqrt{\boldsymbol{w}^\top B_\pi^+ \left(\lambda\Lambda_{\Sigma_\pi} + \Sigma_\pi\right)(B_\pi^+)^\top \boldsymbol{w}} + \rho\|C_\pi^+\| \right) \cdot \sqrt{\frac{1}{N_\pi(t-1)}}
$$

Let $Z(\rho,\pi) \triangleq \max_{\boldsymbol{w}\in\triangle_d}\left(\sqrt{\boldsymbol{w}^\top B_\pi^+ \left(\lambda\Lambda_{\Sigma_\pi}+\Sigma_\pi\right)(B_\pi^+)\top\boldsymbol{w}}+\rho\|C_\pi^+\|\right)$. Then, we have

$$\left|f(\boldsymbol{w})-\hat{f}_{t-1}(\boldsymbol{w})\right| \leq 7\cdot Z(\rho,\pi)\sqrt{\frac{\ln t + \tilde{d}\ln(1+e/\lambda)}{N_\pi(t-1)}}.$$

Let $\mathcal{L}^{\text{exploit}}(t)$ denote the event that algorithm MC-ETE does the exploitation at timestep $t$. For any $t > 0$, if $\mathcal{L}^{\text{exploit}}(t)$ occurs, we have $N_\pi(t-1) > \psi(t)$. Thus

$$\left|f(\boldsymbol{w})-\hat{f}_{t-1}(\boldsymbol{w})\right| < 7\cdot Z(\rho,\pi)\sqrt{\frac{\ln t + \tilde{d}\ln(1+e/\lambda)}{\psi(t)}}$$

$$= 7\cdot Z(\rho,\pi)\sqrt{d\left(\ln t + \tilde{d}\ln(1+e/\lambda)\right)}\cdot t^{-\frac{1}{3}}$$

According to Lemmas 5,6, for any $t \geq 2$, we bound the probability of event $\neg(\mathcal{M}_t\cap\mathcal{N}_t)$ by

$$\Pr\left[\neg(\mathcal{M}_t\cap\mathcal{N}_t)\right] \leq \frac{6d^2}{t^2}+\frac{1}{t\ln^2 t}$$

$$\leq \frac{6d^2}{t\ln^2 t}+\frac{1}{t\ln^2 t}$$

$$= \frac{7d^2}{t\ln^2 t}$$

The expected regret of algorithm MC-ETE can be divided into two parts, one due to exploration and the other due to exploitation. Then, we can obtain

$$\mathbb{E}[\mathcal{R}(T)] \leq N_\pi(T)\cdot\tilde{d}\Delta_{\max}+\sum_{t=1}^T\mathbb{E}[\Delta_t|\mathcal{L}^{\text{exploit}}(t)]$$

$$\leq(\psi(T)+1)\cdot\tilde{d}\Delta_{\max}+\sum_{t=1}^T\left(\mathbb{E}[\Delta_t|\mathcal{L}^{\text{exploit}}(t),\mathcal{M}_t\cap\mathcal{N}_t]+\mathbb{E}[\Delta_t|\mathcal{L}^{\text{exploit}}(t),\neg(\mathcal{M}_t\cap\mathcal{N}_t)]\cdot\Pr\left[\neg(\mathcal{M}_t\cap\mathcal{N}_t)\right]\right)$$

$$=O\left(\left(\frac{T^{\frac{2}{3}}}{d}+1\right)\cdot\tilde{d}\Delta_{\max}+\sum_{t=1}^T\left(Z(\rho,\pi)\sqrt{d(\ln t+\tilde{d}\ln(1+e/\lambda))}\cdot t^{-\frac{1}{3}}+\Delta_{\max}\cdot\frac{d^2}{t\ln^2 t}\right)\right)$$

$$=O\left(T^{\frac{2}{3}}d\Delta_{\max}+d^2\Delta_{\max}+Z(\rho,\pi)\sqrt{d(\ln T+d^2\ln(1+\lambda^{-1}))}\cdot T^{\frac{2}{3}}+d^2\Delta_{\max}\right)$$

$$=O\left(Z(\rho,\pi)\sqrt{d(\ln T+d^2\ln(1+\lambda^{-1}))}\cdot T^{\frac{2}{3}}+d\Delta_{\max}\cdot T^{\frac{2}{3}}+d^2\Delta_{\max}\right)$$

$$=O\left(Z(\rho,\pi)\sqrt{d(\ln T+d^2\ln(1+\lambda^{-1}))}\cdot T^{\frac{2}{3}}+d\Delta_{\max}\cdot T^{\frac{2}{3}}\right)$$

Choosing $\lambda=\frac{1}{2}$ and using $\Lambda_{\Sigma_\pi}=\Sigma_\pi$, we obtain

$$\mathbb{E}[\mathcal{R}(T)]=O\left(Z(\rho,\pi)\sqrt{d(\ln T+d^2)}\cdot T^{\frac{2}{3}}+d\Delta_{\max}\cdot T^{\frac{2}{3}}\right),$$

where $Z(\rho,\pi)=\max_{\boldsymbol{w}\in\triangle_d}\left(\sqrt{\boldsymbol{w}^\top B_\pi^+\Sigma_\pi(B_\pi^+)\top\boldsymbol{w}}+\rho\|C_\pi^+\|\right)$. $\qquad\square$