# OpenReview forum: "Continuous Mean-Covariance Bandits"
_NeurIPS.cc/2021/Conference — NeurIPS 2021 Poster_

### Official Review · Reviewer_u6Hw · 2021-07-15

**Rating:** 7
**Confidence:** 4

**Summary:**

Existing literature on risk-aware bandits focuses on risk measures of individual
options such as variance. The paper considers decision making problems with correlated
options. A novel Continuous Mean-Covariance Bandit (CMCB) model is proposed. Agent chooses
weight vectors on given options and observes noisy feedback. The agent's
objective is to balance reward and risk, which is measured by option covariance.
Three settings are considered: full-information, semi-bandit, and full-bandit.
Algorithms and regret upper bounds are provided for all the three settings. Lower bounds
are proved for the full-information and semi-bandit setting. The results are validated
using numerical experiments.

**Limitations And Societal Impact:**

1. I think there is a $d^{3/2}$ dependence in the upper bound in SB case while the lower bound
has an interesting dependence on $c$ and $d.$ When $c$ is small, the problem is almost like
the full information setting and the lower bound also has that flavor. The regret upper bound
doesn't seem to take this into consideration. Do you have ideas on how the regret bound can be
improved? More importantly, it might be good to provide intuition on setting the
regularization parameter and how it depends on $c.$

2. Please explain what $\Delta_{\max}$ is in Theorem~5. From the appendix, it seems that is a consequence
of the exploration phase but I couldn't find a definition or an explanation.

3. Is it possible to extend your results to the case where the noise is sub-exponential or heavy-tailed
with bounded covariance? You have used results from [1] for covariance concentration and the results
proved there hold for distributions satisfying a particular definition of sub-Exponential
distribution which is more general than sub-Gaussian.

[1] Perrault, Pierre, Michal Valko, and Vianney Perchet. "Covariance-adapting algorithm for semi-bandits with application to sparse outcomes." Conference on Learning Theory. PMLR, 2020.

**Main Review:**

Originality: The problem considered is interesting and difficult. There is novelty in the
technical analysis of the problem. It is clear how this work differs from previous work
on risk-aware bandit problems. Related work seems to be adequately cited.

Quality: The submission is technically sound. The claims are well supported by the theoretical
analysis and experiments. The authors also provide enough commentary on the
intuition behind their analysis and the results. This is a complete piece of work. The authors
are honest about the limitations and weaknesses of their results.

Clarity: The paper is well written. It is well organized and it adequately informs the reader.
I do have some minor queries in Limitations section and their redressal could make the paper
more informative.

Significance: The results are compelling. The risk-aware bandit literature has been growing
in the past few years and I think this paper has nice techniques as well as open problems
that would be interesting to other researchers. In particular, the paper addresses the
vector analogue of standard mean-variance bandit problem and I think that is non-trivial.

**Time Spent Reviewing:**

4 hours

---

> ### Author Response · Authors · 2021-08-10
> **Response to Reviewer u6Hw**
>
> Thank you very much for your time and effort in reviewing our paper.
>
> 1 (Improvement of $d^{\frac{3}{2}}$)
>
> Improving the factor $d^{\frac{3}{2}}$ in Theorem 3 is an interesting and challenging problem. This dependence comes from the decomposition of the confidence radius along different dimensions. We have tried other techniques for confidence intervals in linear bandits to alleviate this dependence, but did not obtain a better factor since our confidence radius contains the upper confidence bound of covariance matrix. We think that an advanced matrix analysis method that reserves matrix structures and avoids the decomposition along dimensions may help improve the dependence on $d$, and plan to further investigate the problem along this line.
>
>
> The choice of $\lambda$ leads to a tradeoff between the regularization term and the confidence factor. Specifically, a large $\lambda$ will lead to a large bias due to the regularization term, while a small lambda will result in a looser confidence factor. Since MC-UCB uses a worst-case confidence region,
> its performance is robust across different values of $c$. In our experiments, we set $\lambda=1$, which achieves both nice theoretical guarantees and good empirical performance.
>
>
> 2 (Definition of $\Delta_{\text{max}}$)
>
> Thank you for raising the problem. $\Delta_{\text{max}}=f(\boldsymbol{w}^*)-\min_{\boldsymbol{w} \in \triangle_d} f(\boldsymbol{w})$ denotes the maximum gap between the best action and sub-optimal action, i.e., the maximum instantaneous regret of a pull.
> We will add this definition in our revised paper.
>
> 3 (Extension to Sub-exponential Noise)
>
> Thank you for raising this insightful question. [1] considers the discrete (combinatorial) decision space and uses a novel gap-dependent analysis based on the covering-argument technique, which enables them to work for the sub-exponential distributions. However, their analytical techniques cannot be directly applied to our continuous decision space.
>
> In our current work, the construction and analysis of confidence regions are based on the sub-Gaussian distributions. We agree with the reviewer that deriving regret bounds for general sub-exponential distributions is interesting and worth further investigation. It will be a subject of our future work.

---

> > ### Comment · Reviewer_u6Hw · 2021-08-25
> > **Thank you for the response**
> >
> > Thank you for your response. I will stick to my score.

---

> > > ### Author Response · Authors · 2021-08-26
> > > **Further Response to Reviewer u6Hw**
> > >
> > > Thank you very much for your time and effort in reviewing our paper!

---

### Official Review · Reviewer_1ysV · 2021-07-16

**Rating:** 7
**Confidence:** 3

**Summary:**

This paper studies continuous mean covariance bandits under three feedback scenarios, and proposes three algorithms. For full information (FI) and semi-bandit (SB) feedback, the authors also derive matching lower bounds up to logarithmic factors, showing that the proposed algorithms are  essentially optimal. For the full bandit (FB) setting, they propose an Explore then Exploit algorithm, which has a T^(2/3) regret, and lower bound is left open. The experimental section validates the performance of the proposed algorithms for various correlation structures.

**Main Review:**

The problem considered is well motivated and challenging, and the ability to work with arbitrary correlation structure is welcome. The noise is restricted to sub-Gaussian, which is okay for now.

The paper does a good job of presenting the upper and lower bounds and highlighting the analytical novelties involved in the proof. One gets an impression that for the FB setting, there should be a better algorithm than ETE, but the challenges involved are non-trivial.


Overall, this paper considers a good problem and is competently executed.

**Time Spent Reviewing:**

5

---

> ### Author Response · Authors · 2021-08-10
> **Response to Reviewer 1ysV**
>
> Thank you very much for your time and effort in reviewing our paper. We appreciate your positive comments.
>
> We agree with the reviewer that designing algorithms that go beyond ETE for the FB setting is an interesting  direction and will be a subject of our future research.

---

> > ### Comment · Reviewer_1ysV · 2021-09-11
> > **After rebuttal**
> >
> > Thank you for your detailed responses. I retain my score.

---

> > > ### Author Response · Authors · 2021-09-15
> > > **Response to the After-rebuttal Comment of Reviewer 1ysV**
> > >
> > > Thank you so much for your appreciation and effort in reviewing our paper!

---

### Official Review · Reviewer_7HJc · 2021-07-16

**Rating:** 4
**Confidence:** 4

**Summary:**

This paper studied the mean-variance bandit (i.e. the reward function of choosing an action w is $w^\top \theta^\star - \rho w^\top \Sigma^\star w^\top$) with a continuous action space on the probability simplex. The authors considered three settings:

1. Full-information setting, where a noisy observation of $\theta^\star$ could be observed at each round. Here a sqrt(T) regret (up to log factors) is proved, with both upper and lower bounds. The algorithm is simple: collect past observations and purely exploit at every round.

2. Semi-bandit setting, where w can only take a few non-zero entries, and only the noisy entries of $\theta^\star$ chosen according to w are observed. Here a sqrt(dT) regret (up to log factors) is proved, with both upper and lower bounds. The algorithm is a manifestation of the UCB idea: find confidence bounds of $\theta^\star$ and $\Sigma^\star$, then play the action which maximizes the UCB of the reward.

3. Full-bandit setting, where only a noisy version of $w^\top \theta^\star$ could be observed. Here the authors proved an upper bound of the order O(d*T^{2/3}), and the algorithm is to first explore on O(d^2) properly chosen basis vectors to estimate the mean and covariance, and then exploit after gathering sufficiently many observations.

Experimental results are also given at the end.

**Main Review:**

The mean-variance bandit is of relevance in practical scenarios (also listed by the authors under each feedback model), and a continuous decision space is natural. So this problem is of sufficient theoretical and practical interest. However, despite the contributions listed in the summary, I think this paper suffers from the following problems:

1. Both the idea and the analysis are too simple. Both the pure-exploitation algorithm in the full-information scenario and the UCB-type algorithm in the semi-bandit scenario are very standard, also with standard proof techniques. I have to say that there are some nice points in the technical argument (such as the arguments after Eqn. (3)), but those are not sufficient for a solid contribution. The most challenging part is the full-bandit setting, which is the part I wish the authors to do the best job, is not very satisfactory (see next point).

2. The regret analysis for the full-bandit setting is potentially loose without a matching lower bound. This point is important to me, mainly because of my expectation that the current upper bound may not be tight. Currently an explore-then-exploit algorithm is employed, but it is widely known in the bandit literature that this type of algorithm may give a loose regret bound; for example, even for the vanilla multi-armed bandit problem, a naive explore-then-commit policy suffers from the O(T^{2/3}) regret. Therefore, in my opinion it is unfortunate that the authors can only analyze such type of policies, leaving an important question on the tight behavior of the optimal regret.

3. The proof of the lower bound in Theorem 4 is incorrect. At my first looking I was very surprised that the log T factor is also in the lower bound, but after checking the proof I found the mistake in the argument. The main mistake is the assumption above Eqn. (18): the authors essentially used the proof of the gap-dependent lower bound, and assumed that the expected regret under one instance is o(T^a) for any a>0. However this is not the case in the proof of Theorem 4 - a proof by contradiction in Theorem 4 can only make the above assumption for a > 1/2, because a polynomial regret with low degree is still okay. The argument used by the authors is to establish the logarithmic regret lower bound, which cannot be directly applied to argue a polynomial regret lower bound. Also, the little o notation here is very dangerous, because it hides the dependence on other parameters such as c and d. Consequently, now I also don't know of the correct dependence on (c,d) in corrected Theorem 4 now.

4. Continuing on the point of lower bounds, the proof of Theorem 2 seems to be over-complicated. Even for the static version of this problem where we only care about finding a near-optimal action at time T+1, a simple Le Cam's two-point method could result in an \Omega(1/sqrt(T)) lower bound, so they add up to \Omega(sqrt(T)). I do not understand why the current argument involves so much computation and spans several pages...

Post-rebuttal feedback:

Thanks to the authors for the detailed feedback, especially pasting full revised proof in the rebuttal - I appreciate it. However, many of my concerns are not satisfactorily addressed, and I also have some comments on the rebuttal:

1. The authors mentioned a few times in the rebuttal that they achieve *tight* regret bounds that *fully* capture the influences of covariance. However, this is a strong statement and does not seem to be the case to me.

1.1. Tightness. The current regret bounds are provably tight *only* in terms of the dependence on $T$, and even this dependence may not be tight in the most interesting case with fully bandit feedback. In terms of other parameters, such as $d, c$ or some properties of $\theta^*, \Sigma^*$, the tightness is still unknown. So in my opinion I cannot fully appreciate the claimed tightness of the results.

1.2. Influence of covariance. The authors only derived *upper bounds* of the regret involving the influence of covariance. The tightness of these upper bounds is unknown - so in principle these bounds could even be said to be a bit arbitrary. Overall it is too strong a claim to say that this influence is fully captured.

2. Comparison to [Degenne and Perchet 2016] seems a bit problematic. In the rebuttal, the authors mentioned that Theorem 1 improves over the result in [Degenne and Perchet 2016]. However, based on my own reading, I believe that [Degenne and Perchet 2016] considered the semi-bandit setting, while Theorem 1 is on the full information setting. So this comparison is a bit unfair.

3. Contribution of Theorem 2: the authors mentioned in the rebuttal that this is a novel lower bound analysis procedure under the Bayesian environment. However, to the reviewer's expertise, this is mostly a standard Bayes computation, and much simpler (also standard) analysis could also lead to the $\Omega(\sqrt{T})$ lower bound.

4. Contributions of Theorems 1 and 3: I agree with the authors that some steps in the upper bound analysis are nice technical contributions - this is also reflected in my original review. Nevertheless, currently these steps are hidden in the proof, and one cannot fully appreciate these steps unless he/she reads the proof carefully. If these are solid contributions of independent interest, the authors should extract them and present them independently in the main text, in order for the reviewer to assess its significance.

5. Full-bandit setting: in my understanding, this should be the most important part of the paper, as the previous fully-exploit and UCB-type algorithms are somewhat light ideas. The reviewer thinks that a tight dependence on $T$ should at least be obtained in this setting, and the explore-then-commit idea are a little bit too light - bandit is an art of combining exploration and exploitation, so algorithms separating these parts (without sufficient motivation, i.e. optimality guarantees of doing so) are often questionable.

6. Corrected proof of Theorem 4: thank you for providing the detailed proof steps - I did a high-level check and they all seem correct. But as I said in my original review, this is still a standard bandit lower bound analysis, especially when only the right dependence on $T$ is obtained. The authors should at least try to find the right dependence on other parameters including $c$ and $d$.

In summary, the reviewer regrets not being able to recommend acceptance in the current form, and decides to keep my score.


**Time Spent Reviewing:**

3

---

> ### Author Response · Authors · 2021-08-10
> **Response to Reviewer 7HJc - Part 1/2**
>
> Thank you very much for your time and effort in reviewing our paper.
>
> 1 (Novelty of Idea and Analysis)
>
> We first would like to emphasize that the main contributions of our paper include: (1)  formulating a novel risk-aware bandit problem for correlated arms and explicitly quantifying how *arbitrary* covariance structures impact leaning performance in results, (2) designing algorithms for three practical feedback settings that achieve optimal regrets (within logarithmic factors), and (3) developing novel analytical techniques that capture the influences of covariance on regret performance.
>
> Compared to prior works [Degenne and Perchet 2016, Perrault et al. 2020] on covariance-based bandits, which assume extra knowledge on covariance, consider only positively-correlated and independent cases or depend on loose universal upper bounds in results, we allow $arbitrary$ covariance structures without extra assumptions, and achieve tight regret bounds that fully capture the influences of covariance.
>
> Also, to achieve our results, we develop these non-trivial techniques:
>
> (1) In Theorem 1, we obtain an explicit covariance-based risk bound, instead of applying a loose universal covariance upper bound as [Degenne and Perchet 2016]. This enables us to quantify the impact of arbitrary covariance structure in analysis.
>
> (2) In Theorem 2, we contribute a novel lower bound analysis procedure under the Bayesian environment. We analyze the posterior of expected reward vector and bound the perturbation of the optimal action/optimal value due to estimate deviation, which takes into account the boundary constraints and degenerate cases.
>
> (3) In Theorem 3, we establish a tight concentration using the estimated upper confidence bound instead of the universal bound in [Degenne and Perchet 2016], and handle challenges due to different number of observations in different dimensions by integrating confidence radius.
>
>
> 2 (Full-bandit Setting)
>
>  The full-bandit setting is challenging with the mean-covariance metric, and the key difficulty lies in the fact that one needs to estimate the whole covariance matrix using only aggregate feedback. Specifically, in this setting, we only observe a single outcome $y_t=\boldsymbol{w}_t^\top (\boldsymbol{\theta}^*+\boldsymbol{\eta}_t)$, where the aggregate noise $\boldsymbol{w}_t^\top \boldsymbol{\eta}_t$ is composed of $d$ different sub-noises, and we need to estimate the covariance for every sub-noise pair with only a single aggregate feedback.
>
> To our best knowledge, there is no adaptive covariance estimator for such limited feedback in Statistics. The most practical solution is to fix a set of well-designed $\boldsymbol{w}_t$ (ETE), as done in many partially observable bandit problems, e.g., [Lin et al. 2014, Chen et al. 2018].
>
> Nonetheless, we develop novel techniques for the FB setting to explicitly quantify the impacts of covariance on learning performance, e.g., building a covariance-adapting concentration with key matrices $B^+_\pi, C^+_\pi$, and constructing a super-martingale based on the aggregate noise in exploration rounds to bound the concentration. Our results and analytical techniques are both novel and non-trivial, and may find other applications in the field of bandit research.

---

> > ### Author Response · Authors · 2021-08-10
> > **Response to Reviewer 7HJc - Part 2/2**
> >
> > 3 (Proof of Theorem 4)
> >
> > Thank you for your comment regarding the use of the $o(T^a)$ assumption. Below, we will show that the proof can be easily fixed by using the gap-independent analysis procedure in [Auer et al. 2002], which does not require this assumption.
> >
> > Before we present the revised proof, we emphasize that the key contributions of the paper include formulating the first risk-aware bandit problem with option correlations, designing practical algorithms, deriving regret upper bounds that characterize the impact of arbitrary correlation structure on learning performance (Theorems 1,3,5), and demonstrating algorithm optimality with lower bounds (Theorems 2, 4).
> >
> > Theorem 4 aims to demonstrate the optimality of MC-UCB with respect to $T$. Below, we show that the $\tilde{O}(\sqrt{T})$ regret upper bound of MC-UCB does match the lower bound $\Omega(\sqrt{cdT})$ with respect to $T$ (up to logarithmic factors). Therefore, our claim that "MC-UCB is optimal with respect to T (up to logarithmic factors)" remains *correct*, and our contributions of designing optimal algorithms and deriving regret upper bounds that capture the impacts of arbitrary correlation structures are not affected.
> >
> > We now formally present the slightly revised lower bound in Theorem 4 to $\Omega(\sqrt{cdT})$ (just removing the $\sqrt{\log(T)}$ factor) as follows:
> >
> > **Theorem 4.** There exists an instance of CMCB-SB, for which any algorithm has an expected cumulative regret bounded by $\Omega(\sqrt{cdT})$.
> >
> >
> > **Proof.**  First, we construct some instances with $d \geq 4$, $\frac{2}{d} \leq c \leq \frac{1}{2}$, $\Sigma_*=I$ and $\boldsymbol{\theta}_t \sim N(\boldsymbol{\theta}^*, I)$.
> >
> > Let $I_J$ be a random instance constructed as follows: we uniformly choose a dimension $J$ from $[d]$, and the expected reward vector $\boldsymbol{\theta}^*_{J}$ has $\frac{1}{2}+\varepsilon$ on its $J$-th entry and $\frac{1}{2}$ elsewhere, where $\varepsilon \in (0, \frac{1}{2}]$ will be specified later. Let $I_u$ be a uniform instance, where $\boldsymbol{\theta}^*_u$ has all its entries to be $\frac{1}{2}$.
> >
> > Let $\Pr_J[\cdot]$ and $\Pr_u[\cdot]$ denote the probabilities under instances $I_J$ and $I_u$, respectively, and let $\Pr_j[\cdot]=\Pr_J[\cdot|J=j]$. Analogously, $E_J[\cdot]$, $E_u[\cdot]$ and $E_j[\cdot]=E_J[\cdot|J=j]$ denote the expectation operations.
> >
> > Fix an algorithm $\mathcal{A}$. Let $S_t \in \\{\mathbb{R} \cup \\\{\perp\\\}\\}^d$ be a random variable vector denoting the observations at timestep $t$, obtained by running $\mathcal{A}$. Here $\perp$ denotes no observation on this dimension. Let $Q_{\perp}$ denote the distribution on support $\\{\perp\\}$ which takes value $\perp$ with probability 1.
> >
> > In CMCB-SB, if $w_{t,i}>0$, we can observe the reward on the $i$-th dimension, i.e., $S_{t,i}=\theta_{t,i}$; otherwise, if $w_{t,i}=0$, we cannot get observation on the $i$-th dimension, i.e., $S_{t,i}=\perp$.
> > Let $D_J$ be the distribution of observation sequence $S_1, \dots S_t$ under instance $I_J$, and $D_j=D_{J|J=j}$ is the distribution conditioned on $J=j$.
> > Let $D_u$ be the distribution of observation sequence $S_1, \dots S_t$ under instance $I_u$.
> >
> >
> > For any $i \in [d]$, let $N_i=\sum_{t=1}^{T}\mathbb{I}\\\{w_{t,i}>0\\\}$ be the number of pulls that has a positive weight on the $i$-th dimension, i.e., the number of observations on the $i$-th dimension.
> >
> >
> > Following the analysis procedure of [Lemma A.1, Auer et al. 2002], we have
> >
> > $$
> > KL(D_j\\\|D_u)
> > =  \sum_{t=1}^{T} KL(D_u[S_t|S_1,\dots,S_{t-1}]\\\|D_j[S_t|S_1,\dots,S_{t-1}])
> > $$
> > $$
> > =  \sum_{t=1}^{T} \sum_{i=1}^{d} \left(\Pr[w_{t,i}>0] \cdot KL\left(N(\theta^*_{u,i},1) \\\| N(\theta^*_{j,i},1)\right)
> > +\Pr[w_{t,i}=0] \cdot KL( Q_{\perp} \\\| Q_{\perp}) \right)
> > $$
> > $$
> > =  \sum_{t=1}^{T} \left( \Pr[w_{t,j}>0] \cdot KL\left(N(\frac{1}{2},1) \\\| N(\frac{1}{2}+\varepsilon,1)\right)
> > +\sum_{i \neq j}^{d} \Pr[w_{t,i}>0] \cdot KL\left(N(\frac{1}{2},1) \\\| N(\frac{1}{2},1)\right) \right)
> > $$
> > $$
> > =  \frac{1}{2} \varepsilon^2 \cdot \sum_{t=1}^{T} \Pr[w_{t,j}>0]
> > $$
> > $$
> > =   \frac{1}{2} \varepsilon^2 E_u[N_j]
> > $$
> > Here the first equality comes from the chain rule of entropy [Cover and Thomas, 1991]. The second equality is due to that given $S_1,\dots,S_{t-1}$, if $w_{t,i}>0$, the conditional distribution of $S_t$ is $N(\theta^*_{\cdot,i},1)$, where "$\cdot$" refers to the subscript of instances; otherwise, if $w_{t,i}=0$, $S_t$ is  $\perp$ deterministically. The third equality is due to that $\boldsymbol{\theta}^*_u$ and $\boldsymbol{\theta}^*_j$ only have one different entry on the $j$-th dimension.
> >
> >
> > Let $\\\|\cdot\\\|$ with subscript $TV$ denote the total variance distance, and let $KL(\cdot\\\|\cdot)$ denote the Kullback–Leibler divergence.
> > Using Eq. (28) in the analysis of [Lemma A.1, Auer et al. 2002] and Pinsker's inequality, we have
> > $$
> > E_j[N_j] \leq  E_u[N_j] + T \\\|D_j-D_u\\\|_{TV}
> > $$
> > $$
> > \leq  E_u[N_j] + T \sqrt{ \frac{1}{2} KL(D_j\\\|D_u) }
> > $$
> > $$
> > =  E_u[N_j] +\frac{T  \varepsilon}{2} \sqrt{  E_u[N_j] }
> > $$
> >
> > Let $m = \left \lfloor \frac{1}{c} \right \rfloor \leq \frac{d}{2}$ denote the maximum number of positive entries for a feasible action, i.e., the maximum number of observations for a pull.
> > Performing the above argument for all $j \in [d]$ and using $\sum_{j \in [d]} E_u[N_j] \leq mT$, we have
> >
> > $$
> > \sum_{j \in [d]} E_j[N_j] \leq   \sum_{j \in [d]} E_u[N_j] +\frac{T \varepsilon}{2} \sum_{j \in [d]} \sqrt{ E_u[N_j] }
> > $$
> > $$
> > \leq  mT+\frac{T \varepsilon}{2}  \sqrt{d \sum_{j \in [d]} E_u[N_j]}
> > $$
> > $$
> > \leq  mT+\frac{T \varepsilon}{2}  \sqrt{d mT}
> > $$
> > and thus
> > $$
> > \begin{aligned}
> > E_J[N_J] = \frac{1}{d} \sum_{j \in [d]} E_j[N_j]
> > \leq \frac{mT}{d} + \frac{T \varepsilon}{2} \sqrt{\frac{mT}{d}}
> > \end{aligned}
> > $$
> >
> > Letting $\rho \leq \frac{\varepsilon}{2(1-c)}$, the expected reward (linear) term dominates $f(\boldsymbol{w})$, and the best action $\boldsymbol{w}^*$ under $I_J$ has the weight 1 on the $J$-th entry and 0 elsewhere.
> >
> > Recall that $m \leq \frac{1}{c}$. For each pull that has no weight on the $J$-th entry,  algorithm $\mathcal{A}$ must suffer a regret at least
> >
> > $$
> > (\frac{1}{2}+\varepsilon - \rho)-(\frac{1}{2}-\rho \cdot\frac{1}{m})
> > $$
> > $$
> > \geq  \varepsilon-\frac{m-1}{m}\rho
> > \geq  \varepsilon-\frac{m-1}{m} \cdot \frac{\varepsilon}{2(1-c)}
> > \geq  \varepsilon-\frac{m-1}{m} \cdot \frac{\varepsilon}{2(1-\frac{1}{m})}
> > =  \frac{\varepsilon}{2}
> > $$
> >
> > Thus, the regret is lower bounded by
> >
> > $$
> > E[R(T)] \geq  ( T - E_J[N_J]) \cdot \frac{\varepsilon}{2}
> > $$
> > $$
> > \geq (T - \frac{mT}{d} - \frac{T \varepsilon}{2} \sqrt{\frac{mT}{d}} ) \cdot \frac{\varepsilon}{2}
> > $$
> > $$
> > =  \Omega ( T \varepsilon - T \varepsilon^2 \sqrt{\frac{mT}{d}}  ) ,
> > $$
> > where the last equality is due to $m \leq \frac{d}{2}$.
> >
> > Letting $\varepsilon=a_0 \sqrt{ \frac{d}{Tm} }$ for small enough constant $a_0$, we obtain the regret lower bound $\Omega(\sqrt{\frac{dT}{m}})=\Omega(\sqrt{cdT})$. $\square$
> >
> > References:
> > [1] P. Auer, N. Cesa-Bianchi, Y. Freund, R.E. Schapire. The non-stochastic multi-armed bandit problem. SIAM Journal on Computing, 2002.
> > [2] Thomas M. Cover and Joy A. Thomas. Elements of information theory. Wiley, 1991.
> >
> >
> > 4 (Le Cam's Method)
> > Thank you for suggesting this interesting alternative analysis direction. We will certainly look into the proof idea. Note that our main goal in this paper is to formulate a new risk-aware bandit problem with option correlations, design efficient algorithms, and derive novel regret upper bounds that can explicitly characterize the impacts of covariance on learning performance. The provided lower bounds aim to demonstrate the optimality of our algorithms, and could potentially be proven with different proof procedures.

---

> ### Author Response · Authors · 2021-08-28
> **Response to the Additional Comments of Reviewer 7HJc**
>
> Thank you for your reply. We will certainly revise our paper accordingly. Below, we provide our response to your additional comments.
>
> 1 (Tightness & Influences of Covariance)
>
> 1.1 (Tightness)
>
> We clarified that our regret bounds is tight with respect to $T$ in our submission (Lines 235,259,269). We did not claim the tightness for other problem parameters.
>
> Please note that most of works in the risk-aware/covariance-based bandit literature [Sani et al. 2012; Degenne and Perchet 2016; Perrault et al. 2020; Zhu and Tan 2020] also mainly focus on the tightness with respect to $T$. For example, prior covariance-based bandit works [Degenne and Perchet 2016; Perrault et al. 2020]  investigated optimal regrets with respect to $T$, and did not prove the (non-asymptotic) tightness for other problem parameters, e.g., $\Sigma^*,m,\Delta_{i,\text{min}}$, either. Prior mean-variance bandit works [Sani et al. 2012; Zhu and Tan 2020] also focused on the optimality with respect to $T$ and did not provide matching bounds for other problem parameters, e.g., $K,\Delta_{i}, \Gamma_{i,\text{max}}$.
>
> Although the lower bound for the full-bandit setting remains open, we remark that mean-covariance bandit with full-bandit feedback is a very challenging problem due to the additional hardness in estimating the whole covariance matrix under severely limited feedback. To our best knowledge, we are the first to study this problem in the risk-aware bandit literature and provide non-trivial regret guarantees that can capture the influences of covariance. (See Reply 5 for our detailed response to the question on full-bandit setting.)
>
> 1.2 (Influence of Covariance)
>
> What we meant by "fully capture" is that our results capture the covariance influences for  *arbitrary correlation structures*. In contrast, prior covariance-based bandit works [Degenne and Perchet 2016; Perrault et al. 2020] either did not incorporate the actual covariance in their results or only captured the positively-correlated and independent cases. We agree with the reviewer that this should be made more precise. We will modify the claim "fully capture" to "our regret upper bounds capture the influences of arbitrary covariance structures" to avoid confusion.
>
> There is no existing risk-aware bandit work that proves the tightness for covariance factor in both upper and lower bounds, to our best knowledge. Our work has taken the first step in risk-aware bandits to characterize the influences of arbitrary correlation structures in upper bounds, and can serve as the basis for late studies aiming to fully capture the covariance influences in both upper and lower bounds. We also plan to investigate this direction in our future work.
>
> 2 (Comparison to [Degenne and Perchet 2016])
>
> We agree with the reviewer that [Degenne and Perchet 2016] studies different feedback setting from our Theorem 1 and they cannot be directly compared.
>
> In our original rebuttal, we meant that our Theorem 1 explicitly quantifies the impacts of covariance *itself* (for arbitrary correlation structures), instead of using a loose universal covariance upper bound to represent the regret as [Degenne and Perchet 2016].  We have revised the corresponding sentence in our original rebuttal to make it more accurate.
>
> 3 (Theorem 2)
>
> Please note that our proof of Theorem 2 includes a non-trivial analysis for the perturbation of optimal solution/value due to estimate deviation and carefully handles the degeneration cases.
>
> A standard Bayesian computation of the estimate deviation for $\hat{\boldsymbol{\theta}}_t$ cannot be directly applied to prove Theorem 2. This is because our regret definition $f(\boldsymbol{w}^*)-f(\boldsymbol{w}_t)$ is the mean-variance difference between the best action $\boldsymbol{w}^*$ and the chosen action $\boldsymbol{w}_t$,  both under true problem parameters $\boldsymbol{\theta}^*,\Sigma^*$, *instead of* the difference between the true parameters $\boldsymbol{\theta}^*,\Sigma^*$ and estimated parameters $\hat{\boldsymbol{\theta}}_t,\hat{\Sigma}_t$.
>
> To prove Theorem 2, we first calculate the estimate deviation of problem parameter $\Delta \boldsymbol{\theta}_t$, and bound the perturbation of chosen action $\Delta \boldsymbol{w}_t$ due to  deviation $\Delta \boldsymbol{\theta}_t$, and then put $\Delta \boldsymbol{w}_t$ into function $f(\cdot)$ to analyze the disturbance of mean-covariance values $f(\boldsymbol{w}^*)-f(\boldsymbol{w}_t)$ under true problem parameters $\boldsymbol{\theta}^*,\Sigma^*$ (Lines 604-607).
> Moreover, in the analysis of perturbation, we also need to discuss whether the perturbed action $\boldsymbol{w}_t$ falls outside the feasible region. We handle these degeneration cases by constructing high probability events to restrict $\boldsymbol{w}_t$ to stay in the feasible region (Lines 591-602).
>
> We thank the reviewer for raising a potentially simpler proof direction and will certainly look into it. But we still want to remark that, Theorem 2 is given mainly to demonstrate the optimality of MC-Empirical (within logarithmic factors), and our main contributions include the novel formulation of mean-covariance bandit problem and innovative regret analysis that can capture the influences of arbitrary covariance structures.
>
> 4 (Contributions of Theorems 1,3)
>
> For Theorem 1, we have presented a proof sketch in Lines 170-184 and highlighted the novelty of exploiting the property of sample strategy to bound the risk of chosen actions in Lines 160-161, 178-183.
>
> For Theorem 3, we have presented the formal equation of our constructed novel  confidence region in Lines 240-241, and emphasized the novelty of incorporating the estimated covariance into the confidence region rather than applying a universal upper bound as in [Degenne and Perchet 2016] in Lines 246-249.
>
> Due to space limit, we did not include all proof details in the main text. But following the reviewer's suggestion, we will certainly further highlight the crucial proof steps in our main text in order for readers to understand our technical contributions.
>
> 5 (Full-bandit Setting)
>
> Please notice that the mean-covariance bandit in the full-bandit setting is a very challenging problem, where one needs to estimate each entry of the covariance matrix using only an aggregate random outcome that contains $d$ different sub-noises.
> There is no existing adaptive covariance estimator (offline) for such limited feedback in Statistics to our best knowledge. Thus, it is difficult to design adaptive online algorithms for this setting. Despite this hardness, we firstly provide a practical algorithm with a non-trivial regret guarantee that can characterize the covariance influences.
>
> We agree with the reviewer that bandit is an art of combining exploration and exploitation. But we also note that for many difficult bandit problems, e.g., partial monitoring games [Lin et al. 2014; Chaudhuri and Tewari 2016] and contextual bandits with volatile arms [Chen et al. 2018], explore-then-exploit is the most viable solution and also guarantees non-trivial regret performance. A large number of nice bandit works also used the explore-then-exploit policy, e.g.,
>
> [1] Lin et al. Combinatorial partial monitoring game with linear feedback and its applications. ICML, 2014.
> [2] Chaudhuri and Tewari. Phased exploration with greedy exploitation in stochastic combinatorial partial monitoring games. NIPS, 2016.
> [3] Chen et al. Contextual combinatorial multi-armed bandits with volatile arms and submodular reward. NIPS, 2018.
> [4] Dudík et al. Contextual dueling bandits. COLT, 2015.
> [5] Sébastien and Budzinski. Coordination without communication: optimal regret in two players multi-armed bandits. COLT, 2020.
>
> We emphasize that our main contributions in the full-bandit setting include proposing a practical solution and deriving the regret guarantee that captures the impacts of covariance. To achieve this, we also develop novel analytical techniques, e.g., establishing a covariance-adapting concentration with key matrices $B^+_\pi, C^+_\pi$, and constructing a super-martingale based on aggregate noises in exploration rounds to prove the concentration. Our results and analytical techniques are novel and non-trivial for the risk-aware bandit literature.
>
> 6 (Theorem 4)
>
> We will certainly take the improvement of dependence on $c,d$ as a subject of our future work. At the meanwhile, we would like to remark again that most of prior works in the risk-aware/covariance-based bandit literature [Sani et al. 2012; Degenne and Perchet 2016; Cassel et al. 2018; Perrault et al. 2020; Zhu and Tan 2020] mainly investigated the optimality with respect to $T$ and did not provide matching bounds for other problem parameters, e.g., $\Sigma^*,m,\Delta_{i,\text{min}},\Gamma_{i,\text{max}}$, either. In this literature, the tightness for $T$ is the primary focus and the tightness for other parameters is an ever-present problem, which we will also make efforts to solve in our future work.
>
> **Overall Response**
>
> We thank the reviewer for suggesting some potential directions to further improve our work, e.g., the tightness with respect to $c,d$. We still would like to emphasize that our contributions mainly focus on
>
> 1. We formulate a novel risk-aware bandit problem for correlated options and explicitly quantify the impacts of *arbitrary* covariance structures.
> 2. We design algorithms for three practical feedback settings and achieve optimal regrets with respect to $T$ for the FI and SB settings (within logarithmic factors).
> 3. We develop novel analytical techniques for capturing the covariance influences, e.g., exploiting the property of sample strategy to bound the risk of chosen actions (Lines 178-183) and establishing an innovative confidence region with estimated covariance (Lines 240-241).
>
> We believe that the current score '4' does not fully reflect our contributions, and sincerely hope that the reviewer could reconsider his/her rating score. Thank you very much!

---

### Official Review · Reviewer_kCrL · 2021-07-21

**Rating:** 6
**Confidence:** 3

**Summary:**

This paper studies the risk-aware multi-armed bandits problem where arms (options) can be correlated. The authors consider different feedback settings of the problem: full-information (learner observes reward from all arms), semi-bandit (learner only observes reward from a subset of arms), and full-bandit (learner observed only single reward for selected arms)).
The goal is to learn a policy for sequentially selecting a weights vector for different arms (options) that achieves the best trade-off between reward and covariance-based risk.

The authors develop and analyze three algorithms (MC-Empirical (full-information setting), MC-UCB (semi-bandit setting), and MC-ETE (full-bandit setting)) for the risk-aware multi-armed bandits problem with correlated arms. The authors have shown that the problem-independent regret of MC-Empirical and MC-UCB matches with their lower bounds (up to a logarithmic factor), and MC-ETE has sub-linear regret ($O(T^{2/3} )$). Experimental results verify the performance of the proposed algorithms.

**Ethical Concerns:**

I do not find any ethical issue with the paper.

**Limitations And Societal Impact:**

Since this work is a theoretical paper, I do not find any direct negative societal impact.

**Main Review:**

Originality:  This paper considers a new risk-aware multi-armed bandits model named continuous mean-covariance bandit
(CMCB), which considers correlated arms with continuous decision space (selecting weights for different arms). The authors have developed three algorithms for the CMCB problems with different feedback structures with theoretical guarantees.
The authors have adequately cited the prior work and differentiated their work from previous contributions.

Quality: I find the paper technically sound, but I have not checked all the theoretical proofs in the Appendix.  The authors' claims are supported by theoretical analysis and experimental results.

Clarity: The paper is well-organized and well written. But I have a few questions for the authors (based on your response, I may increase my score):
1. Could you explain how the updates happen in the 'Initialize' part (last line in Step 2) of MC-UCB?
2. I think $c$ should be input to MC-UCB. Right?
3. $N_i(t)$ and $J_{t,i}$ is not defined in algorithm. To completeness and avoid confusion, it can be $N_{ii}(t)$ and $J_{t,ii}$.
4. Could you give intuition about how the estimation of rewards and covariance in MC-ETE works?
5. In bandits literature, the 'bandit' feedback means the same as 'full-bandit' feedback. I thought that the 'full-bandit' feedback is different than 'bandit' feedback. So using the 'full-bandit' instead of 'bandit' may cause some confusion.

Significance: This work is a good contribution to risk-aware multi-armed bandits literature.
To the best of my knowledge, the authors are the first to consider risk-aware multi-armed bandits problem with correlated arms. They have given optimal (up to logarithmic factor) algorithms for full-information and semi-bandit setting and an algorithm with sub-linear regret for full-bandit setting.
Other researchers can use the techniques and policies developed in the paper for their future work on similar problems.


Post-rebuttal feedback: After reading other reviewers' comments and responses from authors, I am inclined to keep my score as is.

**Time Spent Reviewing:**

11

---

> ### Author Response · Authors · 2021-08-10
> **Response to Reviewer kCrL**
>
> Thank you very much for your time and effort in reviewing our paper.
>
> 1 (Initialization)
>
> In MC-UCB, the last line of step 2 includes the following update operations.
> (i) For each $i \in [d]$, update the number of observations in the $i$-th dimension, i.e., $N_i(d^2) \leftarrow 2d-1$.
> (ii) For each $i,j \in [d], i \neq j$, update the number of simultaneous observations in the $i$-th and $j$-th dimensions, i.e., $N_{ij}(d^2) \leftarrow 2$ (the assignment of $N_i(d^2), N_{ij}(d^2)$ is due to the initialized $d^2$ pulls).
> (iii) Update the empirical mean $\hat{\boldsymbol{\theta}}^*_{d^2}$ and empirical covariance $\hat{\Sigma}^*_{d^2}$ of these $d^2$ pulls, using the equations in Steps 10,11.
>
> We omitted these details in MC-UCB, because $N_i(d^2)$ and $N_{ij}(d^2)$ are implied by the description of initialized pulls in Step 2, and the calculation of $\hat{\boldsymbol{\theta}}^*_{d^2}$ and $\hat{\Sigma}^*_{d^2}$ is the same as in Steps 10,11.
>
> We will certainly include these details in our revision.
>
> 2 (Input $c$)
>
> Thank you for spotting this typo. Yes, $c$ is a parameter given by the CMCB-SB problem, and is an input of MC-UCB.
>
>
> 3 ($N_i(t)$ and $J_{t,i}$)
>
>  In the description of algorithm MC-UCB (lines 240-241), we have defined $N_i(t)$ and $J_{t,i}$ as the shorthand for $N_{ii}(t)$ and $J_{t,ij}$, respectively. We will change them to $N_{ii}(t)$ and $J_{t,ij}$ to avoid confusion in our revision.
>
>
> 4 (Estimation in MC-ETE)
>
> The intuition of the estimation in MC-ETE is as follows. To handle the limited full-bandit feedback, we fix a set of well-chosen actions. By repeatedly exploring the designed action $\boldsymbol{v}$, we can estimate the aggregate expected reward $\boldsymbol{v}^\top  \boldsymbol{\theta^*}$ and aggregate covariance $\boldsymbol{v}^\top \Sigma^*  \boldsymbol{v}$.
>
> Since the action vector $\boldsymbol{v}$ is known to us (designed by us), we can choose a set of $\boldsymbol{v}$, which can reveal all entries of $\boldsymbol{\theta^*}$ and $\Sigma^*$ (by choosing a set of full rank). Then, by performing a linear transformation using the feature matrices $B^+_{\pi}$ and $C^+_\pi$ of the designed action set, we can effectively transform estimators of the aggregate rewards $\boldsymbol{v}^\top  \boldsymbol{\theta^*}$ and aggregate covariance $\boldsymbol{v}^\top \Sigma^*  \boldsymbol{v}$ to estimators of the true expected reward vector $\boldsymbol{\theta^*}$ and covariance matrix $\Sigma^*$.
>
>
>
> 5 (Bandit feedback)
>
> Thank you for pointing out this problem.
> We used "full-bandit" feedback to distinguish it from "semi-bandit" feedback. We will revise "full-bandit" feedback to "bandit" feedback in our revised paper to avoid confusion.

---

> ### Author Response · Authors · 2021-09-03
> **Response to the Post-rebuttal Feedback of Reviewer kCrL**
>
> Thank you so much for your time and effort in reviewing our paper!

---

### Decision · Program_Chairs · 2021-09-28

**Decision:**

Accept (Poster)

**Comment:**

This paper considers the mean-variance bandit setting with a continuous action space on the probability simplex. They considered three settings: the full information setting, the semi-bandit setting, and the full-bandit setting.

This problem is interesting, and the authors and reviewers have engaged in detailed technical discussions. After reading all the discussions, I regretfully have to say that I cannot recommend acceptance. The major reason is that the technical contribution of this paper is too weak.

Bandit has a very rich literature, and the bar of a good paper in this literature has become very high since it is already very standard to apply well-known techniques to obtain some bounds, and usually in the literature people focus on very refined questions such as getting the best instance dependent bound, getting the exact exponent of the logarithmic factor, etc. It would not cross the threshold if one just has some sublinear rate.

In particular, for this paper, it is quite clear that the authors are not familiar with the general bandit literature, have made mistakes in their claims, and have used overly sophisticated arguments to show simple results. What is more, in the most interesting full-bandit case, the authors only analyzed an extremely simple algorithm with a suboptimal rate.

I would recommend that the authors slow down and study the bandit literature more carefully, and investigate the full-bandit setting with more patience. This is a good question, but the current contribution is not enough for a NeurIPS paper. I believe the authors have the ability to do it, and after it is done, this could be a much stronger submission.

**Consistency Experiment:**

NeurIPS has a long history of experimentation. In 2014, NeurIPS ran an experiment in which 10% of submissions were reviewed by two independent committees to quantify the randomness in the review process. This year, we repeated a variant of this experiment to see how the quality of the review process has changed over time.  This paper was part of the experiment and was therefore assigned to two committees (consisting of reviewers, an Area Chair, and a Senior Area Chair) that reached independent decisions.  If both committees made the same recommendation, this recommendation was followed. If a single committee recommended acceptance, the paper was accepted (with the exception of a few cases in which the other committee identified what we considered a fatal flaw, e.g., an error in a key result).

This copy’s committee reached the following decision: **Reject**

The other committee assigned to the paper recommended **Accept (Poster)**.  You can find the other set of reviews, along with any follow up discussion with the authors here:
https://openreview.net/forum?id=pbAmqUUHsQ